# Efficient and precise single-cell reference atlas mapping with Symphony

Joyce B. Kang [1,2,3,4,5], Aparna Nathan [1,2,3,4,5], Kathryn Weinand[1,2,3,4,5], Fan Zhang [1,2,3,4,5], Nghia Millard[1,2,3,4,5], Laurie Rumker[1,2,3,4,5], D. Branch Moody[3], Ilya Korsunsky[1,2,3,4,5,7 ✉] & Soumya Raychaudhuri [1,2,3,4,5,6,7 ✉]

Recent advances in single-cell technologies and integration algorithms make it possible to construct comprehensive reference atlases encompassing many donors, studies, disease states, and sequencing platforms. Much like mapping sequencing reads to a reference genome, it is essential to be able to map query cells onto complex, multimillion-cell reference atlases to rapidly identify relevant cell states and phenotypes. We present Symphony (https://github.com/immunogenomics/symphony), an algorithm for building large-scale, integrated reference atlases in a convenient, portable format that enables efficient query mapping within seconds. Symphony localizes query cells within a stable low-dimensional reference embedding, facilitating reproducible downstream transfer of reference-defined annotations to the query. We demonstrate the power of Symphony in multiple real-world datasets, including (1) mapping a multi-donor, multi-species query to predict pancreatic cell types, (2) localizing query cells along a developmental trajectory of fetal liver hematopoiesis, and (3) inferring surface protein expression with a multimodal CITE-seq atlas of memory T cells.

[1] Center for Data Sciences, Brigham and Women's Hospital, Boston, MA, USA. [2] Division of Genetics, Department of Medicine, Brigham and Women's Hospital and Harvard Medical School, Boston, MA, USA. [3] Division of Rheumatology, Inflammation, and Immunity, Department of Medicine, Brigham and Women's Hospital and Harvard Medical School, Boston, MA, USA. [4] Department of Biomedical Informatics, Harvard Medical School, Boston, MA, USA. [5] Program in Medical and Population Genetics, Broad Institute of MIT and Harvard, Cambridge, MA, USA. [6] Versus Arthritis Centre for Genetics and Genomics, Centre for Musculoskeletal Research, Manchester Academic Health Science Centre, The University of Manchester, Manchester, UK. [7] These authors contributed equally: Ilya Korsunsky, Soumya Raychaudhuri. ✉email: ikorsunsky@bwh.harvard.edu; soumya@broadinstitute.org

Advancements in single-cell RNA-sequencing (scRNA-seq) have launched an era in which individual studies can routinely profile $10^4$–$10^6$ cells[1–3], and multimillion-cell datasets are already emerging[4,5]. Single-cell resolution enables the discovery and refinement of cell states across diverse clinical and biological contexts[6–11]. To date, most studies redefine cell states from scratch, making it difficult to compare results across studies and thus hampering reproducibility. Coordinated large-scale efforts, exemplified by the Human Cell Atlas (HCA)[12], aim to establish comprehensive and well-annotated reference datasets comprising millions of cells that capture the broad spectrum of cell states. Building these reference atlases requires integrating multiple datasets that may have been collected under different technical and biological conditions. Hence, reference construction requires application of one of many recently developed single-cell integration algorithms[13–19]. Our group previously developed Harmony[15], a fast, accurate, and well-reviewed method[20] that is able to explicitly model complex study design, a property that makes it suitable for integrating complex datasets into reference atlases[21–24]. The potential to define common cell states using reference maps has already been demonstrated[25,26]. For example, we built an integrated reference of ~80,000 single-cell profiles of fibroblasts from human lung, synovium, salivary gland, and intestine and successfully mapped fibroblasts from human skin and mouse synovium, lung, and intestine to analyze conserved states across tissues and species[25]. Once such reference atlases are painstakingly constructed, interpretation of new datasets requires the ability to quickly map single-cell profiles into these reference atlases. This enables interpretation of new datasets by transferring annotations and metadata of interest from nearby reference cells.

Fast mapping of query cells against a large, stable reference is a well-recognized open problem[27] and active area of research[28–30]. One inefficient but accurate approach to project reference and query cells into a joint embedding is to integrate both sets of cells together de novo, resulting in what might be considered a "gold standard" embedding. While this approach is reasonable for relatively small reference datasets, it is intractable for atlas-sized references with millions of cells. It requires users to rebuild the reference for each analysis, which may be computationally challenging and require administratively cumbersome exchanges of large-scale datasets. Furthermore, de novo integration may corrupt the reference embedding once a reference is carefully constructed and annotated. It is instead preferable to freeze the reference when mapping new query cells onto it.

Here, we define reference mapping to mean placing query cells within the same embedding as integrated reference cells without requiring access to the raw data on all individual reference cells. Importantly, this embedding does not take advantage of any particular annotation, such as cell type labels, which may be refined or updated over time. This is in contrast to automated cell type classifiers, such as scmap[31], which assign rigid annotations based on reference datasets in a supervised manner. Reference mapping approaches introduced so far include Seurat[30], which is compatible with Seurat anchor-based integration[18], and scArches, which is compatible with autoencoders such as scANVI[32] and trVAE[33]. These approaches separate reference building, which integrates datasets in the reference into a low-dimensional embedding, from query mapping, which uses a compressed version of the reference to efficiently map cells into the reference embedding. They further contrast with de novo integration methods like BBKNN[34], Seurat anchor-based integration[18], and Harmony[17], which enable reference building but are slow and require access to the raw data and batch information on individual reference cells. High-quality reference mapping requires both a framework to efficiently store an integrated reference, and a fast and accurate procedure to map query datasets.

An ideal reference mapping algorithm must meet several key requirements. First, similar to de novo integration algorithms, it must be able to remove confounding signals due to complex study design in both the reference and query. In addition, it must be

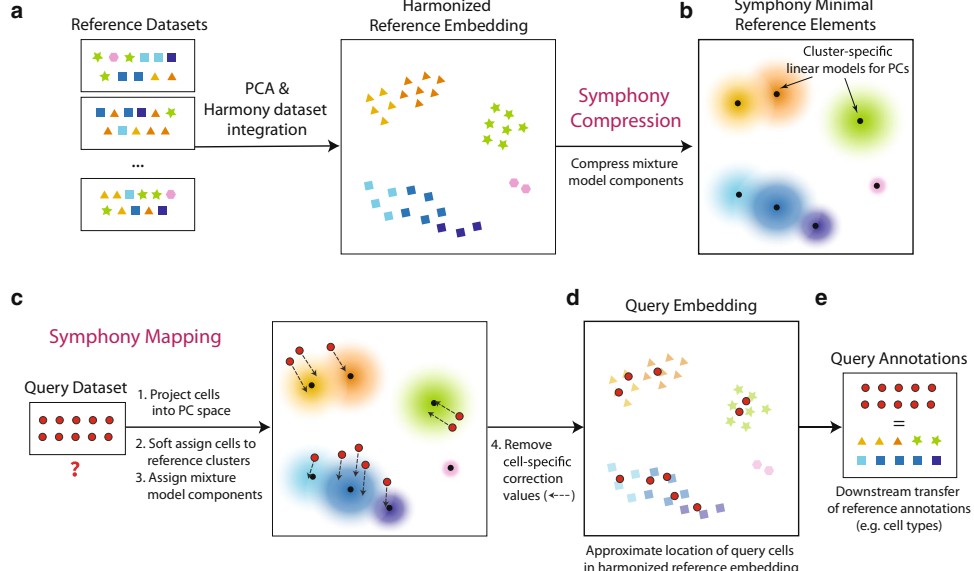

**Fig. 1 Symphony overview.** Symphony comprises two algorithms: Symphony compression (**a**, **b**) and Symphony mapping (**c**, **d**). **a** To construct a reference atlas, cells (colored shapes) from multiple datasets are embedded in a lower-dimensional space (e.g., PCA), in which dataset integration (Harmony) is performed to remove dataset-specific effects. Shape indicates distinct cell types, and color indicates finer-grained cell states. **b** Symphony compression represents the information captured within the harmonized reference in a concise, portable format based on computing summary statistics for the reference-dependent components of the linear mixture model. Symphony returns the minimal reference elements needed to efficiently map new query cells to the reference. **c** Given an unseen query dataset (red circles) and compressed reference, Symphony mapping precisely localizes the query cells to their appropriate locations within the integrated reference embedding (**d**). Reference cell locations do not change during mapping. **e** The resulting joint embedding can be used for downstream transfer of reference-defined annotations to the query cells.

able to scale to large datasets, map with high accuracy, and enable inference of diverse query cell annotations based on reference cells. We present Symphony, a novel algorithm to compress a large, integrated reference and map query cells to a precise location in the reference embedding within seconds. Through multiple real-world dataset analyses, we show that Symphony can enable accurate downstream inference of cell type, developmental trajectory position, and protein expression, even when the query itself contains complex confounding technical and biological effects.

## Results

**Symphony compresses an integrated reference for efficient query mapping.** Symphony comprises two main algorithms: reference compression and mapping (Methods, Supplementary Fig. 1a). Symphony *reference compression* captures and structures information from multiple reference datasets into an integrated and concise format that can subsequently be used to map query cells (Fig. 1a, b). Symphony builds upon the linear mixture model framework first introduced by Harmony[17]. Briefly, in a low-dimensional embedding, such as principal component analysis (PCA), the model represents cell states as soft clusters, in which a cell's identity is defined by probabilistic assignments across one or more clusters. For de novo integration of the reference datasets (using Harmony), cells are iteratively assigned soft-cluster memberships, which serve as weights in a linear mixture model to remove unwanted covariate-dependent effects. Then, Symphony compresses the reference into a mappable entity, leveraging the reference-learned model parameters to add new query cells to the embedding. It maps cells into the reference without any iterative assignment and keeps reference cells stable.

To store the reference efficiently without saving information on individual reference cells, Symphony computes summary statistics learned in the low-dimensional space (Fig. 1b, Methods), returning computationally efficient data structures containing the "minimal reference elements" needed to map new cells. These include the means and standard deviations used to scale the genes, the gene loadings from PCA (or another low-dimensional projection, e.g., canonical correlation analysis [CCA]) on the reference cells, soft-cluster centroids from the integrated reference, and two "compression terms" (a $k \times 1$ vector and $k \times d$ matrix, where $k$ is the number of clusters and $d$ is the dimensionality of the embedding) (Methods, Supplementary Methods, Supplementary Fig. 1b).

To map new query cells to the compressed reference, we apply Symphony *mapping*. The algorithm approximates integration of reference and query cells de novo (Methods), but uses only the minimal reference elements to compute the mapping (Supplementary Fig. 1c). First, Symphony projects query gene expression profiles into the same uncorrected low-dimensional space as the reference cells (e.g., PCs), using the saved scaling parameters and reference gene loadings (Fig. 1c). Second, Symphony computes soft-cluster assignments for the query cells based on proximity to the reference cluster centroids. Finally, to correct unwanted user-specified technical and biological effects in the query data, Symphony assumes the soft-cluster assignments from the previous step and uses stored mixture model components to estimate and regress out the query batch effects (Fig. 1d). Importantly, the reference cell embedding remains stable during mapping. Embedding the query within the reference coordinates enables downstream transfer of annotations from reference cells to query cells, including discrete cell type classifications, quantitative cell states (e.g., position along a trajectory), or expression of missing genes or proteins (Fig. 1e).

**Symphony approximates de novo integration of PBMCs without reintegration of reference datasets.** As we demonstrate in the Methods, Symphony is equivalent to running de novo Harmony integration if three conditions are met: (I) all cell states represented in the query dataset are captured by the reference dataset, (II) the number of query cells is much smaller than the number of reference cells, and (III) the query dataset has a design matrix that is independent of reference datasets (i.e., non-overlapping batches in reference and query). As the scope of available single-cell atlases continues to grow, it is reasonable to assume that reference datasets are large and all-inclusive, making conditions (I) and (II) well-supported. Condition (III) is also typically met if the query data was generated in separate experiments from the reference.

To demonstrate that Symphony mapping closely approximates running de novo integration on all cells, we applied Symphony to 20,571 peripheral blood mononuclear cells (PBMCs) assayed with three different 10x technologies: 3'v1, 3'v2, and 5'. We performed three mapping experiments. For each, we built an integrated Symphony reference from two technologies, then mapped the third technology as a query. The resulting Symphony embeddings were compared to a gold standard embedding obtained by running Harmony on all three datasets together. Visually, we found that the Symphony embedding for each mapping experiment (Fig. 2a) closely reproduced the overall structure and cell type information of the gold standard embedding (Fig. 2b). To quantitatively assess the degrees of dataset mixing we use the Local Inverse Simpson's Index (LISI)[17] metric. For a given categorical label assigned to each cell (in this case, technology), LISI indicates the effective number of categories represented in the local neighborhood of each cell; higher LISI scores correspond to better mixing of cells across batches. LISI scores in Symphony embeddings (mean LISI = 2.12, 95% CI [2.12, 2.13]) and de novo integration embeddings (mean LISI = 2.15, 95% CI [2.14, 2.16]) were nearly identical (Fig. 2c, Methods).

To directly assess similarity of the local neighborhood structures, we computed the correlation between the local neighborhood adjacency graphs generated by Symphony and de novo integration. We define a new metric called k-nearest-neighbor correlation (k-NN-corr), which quantifies how well the local neighborhood structure in a given embedding is preserved in an alternative embedding by looking at the correlation of neighbor cells sorted by distance (Supplementary Fig. 2a–e; Methods). Anchoring on each query cell, we calculate (1) the pairwise distances to its $k$ nearest reference neighbors in the gold standard embedding and (2) the distances between the same query-reference neighbor pairs in the alternate embedding (Methods), then calculate the Spearman correlation between (1) and (2). k-NN-corr ranges from -1 to +1, where +1 indicates a perfectly preserved sorted ordering of neighbors. We find that for $k = 500$, the Symphony embeddings produce a k-NN-corr > 0.4 for 87.7% of cells (and positive k-NN-corr for 99.9% of cells), demonstrating that Symphony not only maps query cells to the correct broad cluster but also preserves the distance relationships between nearby cells in the same local region (Fig. 2d). As a comparison, we calculated k-NN-corr for a simple PC projection of the query cells (with no correction step) using the original reference gene loadings prior to integration and observed significantly lower correlations (Wilcoxon signed-rank $p$ <2.2e-16), with k-NN-corr >0.4 for 50.2% of cells (Supplementary Fig. 2f).

**Symphony enables accurate cell type classification of PBMCs across technologies.** If Symphony is effective, then cells should be mapped close to cells of the same cell type, enabling accurate cell type classification. To test this, we performed post-mapping query cell type classification in the 10x PBMCs example from above. Once

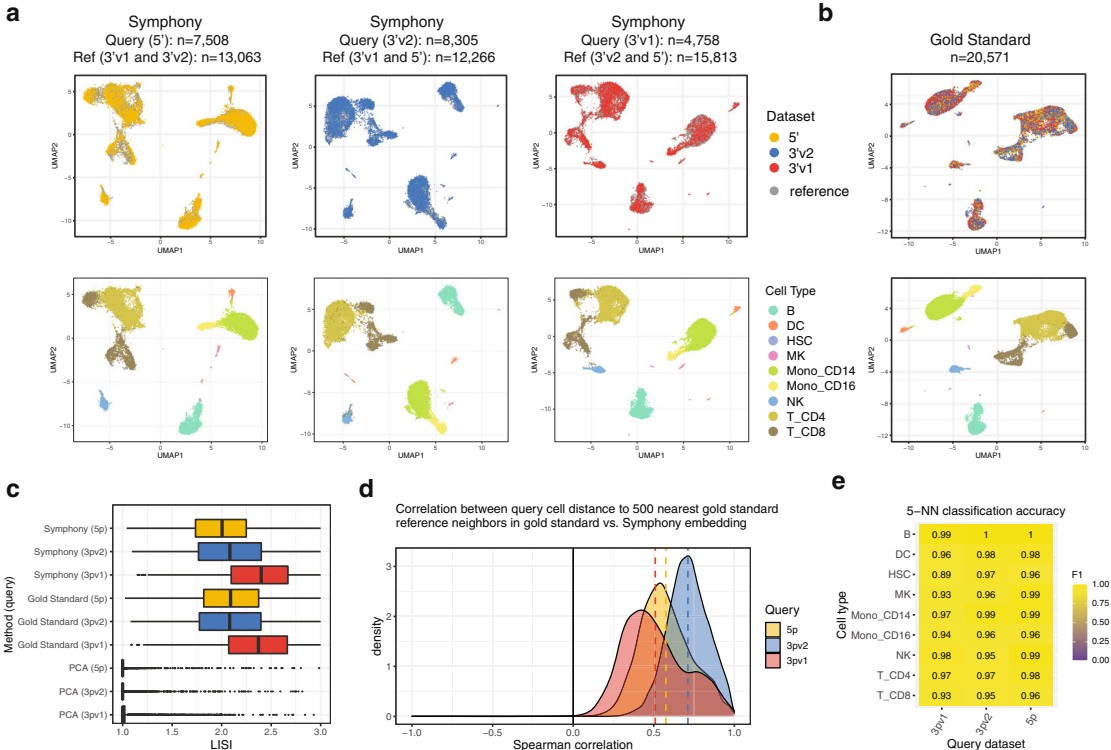

**Fig. 2 Symphony approximates de novo integration without reintegration of the reference cells.** Three PBMC datasets were sequenced with different 10x protocols: 5′ (yellow, n = 7508 cells), 3′v2 (blue, n = 8305 cells), and 3′v1 (red, n = 4758 cells). We ran Symphony three times, each time mapping one dataset onto a reference built from integrating the other two. **a** Symphony embeddings generated across the three mapping experiments (columns). Top row: cells colored by query (yellow, blue, or red) or reference (gray), with query cells plotted in front. Bottom row: cells colored by cell type: B cells (B), dendritic cells (DC), hematopoietic stem cells (HSC), megakaryocytes (MK), CD14 + or CD16 + monocytes (Mono_CD14, Mono_CD16), natural killer cells (NK), or CD4 + or CD8 + T cells (T_CD4, T_CD8), with query cells plotted in front. **b** For comparison, gold standard de novo Harmony embedding colored by dataset (top) and cell type (bottom). **c** Distribution of technology LISI scores for query cell neighborhoods in the Symphony, gold standard, and a standard PCA embeddings on all cells, colored by query dataset. Boxplot center line represents the median; lower and upper box limits represent the 25% and 75% quantiles, respectively; whiskers extend to box limit ±1.5 × IQR; outlying points plotted individually. **d** Distributions of k-NN-corr (Spearman correlation between the distances between the neighbor-pairs in the gold standard embedding and the distances between the same neighbor-pairs in the Symphony embedding) across query cells for k = 500, colored by query dataset. Dotted vertical lines denote mean k-NN-corr. **e** Classification accuracy as measured by cell type F1-scores for query cell type annotation using 5-NN on the Symphony embedding.

query cells are mapped into the reference low-dimensional feature embedding, users can choose any downstream model to predict query labels from the reference cells using their shared harmonized features as input (Methods). To demonstrate this, we used a simple and intuitive k-NN classifier to annotate query cells across 9 cell types based on the majority vote of each query cell's 5 nearest reference neighbor cells in the harmonized embedding and compared the predictions to the ground truth labels assigned a priori with lineage-specific marker genes (Methods, Supplementary Tables 2 and 3). Across all three experiments, predictions using the Symphony embeddings achieved 97.5% accuracy overall, with a median cell type F1-score (harmonic mean of precision and recall, ranging from 0 to 1) of 0.97 (Fig. 2e, Supplementary Table 4). This indicates that Symphony appropriately localizes query cells in harmonized space to enable the accurate transfer of cell type labels.

Automatic cell type classification represents an open area of research[31,35–38]. Existing supervised classifiers assign a limited set of labels to new cells based on training data and/or marker genes. To benchmark Symphony-powered downstream inference against existing classifiers, we followed the same procedure as a benchmarking analysis in Abdelaal et al.[35]. The benchmark compared 22 cell type classifiers on the PbmcBench dataset consisting of two PBMC samples sequenced using 7 different protocols[39]. For each protocol train-test pair (42 experiments) and donor train-test pair

(additional 6 experiments; Methods), we built a Symphony reference from the training dataset then mapped the test dataset. We used the resulting harmonized feature embedding to predict query cell types using three downstream models: 5-NN, SVM with radial kernel, and multinomial logistic regression. The Symphony-based classifiers achieve consistently high cell type F1-scores (average median F1 of 0.79–0.87) comparable to the top three supervised classifiers for this benchmark (scmap-cell, singleCellNet, and SCINA, average median F1 of 0.77–0.83; Fig. 3a and Supplementary Fig. 3). As discussed in Abdelaal et al.[35], some classifiers (including SCINA) leave low-confidence cells as "unclassified." Hence, for the Symphony-based k-NN model, we also enabled the option for Symphony to leave cells as unclassified based on a prediction confidence score (Methods), which measures the proportion of reference neighbors with the winning vote. For this option, we only assigned labels for cells with >60% confidence (which excluded ~14% of cells). Notably, a limitation of this benchmark is that the reference in each experiment consists of a single dataset (no reference integration involved).

**Symphony maps against a large reference within seconds.** To demonstrate scalability to large reference atlases, we evaluated Symphony's computational speed. We downsampled a large

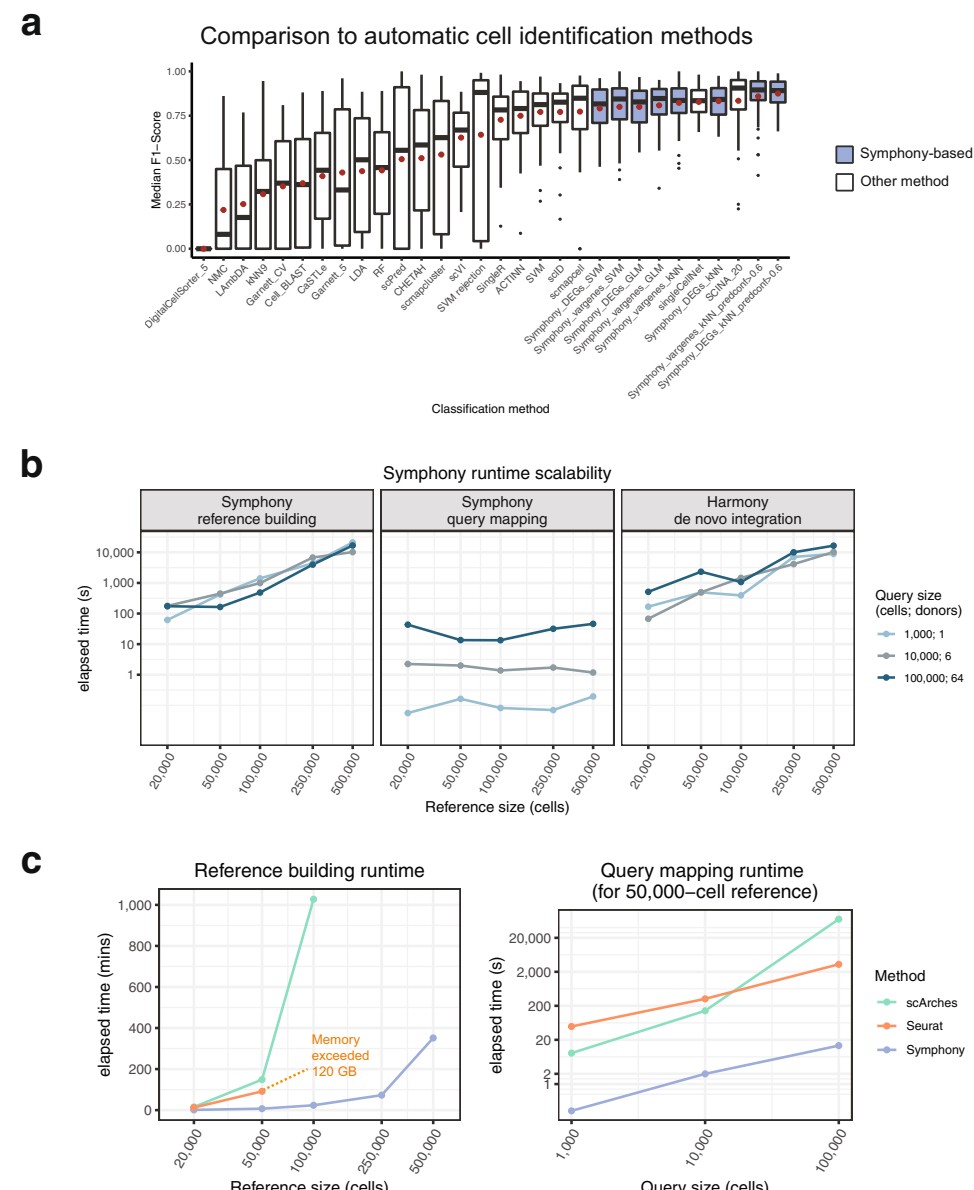

**Fig. 3 Symphony matches performance of top supervised classifiers and maps to large references within seconds. a** Following the cross-technology PBMC benchmarking from Abdelaal et al.[35], we ran a total of 48 train-test experiments per Symphony-based classifier. Two different versions of the Symphony feature embeddings were generated depending on variable gene selection method: top 2000 variable genes (vargenes) or top 20 differentially genes (DEGs) expressed per cell type. Symphony embeddings were used to train 3 downstream classifiers: k-NN ($k = 5$), SVM with radial kernel, and multinomial logistic regression (GLM) with ridge. Symphony (blue) median cell-type F1-scores across 48 train-test experiments compared to supervised methods (white), demonstrating comparability to top supervised methods and stable performance regardless of downstream classification method. For "predconf>0.6" options, only cells with >60% prediction confidence were included ($\geq$4 out of 5 reference neighbors with winning vote). Boxplot center line represents the median (of median F1-scores); lower and upper box limits represent the 25% and 75% quantiles, respectively; whiskers extend to box limit ±1.5 × IQR; outlying points plotted individually. Red dot indicates mean of median F1-scores across 48 experiments (used for ordering along the x-axis). **b** Total elapsed time (in seconds) required to run Symphony reference building starting from gene expression (left), Symphony query mapping starting from query gene expression (middle), or de novo Harmony integration (right) for different-sized reference (x-axis) and query (colors) datasets downsampled from the memory T-cell CITE-seq dataset. **c** Runtime comparison between Symphony, Seurat, and scArches (colors), for building different-sized references (measured in mins) and mapping different-sized queries onto a 50,000-cell reference (measured in secs, plotted on log scale). Note: all methods were run on Linux CPUs (allotting 4 cores each for Symphony and Seurat, 48 cores for scArches). All jobs were allocated a maximum of 120 GB of memory and 24 h of runtime.

memory T-cell dataset[40] to create benchmark reference datasets with 20,000, 50,000, 100,000, 250,000, and 500,000 cells (from 12, 30, 58, 156, and 259 donors, respectively). Against each reference, we mapped three different-sized queries: 1000, 10,000, and 100,000 cells (from 1, 6, and 64 donors) and measured total elapsed runtime (Fig. 3b). The speed of the reference building

process is comparable to that of running de novo integration since they both start with expression data and require a full pipeline of scaling, PCA, and Harmony integration. However, a reference need only be built and saved once in order to map all subsequent query datasets onto it. For instance, initially building a 500,000-cell reference with Symphony took 5163 s (86.1 min)

and mapping a subsequent 10,000-cell query onto it took only 0.99 s, compared to 4806 s (80.1 min) for de novo integration on all cells. Symphony offers a 5000x speedup in this application. Compared to alternative reference mapping approaches Seurat and scArches, Symphony was 1–3 orders of magnitude faster and the only method to scale to large datasets (>100,000 cells) without requiring prohibitive memory (>120 GB) or runtime (>24 h) requirements (Fig. 3c, Methods, Supplementary Table 5). To directly test Symphony's scalability to multimillion cell atlases, we built a reference of 1.39 million cells (270 samples) from a recent COVID-19 dataset[41] in 17.7 h and mapped a held-out query of 72,781 (14 samples) in 11.0 s (Methods, Supplementary Fig. 4). These results show that Symphony scales efficiently to map against multimillion-cell references, enabling it to power potential web-based queries within seconds.

Importantly, Symphony mapping time does not depend on the number of cells or batches in the reference since the reference cells are modeled post-batch correction (Methods); however, it does depend on the reference complexity (number of centroids $k$ and dimensions $d$) and number of query cells and batches (Supplementary Tables 6 and 7) since the query mapping algorithm solves for the query batch coefficients for each of the reference-defined clusters.

### Symphony maps multi-donor, multi-species study to reference of human pancreatic islet cells.

A query dataset might include data from multiple donors, species, and perturbations that create confounding signals obscuring biological signal of interest. Integration algorithms remove these signals in de novo analysis, and it is essential that reference mapping removes them too. Therefore, we designed Symphony to simultaneously handle both tasks: mapping query to reference cells and integration within the query. To test the ability of Symphony to integrate query datasets during mapping, we analyzed reference and query datasets of pancreatic islet cells in which both the reference and query have complex experimental structure (Fig. 4a). The reference contained 5,887 pancreatic islet cells from 32 human donors across four independent studies[42–45], each profiled with a different plate-based scRNA-seq technology (CEL-seq, CEL-seq2, Smart-seq2, and Fluidigm C1). Cell types were previously annotated using cluster-specific marker genes within each reference dataset separately (Methods). The query contained 8569 pancreatic islet cells from four human donors and 1866 cells from two mice, all profiled with inDrop, a droplet-based scRNA-seq technology absent in the reference[46] (Fig. 4b). PCA of the query dataset alone demonstrated the magnitude of the confounding species and donor signals, emphasizing the need for within-query integration (Supplementary Fig. 5a).

Symphony mapped the multi-species, multi-donor, droplet-based query into the reference by effectively and simultaneously removing the effects of species, donor, and technology (Fig. 4c, d); reference mapping obtained superior integration compared to PCA (mean donor LISI = 2.72 compared to 1.45). We predicted that integrating over three nested sources of variation would make it possible to accurately predict query cell types. Using a simple 5-NN classifier in the harmonized embedding, we observed accurate cell-type prediction. Using ground truth labels defined by the original publication[46], we obtained a median cell type F1-score of 0.96 (overall accuracy 96%) for human and median cell type F1 of 0.95 (overall accuracy 91%) for mouse cells (Supplementary Fig. 5c, d and Supplementary Tables 8 and 9), By mapping against a reference, Symphony is able to overcome strong species effects and simultaneously map analogous cell types between mouse and human.

Next, we evaluated the ability of the other reference mapping algorithms, scArches and Seurat, to integrate the same query dataset. For each mapping method, we built a reference using its compatible de novo integration method (Methods, Fig. 4c and Supplementary Fig. 5b). Symphony obtained higher levels of integration than did Seurat and scArches, both between reference and query as well as donors within the query (Fig. 4e, f and Supplementary Table 10). Symphony mapping achieves comparable donor mixing to that of Harmony de novo integration of all five datasets (mean mapping LISI = 2.67 vs. de novo LISI = 2.55 in human, 2.91 vs. 2.7 in mouse, Supplementary Fig. 6c, d). In contrast, the other mapping methods return less integrated embeddings, when compared to their corresponding de novo methods (mean mapping LISI = 2.04 vs. de novo LISI = 2.96 for Seurat in human, 2.46 vs. 3.09 in mouse; 1.12 vs. 2.52 for scArches/trVAE in human, and 1.24 vs. 3.05 in mouse; Supplementary Fig. 6c, d and Supplementary Table 10). Reference mapping should place query cells into the reference embedding, but not at the expense of disrupting the query's original low-dimensional structure. Therefore, we developed a new metric called within-query k-NN correlation (wiq-kNN-corr), which is similar to the k-NN-corr metric but instead measures how well the original query low-dimensional structure is preserved after mapping. Anchoring on each query cell, we calculate it's (1) distances to the $k$ nearest neighbors in the original query PCA embedding within each query batch (in this case, donor) and (2) the distances to those same $k$ cells after reference mapping. Then, wiq-kNN-corr is the Spearman correlation between (1) and (2), ranging between –1 and 1 where higher values represent better retention of the sorted ordering of original neighbors. We observe that for $k = 500$, Symphony and Seurat exhibit nearly identical wiq-kNN-corr (mean wiq-kNN-corr = 0.59 in human, 0.55 in mouse for Symphony; 0.6 in human, 0.57 in mouse for Seurat), whereas scArches performs more poorly on this metric (0.19 in human, 0.13 in mouse) (Fig. 4g). Finally, we evaluated the cell type prediction accuracy of each method (Methods). We observed that Symphony and Seurat performed comparably well, and both outperformed scArches on both human and mouse cell type prediction (Supplementary Fig. 5c, d and Supplementary Tables 8 and 9).

### Localizing query cells along a reference-defined trajectory of human fetal liver hematopoiesis.

A successful mapping method should position cells not only within cell type clusters but also along smooth transcriptional gradients, commonly used to model differentiation and activation processes over time (Fig. 5a). To test Symphony in a gradient mapping context, we built a reference atlas profiling human fetal liver hematopoiesis, containing 113,063 liver cells from 14 donors spanning 7–17 post-conceptional weeks of age and 27 author-defined cell types, sequenced with 10x 3' chemistry (Fig. 5b and Supplementary Fig. 7a)[47]. Trajectory analysis of immune populations with the force directed graph (FDG) algorithm[47] highlights relationships among progenitor and differentiated cell types (Fig. 5c). Notably, the hematopoietic stem cell and multipotent progenitor population (HSC/MPP) branches into three major trajectories, representing the lymphoid, myeloid, and megakaryocyte-erythroid-mast (MEM) lineages. This reference contains two forms of annotation for downstream query inference: discrete cell types and positions along differentiation gradients.

We mapped a query consisting of 21,414 new cells from 5 of the original 14 donors, sequenced with 10x 5' chemistry (Supplementary Fig. 7c). We first inferred query cell types with k-NN classification (Methods) and confirmed accurate cell type assignment based on the authors' independent query

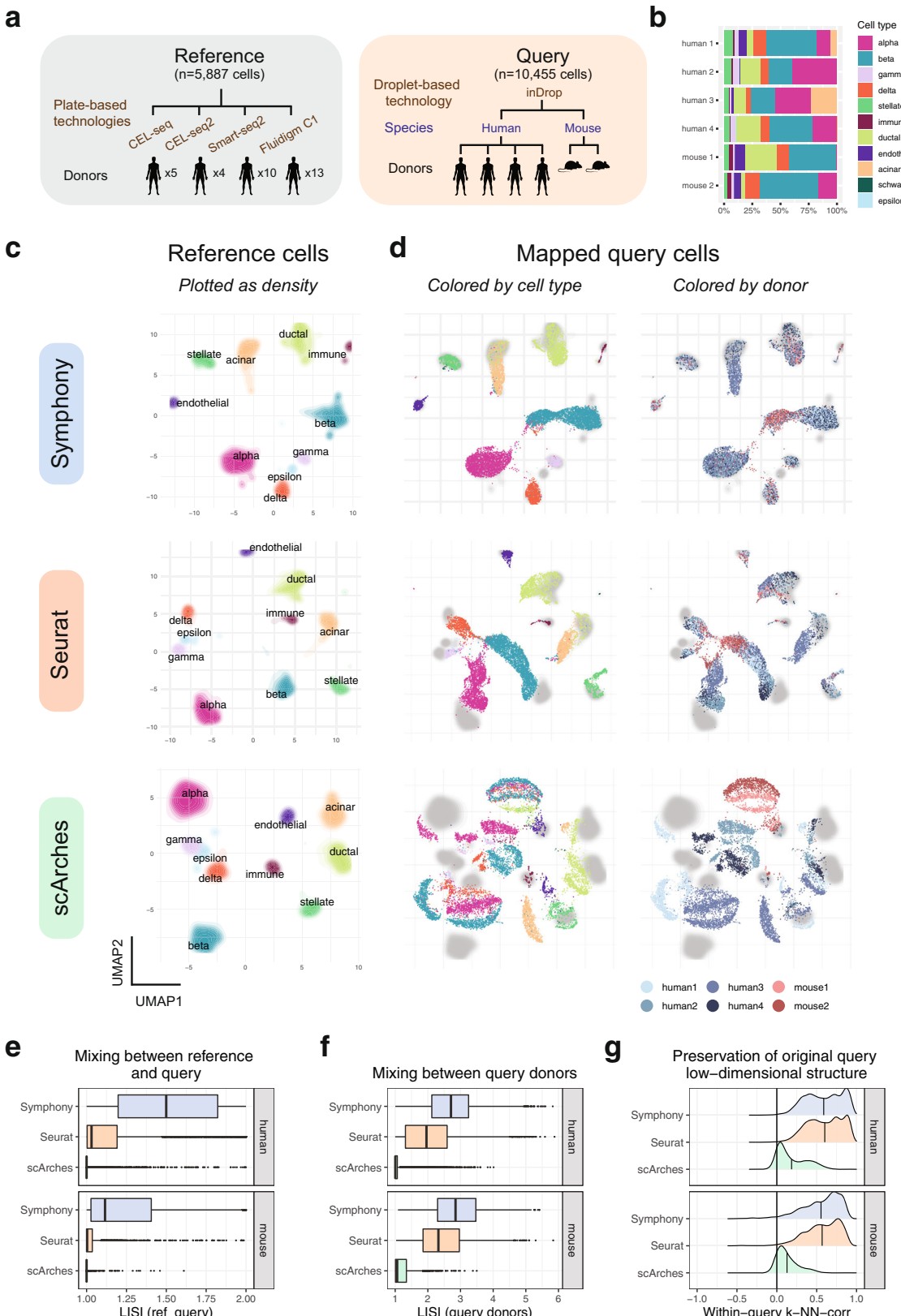

annotations[47], achieving median cell type F1-score of 0.83 and overall accuracy of 85.0% for $k = 30$ (Supplementary Fig. 8a and Supplementary Table 11). Correctly predicted cells generally had a higher proportion of reference neighbors supporting the predicted label (Supplementary Fig. 8b, c). To assess sensitivity to the parameter of $k$ for inference, we tested values of $k$ ranging from 5 to 50 and found that median F1 remained highly stable (0.82–0.84) across choices of $k$ (Supplementary Fig. 8d). To evaluate query trajectory inference, we used the Symphony joint embedding to position query cells from the MEM lineage ($n = 5141$) in the reference-defined trajectory by averaging the FDG coordinates of the 10 nearest reference cells (Supplementary

**Fig. 4 Symphony maps multi-donor, multi-species study to human pancreatic islet cell reference. a** Schematic of mapping experiment with reference ($n = 5887$ cells, 32 donors) built from four human pancreas datasets and query dataset ($n = 10,455$ cells, from four human donors and two mouse donors) sequenced on a new technology (inDrop). **b** Bar plot shows relative proportions of cell types per query donor. We integrated the reference datasets de novo using Harmony, Seurat anchor-based integration, or trVAE, then mapped the query onto the corresponding reference using Symphony, Seurat, or scArches, respectively. UMAP plots of the resulting joint embeddings showing **c** density of integrated reference cells colored by cell type and **d** individual query cells colored by cell type (as defined by Baron et al.[46]) (left) or donor identity (right) with reference densities plotted in the back in gray. Degree of integration for each method was measured by LISI metric between reference and query labels (ref_query) (**e**) and LISI between query donors (**f**) for each query cell neighborhood, faceted by species (human: $n = 8569$ cells from four donors, mouse: $n = 1866$ cells from two donors). Boxplot center line represents the median; lower and upper box limits represent the 25% and 75% quantiles, respectively; whiskers extend to box limit ±1.5 × IQR; outlying points plotted individually. **g** Degree to which the query low-dimensional structure is preserved after mapping, as measured by within-query k-NN correlation (wiq-kNN-corr, with $k = 500$) calculated across all query cells, within each query donor. Vertical lines indicate the mean wiq-kNN-corr.

Fig. 7c). The inferred query trajectory (Fig. 5d) recapitulated known branching from MEM progenitors (MEMPs, brown) into distinct megakaryocyte (green), erythroid (blue, pink), and mast cell (yellow) lineages. Moreover, transitions from MEMPs to differentiated types were marked by gradual changes in canonical marker genes (Fig. 5e): *PPBP* for megakaryocytes, *HBB* for erythrocytes, and *KIT* for mast cells. These gradual expression patterns are consistent with correct placement of query cells along differentiation gradients.

**Symphony helps identify query cell types missing in the reference**. Although the first assumption of Symphony is that the reference is comprehensive, users may not always be aware if their query contains new "unseen" cell states prior to mapping. Symphony will typically map missing query states onto their most similar reference state(s) in these situations. To help users flag unseen cell states, we developed two metrics that help users detect and remove poorly mapping cells (Methods): (1) *per-cell* mapping metric and (2) *per-cluster* mapping metric. These metrics are based on Mahalanobis distance, a multivariate distance metric analogous to the univariate *Z*-score. They measure how far away query cells (1) or user-defined query clusters (2) are from the reference cell states in the low-dimensional embedding, where higher metrics indicate worse mapping.

In general, we found that these metrics were potentially useful for flagging novel cell types (Supplementary Note 1). For example, we tested the metric using the fetal liver hematopoiesis dataset described above and found that the ability to call out a query cell type as novel depends on the cell type as well as what is present in the reference (Supplementary Figs. 9–11). In situations where the missing cell types are very different from the reference (mapping non-immune cell types like fibroblasts, endothelial cells, and hepatocytes onto an immune-only reference), the mapping metrics are able to clearly distinguish the missing cell states as novel (per-cell AUC = 0.997, per-cluster AUC = 1.0, Supplementary Fig. 9). In situations where the novel cell types are very similar to an existing reference cell state, the metrics may have more difficulty in identifying them. For example, when Kupffer cells (specialized tissue-resident liver macrophages) are missing in the reference (Supplementary Fig. 11), they map onto the closely related (immediate precursor) "Monocyte-Macrophage" reference cell state (per-cell AUC = 0.633, per-cluster AUC = 0.963). Our metrics are in general comparable to the Seurat mapping score, though different metrics offer the strongest performance under different scenarios (Supplementary Note 1 and Supplementary Figs. 12 and 13).

**Symphony maps tumor-derived cells onto a healthy atlas**. Given that Symphony maps unseen query cells to their most similar reference type, we hypothesized that Symphony may be able to map tumor-derived cells onto an atlas of corresponding healthy tissue. As an exploratory analysis, we built a reference ($n = 27,203$ cells) of healthy fetal kidney[48] and mapped a renal cell carcinoma (RCC) dataset ($n = 34,326$ cells)[49], transferring reference cell type labels to the query using 10-NN and comparing the predicted labels to the original annotations from Bi et al. (Methods, Fig. 6). As a sanity check, we observed excellent correspondence between the original and predicted annotations for immune and stromal cell types (Fig. 6c). We next examined the mapping results for the cells from the three tumor programs (TP1, TP2, and Cycling Tumor) originally defined by Bi et al. We found that TP1 and TP2 both primarily mapped to the reference "Proximal tubule" cell type and its direct precursor ("Medial S shaped body"); Cycling Tumor primarily mapped to "Medial S shaped body", "Proximal tubule", and "Proliferating distal renal vesicle," concordant with a more actively proliferating phenotype (Fig. 6d). These results are consistent with prior literature, as RCC has been thought to arise from proximal tubule cells[50]. Compared to the immune/stromal compartments, the tumor cells exhibited higher per-cell mapping metrics, indicating that they are less well-represented by the reference (Fig. 6e). This example demonstrates how intentionally mapping novel cell types, such as cancer cells onto a healthy atlas, can potentially provide biologically informative results.

**Extension of Symphony to scATAC-seq data**. We next wondered whether Symphony may be extended to other single-cell modalities, especially scATAC-seq. As a proof-of-concept analysis, we built a reference ($n = 1736$ cells) using a published scATAC-seq dataset of flow-sorted cells capturing hematopoietic differentiation[51,52], leaving out one donor ($n = 298$ cells) to map as a query (Supplementary Fig. 14). We modified Symphony to use the shared open chromatin peaks as input features rather than genes (Methods) and were able to map the query cells such that 84% of cells were assigned their known cell type or the immediate precursor type (Supplementary Fig. 14d, e).

**Inferring query surface protein marker expression by mapping to a reference assayed with CITE-seq**. Recent technological advances in multimodal single-cell technologies (e.g., CITE-seq) make it possible to simultaneously measure mRNA and surface protein expression from the same cells using oligonucleotide-tagged antibodies[53,54]. With Symphony, we can construct a reference from these data, map query cells from experiments that measure only mRNA expression, and infer surface protein expression for the query cells to expand possible analyses and interpretations (Fig. 7a).

To demonstrate this, we used a CITE-seq dataset that measures the expression of whole-transcriptome mRNA and 30 surface proteins on 500,089 peripheral blood memory T cells from 271 samples[40]. We leveraged both mRNA and protein features to build a multimodal reference from 80% of samples ($n = 217$) and

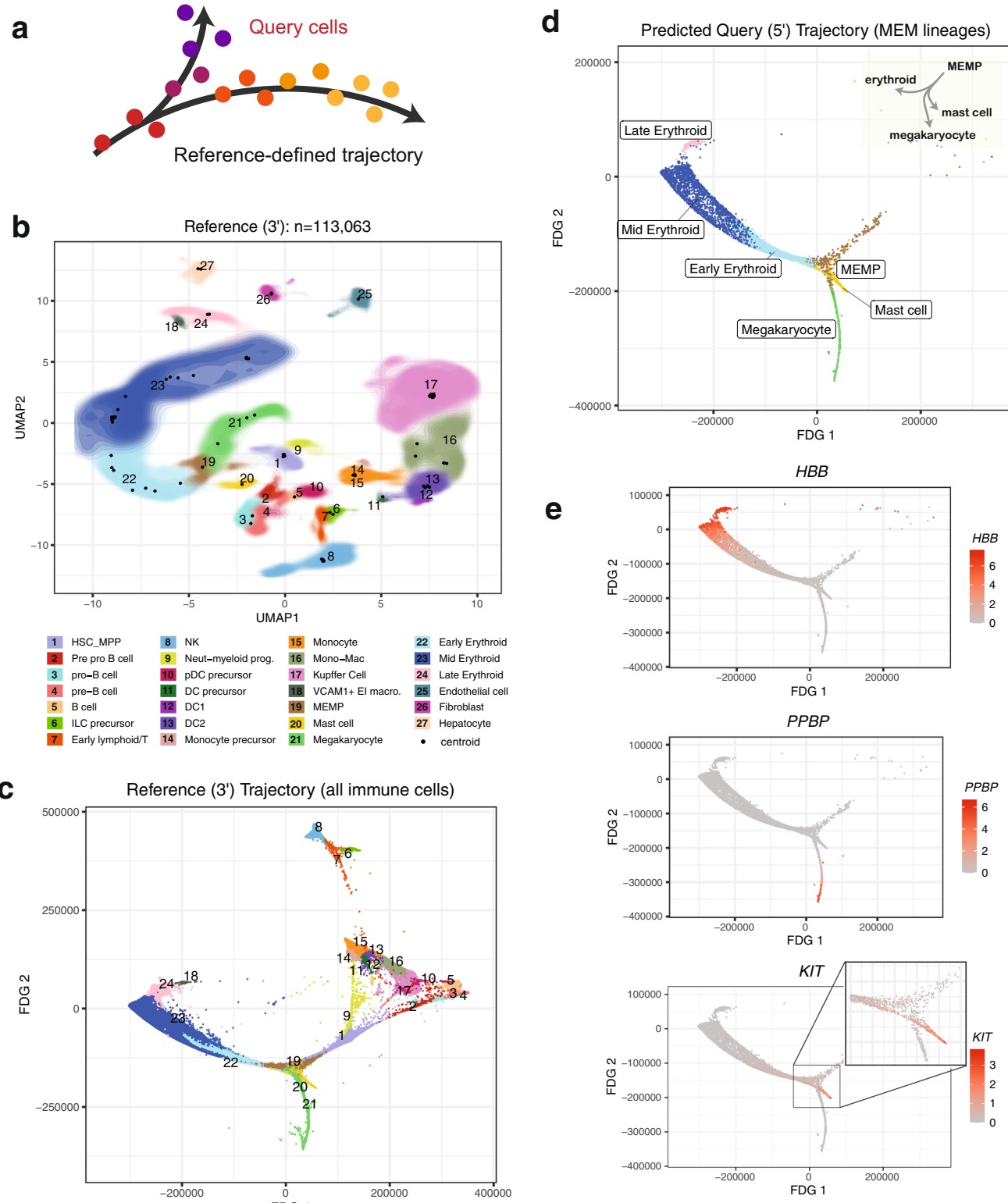

**Fig. 5 Localizing query cells along a trajectory of fetal liver hematopoiesis. a** Schematic showing precise placement of query cells along a continuous reference-defined trajectory. In this example (**b–e**), the reference ($n = 113{,}063$ cells, 14 donors) was sequenced using 10 x 3′ chemistry, and the query ($n = 25{,}367$ cells, 5 donors) was sequenced with 10x 5′ chemistry. **b** Symphony reference colored by cell types as defined by Popescu et al.[47]. Contour fill represents density of cells. Black points represent soft-cluster centroids in the Symphony mixture model. **c** Reference developmental trajectory of immune cells (FDG coordinates obtained from original authors). Query cells in the MEM lineages ($n = 5141$ cells) were mapped against the reference and query coordinates along the trajectory were predicted with 10-NN (**d**). The inferred query trajectory preserves branching within the MEM lineages, placing terminally differentiated states on the ends. **e** Expression of lineage marker genes (*PPBP* for megakaryocytes, *HBB* for erythroid cells, and *KIT* for mast cells). Cells colored by log-normalized expression of gene.

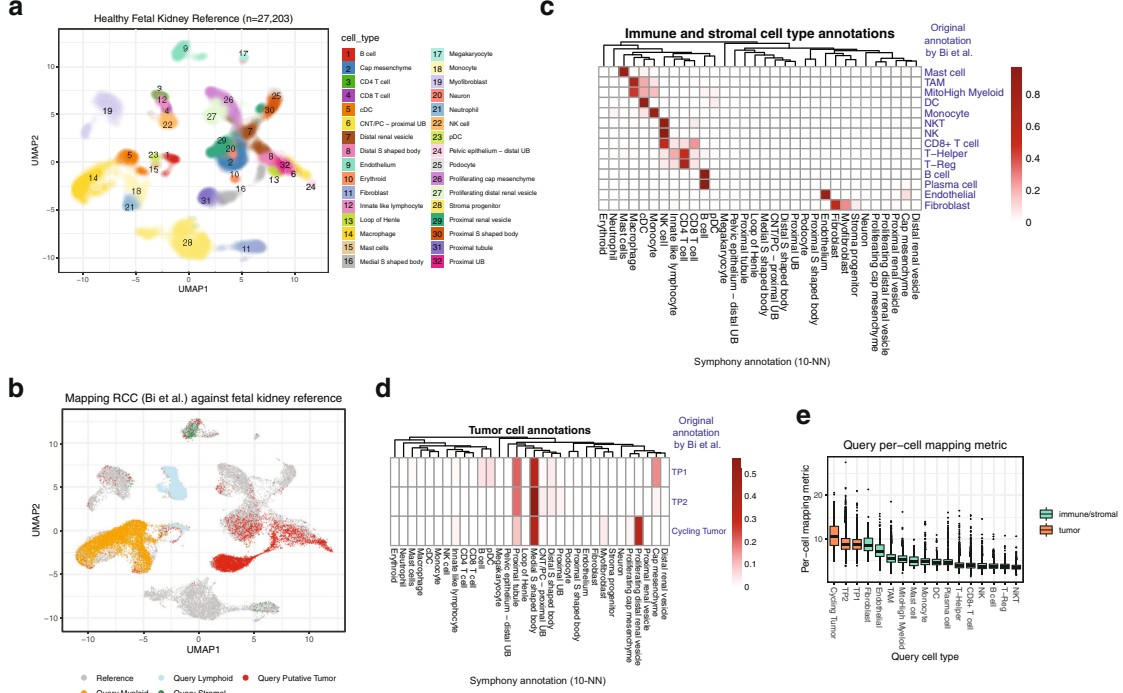

**Fig. 6 Mapping tumor cells onto an atlas of healthy tissue.** We built a reference of healthy fetal kidney[48] and mapped a renal cell carcinoma dataset[49]. **a** UMAP of healthy fetal kidney reference ($n = 27,203$ cells), colored by cell type as defined by the original publication. **b** Mapping tumor query dataset (which contains myeloid, lymphoid, stromal, and tumor compartments) onto the reference. Cells colored by reference (gray) or query compartment (as defined by original authors). **c**, **d** Heatmaps comparing original query cell types (rows), as defined by Bi et al., to the predicted reference cell types from Symphony (columns) for **c** immune and stromal compartments and **d** tumor cells. Color bar indicates the proportion of query cells per original cell type that were predicted to be of each reference type (rows sum to 1). Columns sorted by hierarchical clustering on the average gene expression (all genes) for the cell types to order similar types together. **e** Boxplot of per-cell mapping metric per query cell type (higher values indicate less confidence in the mapping), colored by tumor cells (orange) or immune/stromal (green) as defined in Bi et al. Boxplot shows query cells from 8 donors across 17 cell types: Cycling tumor ($n = 117$ cells), Tumor program 2 (TP2, $n = 4599$), Tumor program 1 (TP1, $n = 3324$), Fibroblast ($n = 91$), Endothelial ($n = 271$), Tumor-associated macrophage (TAM, $n = 5053$), Mitochondrial-High myeloid ($n = 1407$), Mast cell ($n = 39$), Monocyte ($n = 1157$), Dendritic cell (DC, $n = 419$), Plasma cell ($n = 463$), T-Helper ($n = 3284$), CD8 + T cell ($n = 9056$), Natural killer (NK, $n = 2245$), B cell ($n = 962$), T-Regulatory cell (T-Reg, $n = 750$), and Natural killer T cell (NKT, $n = 811$). Boxplot center line represents the median for the cell type; lower and upper box limits represent the 25% and 75% quantiles, respectively; whiskers extend to box limit ±1.5 × IQR; outlying points plotted individually.

map the remaining 20% of samples ($n = 54$). Instead of using PCA, which is best for one modality[55], we used canonical correlation analysis (CCA) to embed reference cells into a space that leverages both. Specifically, CCA constructs a pair of correlated low-dimensional embeddings, one for mRNA and one for protein features, each with a linear projection function akin to gene loadings in PCA. We corrected reference batch effects in CCA space with Harmony and built a Symphony reference (Fig. 7b), saving the gene loadings for the CCA embedding from mRNA features. Then, we mapped the held-out query using only mRNA expression to mimic a unimodal scRNA-seq experiment, reserving the measured query surface protein expression for validation. To mitigate sparsity and variability in detection, we defined ground truth protein values using 50-NN smoothing of the measured values from CITE-seq (i.e., averaging the expression of 50 nearest neighbors in the embedding, Methods). We accurately predicted the surface protein expression of each query cell using the 50-NN average from the nearest reference cells in the harmonized embedding. For all proteins, we found strong concordance between predicted and ground truth expression (Pearson $r$: 0.88–0.99, Fig. 7c, d). For all but three proteins, we achieved comparable results with as few as 5 or 10 nearest neighbors (Supplementary Fig. 15a).

We note that it is also possible to conduct the same analysis with a unimodal PCA-based reference built from the cells' mRNA expression only. This approach has slightly worse performance for some proteins (Pearson $r$: 0.65–0.97, Supplementary Fig. 15b–d), demonstrating that a reference built jointly on both mRNA and protein permits better inference of protein expression than an mRNA-only reference, which is consistent with previous observations that mRNA expression is not fully representative of protein expression[53,54]. This analysis highlights how users can start with a low-dimensional embedding other than PCA, such as CCA, to better capture rich multimodal information in the reference.

## Discussion

Mapping query cells onto large, annotated references in real time and without the need to share sensitive information from the reference datasets is becoming increasingly important for reproducible single-cell analysis. We approached this inherently complex, big-data problem using well-established mathematical methods from integration analysis. We framed reference mapping as a specialized case of integration between one relatively small dataset and a second larger, more comprehensive, and previously integrated dataset. As the reference is already integrated, it is natural to use the same mathematical framework from the integration to perform mapping. For instance, the scArches[28] algorithm uses an autoencoder-based framework to map to references built with autoencoder-based integration algorithms[32,33]. Similarly,

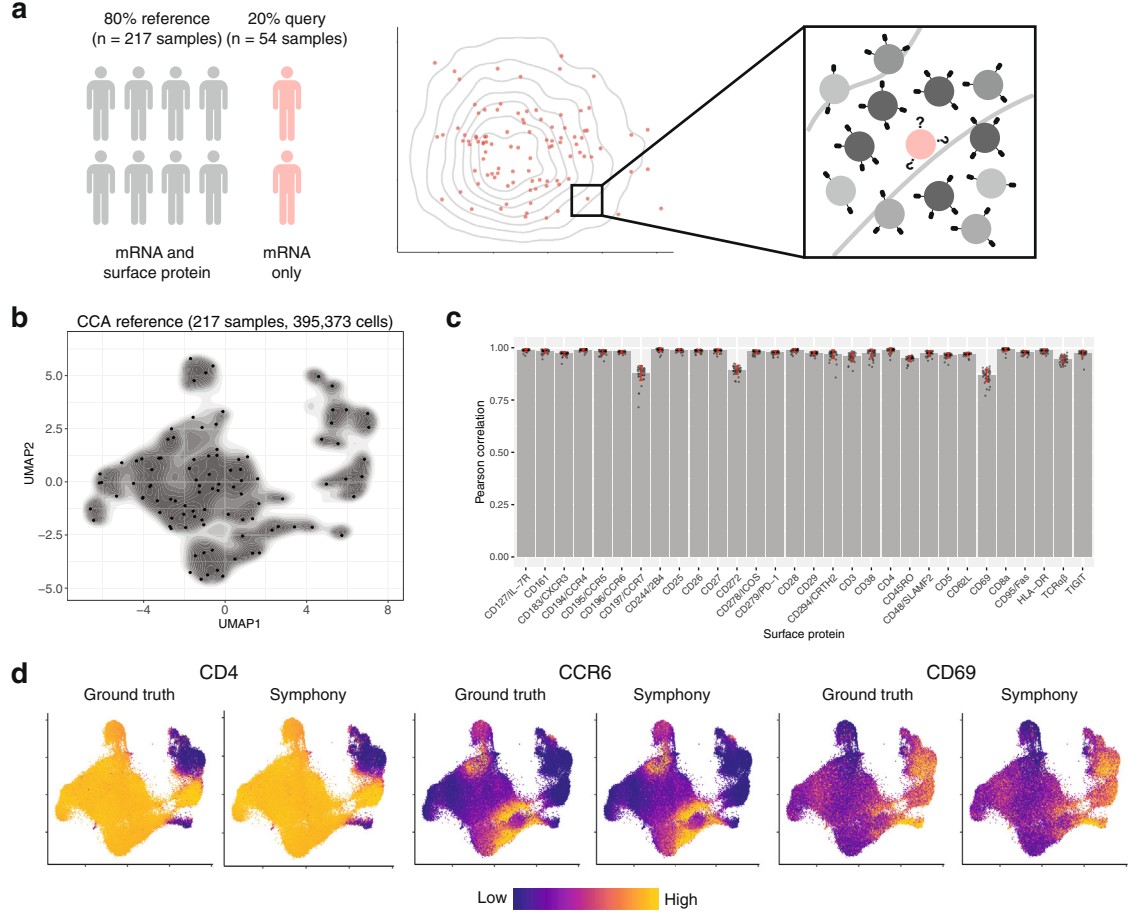

**Fig. 7 Mapping onto a multimodal reference to infer query surface protein expression in memory T cells. a** Schematic of multimodal mapping experiment. The dataset was divided into training and test sets (80% and 20% of samples, respectively). The training set was used to build a Symphony reference, and the test set was mapped onto the reference to predict surface protein expression in query cells (pink) based on 50-NN reference cells (gray). **b** Symphony reference built from mRNA/protein CCA embedding. Contour fill represents density of reference cells ($n = 395,373$ cells from 217 samples). Black points represent soft-cluster centroids in the Symphony mixture model. **c** We measured the accuracy of protein expression prediction with the Pearson correlation between predicted and ground truth expression for each surface protein across query cells in each donor (total $n = 104,716$ cells from 54 samples). Bar height represents the mean per-donor correlation for each protein, error bars represent standard deviation, and individual data points show correlation values per donor. **d** Ground truth and predicted expression of CD4, CCR6, and CD69 based on CCA reference. Ground truth is the 50-NN-smoothed expression measured in the CITE-seq experiment. Colors are scaled independently for each marker from minimum (blue) to maximum (yellow) expression.

Symphony uses the mixture modeling framework to map to references built with Harmony mixture modeling integration. Symphony compresses the reference by extracting relevant reference-derived parameters from the mixture model to map query cells in seconds. With this compression, references can be distributed without the need to share raw expression data or donor-level metadata, which enables data privacy[56]. Symphony compression greatly reduces the size of a reference dataset: for the memory T-cell dataset of 500,089 cells, the raw expression matrix is 8.9 GB, whereas the Symphony minimal reference elements are 1.3 MB.

Useful reference atlases contain annotations absent in the query, such as cell type labels (Fig. 4), trajectory coordinates (Fig. 5), or multimodal measurements (Fig. 7). Reference mapping can also be useful to standardize multiple query datasets derived from different sources into a common embedding or set of labels for downstream analysis, such as testing for differential abundance of cell states between groups (e.g., cases vs. controls)[57–59]. Transfer of annotations from reference to query is an open area of research that includes algorithms for automated cell type classification[31,35–38]. We approach annotation transfer in two steps. We first learn a predictive model in the reference

embedding, and then map query cells and use their reference coordinates to predict query annotations. In this two-step approach, Symphony mapping provides a feature space but is otherwise independent from the choice of downstream inference model. In PBMC type prediction (Fig. 3a), we used Symphony embeddings to train multiple competitive classifiers: k-NN, SVM, and logistic regression. In our specific analyses, we found that a simple k-NN classifier can achieve high performance with only 5–10 neighbors, and modestly outperformed SVM and logistic regression (Fig. 3a). In practice, users can choose more complex inference models if it is warranted for certain annotations. Moreover, we expect prediction results to improve as more standardized annotations emerge, such as pre-defined cell type taxonomies provided by the Cell Ontology[60] project.

Single-cell reference mapping using modalities beyond scRNA-seq poses unique challenges. For example, in scATAC-seq, peaks are not standardized and are typically redefined by peak calling algorithms in each analysis. Hence, it is not immediately clear how to optimally select the best peak features to perform reference mapping when reference and query datasets have been analyzed with different peak sets. One approach may be to remap

query reads to the reference open chromatin regions or binning the genome into small (e.g., 500 or 1 kbp) regions. As another example, multimodal single-cell integration is an important area of active research. For the CITE-seq analysis, we used one strategy (CCA) based on finding shared variation between modalities[40], but alternative approaches have been proposed[30,61] that may be optimal for specific applications.

As mapping is a special case of integration, we expected Symphony mapping to recapitulate the results of de novo Harmony integration. To this end, we defined three conditions under which Symphony and de novo integration with Harmony yield equivalent results. In subsequent examples, we showed that Symphony still performs well when the last two conditions are relaxed. The pancreas query contains more cells than its reference (condition II), while the liver hematopoiesis reference and query overlap in donors (condition III). Condition I, which requires comprehensive cell type coverage in the reference, is less flexible. When the query contains a novel cell type, it will be aligned to its most transcriptionally similar reference cluster. In some cases, this may be advantageous. For example, one can intentionally utilize this behavior to find similar reference cell states, such as mapping tumor cells onto healthy tissue (Fig. 6). Note that condition I only pertains to cell types and not clinical and biological contexts. For instance, we successfully mapped the mouse pancreas query to an entirely human pancreas reference (Fig. 4), because the same pancreatic cell types are shared in both species.

Identification of novel cell-types that have failed to map is an important future direction for mapping algorithms. To identify potentially novel cell-types, we provide two mapping metrics and a prediction confidence score to aid users in flagging and removing poorly mapping cells. We recognize that these metrics may be less informative in cases where the novel population is very similar to an existing reference population. Hence, Symphony does not entirely supplant the need for users interested in novel cell type discovery to conduct de novo analyses of the query alone.

Choosing which reference(s) to use is a key question in a reference-based analysis. When selecting a reference, one should consider (1) the relevance and comprehensiveness of the reference relative to the biological question of interest, (2) similarity of the cell-types being queried, (3) similarity of the technology used to assay the reference versus the query, (4) quality and resolution of cell-level annotations and any associated metadata, including the availability of additional modalities (e.g., CITE-seq), and (5) reference size (number of cells and samples included). For instance, a cell-type-specific embedding like the memory T-cell reference (Fig. 7) may be able to capture more variability within a given cell type compared to an unsorted PBMCs reference (Fig. 2), which may better capture variability across multiple immune populations. Similarly, a reference with only healthy individuals is useful for annotating normal cell types, while a reference with both healthy and diseased individuals is useful for annotating both physiologic and pathologic cell states. It may also be useful to map the query to several references and consider the results in aggregate. For example, one may first map cells to a comprehensive atlas for the tissue or context of interest for coarse-grained annotations, then remap cells from certain cell types onto cell-type-specific references (e.g., T-cell-only) for more fine-grained annotations.

Instead of a single monolithic reference for all cell types across all tissues and disease, we expect the proliferation of multiple, well-annotated specialized references that focus on fine-grained modeling of diverse biological systems. In this initial release of Symphony, we provide eight pre-built reference atlases (Table 1) and an efficient, user-friendly pipeline to facilitate community expansion of high-quality references for the single-cell community. We encourage atlas builders to share their datasets as a mappable reference on open-access data repositories, such as Zenodo. As more large-scale tissue and whole-organism single-cell reference atlases become available in the near future, Symphony will enable investigators to leverage the rich information in these references to perform integrative analyses and transfer reference coordinates and diverse annotations to new datasets in a rapid and reproducible manner.

## Methods

**Symphony overview.** The goal of single-cell reference mapping is to embed newly assayed query cells into an existing comprehensive reference atlas, facilitating the automated transfer of annotations from the reference to the query. The optimal mapping method needs to be able to operate at various levels of resolution, capture continuous intermediate cell states, and scale to multimillion cells[27]. Consider a scenario in which we wish to map a query of $m$ cells against reference datasets with $n$ cells, where $m \ll n$. Unsupervised integration of measurements across donors, studies, and technological platforms is the standard way to compare single-cell datasets and identify cell types. Hence, a "gold standard" reference mapping strategy might be to run Harmony integration on all $m + n$ cells de novo. However, this approach is impractical because it is cumbersome and time-intensive to process all the cell-level data for the reference datasets every time a user wishes to reharmonize it with a query. Instead, we envision a pipeline where a reference atlas need only be carefully constructed and integrated once, and all subsequent queries can be rapidly mapped into the same stable reference embedding.

Symphony is a reference mapping method that efficiently places query cells in their precise location within an integrated low-dimensional embedding of reference cells, approximating de novo harmonization without the need to reintegrate the reference cells. Symphony comprises of two algorithms: reference compression and mapping. Expanding upon the linear mixture model framework introduced in Harmony[17], Symphony compression takes in an integrated reference and faithfully compresses it by capturing the components of the model into efficient data structures. The output of reference compression is the minimal set of elements needed for mapping (Supplementary Fig. 1b). The Symphony mapping algorithm takes as input a new query dataset as well as minimal reference elements and returns the appropriate locations of the query cells within the integrated embedding (Supplementary Fig. 1c).

Once a harmonized reference is constructed and compressed using Symphony, subsequent mapping of query cells executes within seconds. Efficient implementations of Symphony are available as part of an R package at https://github.com/immunogenomics/symphony, along with several precomputed references constructed from public scRNA-seq datasets. The following sections introduce the Symphony model, then describes Symphony compression and mapping in terms of the underlying data structures and algorithms. We also provide Supplementary Methods containing more detailed derivations for reference compression terms.

**Glossary.** We define all symbols for data structures used in the discussion of Symphony below, including their dimensions and possible values. Dimensions are in terms of the following parameters:

$n$ the number of reference cells
$m$ the number of query cells
$N$ the total number of cells ($n + m$)
$g$ the number of genes in the reference after any gene selection
$d$ the dimensionality of the embedding (e.g., PCs); applies to both reference and query.
$b$ the number of batches in the reference
$c$ the number of batches in the query
$k$ the number of clusters in the mixture model for reference integration (representing latent cell states)

**Reference-related symbols:**

| | |
|---|---|
| $\mathbf{G_r} \in \mathbb{R}^{g \times n}$ | Input reference gene expression matrix, prior to scaling. |
| $\mathbf{G_{rs}} \in \mathbb{R}^{g \times n}$ | Scaled reference gene expression matrix. |
| $\mathbf{X_r} \in \{0, 1\}^{b \times n}$ | One-hot design matrix assigning reference cells (columns) to batches (rows). |
| $\mathbf{X'_r} \in \{0\}^{c \times n}$ | Zero matrix assigning reference cells (cols) to *query* batches (rows). All values are 0 because reference cells do not belong to query batches. This term is used in the derivation for the reference compression terms. |
| $\boldsymbol{\mu} \in \mathbb{R}^{g \times 1}$ | Reference gene means used to center each gene for PCA. |
| $\boldsymbol{\sigma} \in \mathbb{R}^{g \times 1}$ | Reference gene standard deviations used to scale each gene for PCA. |

**Table 1 A compendium of pre-built Symphony reference atlases.**

| | Name | Description | Zenodo Link | Data source |
|---|---|---|---|---|
| 1 | 10x PBMCs Atlas | Healthy human PBMCs ($n = 20{,}571$) sequenced using three 10x protocols (3'v1, 3'v2, 5') | Link [https://zenodo.org/record/5090425] | 10x Genomics |
| 2 | Pancreatic Islet Cells Atlas | Pancreatic islet cells ($n = 5887$) from 32 human donors; from 4 separate studies | Link [https://zenodo.org/record/5090425] | Segerstolpe et al.[42], Lawlor et al.[43], Grun et al.[44], Muraro et al.[45] |
| 3 | Fetal Liver Hematopoiesis Atlas | Human fetal liver cells ($n = 113{,}063$) from 14 donors, sequenced with 10x (3') | Link [https://zenodo.org/record/5090425] | Popescu et al.[47] |
| 4 | Healthy Fetal Kidney Atlas | Human fetal kidney cells ($n = 27{,}203$) from 6 samples | Link [https://zenodo.org/record/5090425] | Stewart et al.[48] |
| 5 | Memory T Cell (CITE-seq) Atlas | Human memory T cells ($n = 500{,}089$) from a tuberculosis cohort (259 donors) assayed with CITE-seq | Link [https://zenodo.org/record/5090425] | Nathan et al.[40] |
| 6 | Cross-tissue Fibroblast Atlas | Human fibroblasts ($n = 79{,}148$) from 74 samples spanning 4 inflammatory tissues and corresponding controls | Link [https://sandbox.zenodo.org/record/772596#.YOdFlhNKjIw] | Korsunsky et al.[25] |
| 7 | Cross-tissue Inflammatory Immune Atlas | Immune cells ($n = 307{,}084$) from 125 healthy or disease-affected donors across 6 inflammatory diseases | Link [https://zenodo.org/record/5090425] | Zhang et al.[26] |
| 8 | Tabula Muris Senis (FACS) Atlas | Mouse cells from 23 tissues and organs ($n = 110{,}824$ cells) across the lifespan. | Link [https://zenodo.org/record/5090425] | The Tabula Muris Consortium[62] |

| | |
|---|---|
| $\mathbf{U} \in \mathbb{R}^{g \times d}$ | Gene loadings from the original PCA (before Harmony integration). |
| $\mathbf{Z_r} = \mathbf{\Sigma_r} \mathbf{V_r^T} \in \mathbb{R}^{d \times n}$ | Original (pre-harmonized) PC embedding for reference cells. |
| $\hat{\mathbf{Z}}_\mathbf{r} \in \mathbb{R}^{d \times n}$ | Integrated embedding for reference cells in harmonized PC (hPC) space, as output by Harmony. |
| $\mathbf{R_r} \in [0,1]^{k \times n}$ | Soft cluster assignment of reference cells (cols) to clusters (rows), output by Harmony. Each column is a probability distribution that sums to 1. |
| $\mathbf{Y_{cos}} \in \mathbb{R}^{d \times k}$ | Cluster centroid locations in the harmonized embedding, L2-normalized. |
| $\mathbf{B_r} \in \mathbb{R}^{k \times (1+b) \times d}$ | 3D tensor of the estimated parameters (betas and intercepts) of the linear mixture model for each of $k$ clusters for the reference cells. |
| $\mathbf{N_r} \in \mathbb{R}^{k \times 1}$ | First reference compression term. Vector containing the size of each of the $k$ clusters, effectively the number of reference cells contained within them. |
| $\mathbf{C} \in \mathbb{R}^{k \times d}$ | Second reference compression term. |
| $\text{Ref} = \{\mathbf{\mu}, \mathbf{\sigma}, \mathbf{U}, \mathbf{Y_{cos}}, \mathbf{N_r}, \mathbf{C}\}$ | Set of Symphony minimal reference elements. |

Query-related symbols:

| | |
|---|---|
| $\mathbf{G_q} \in \mathbb{R}^{g \times m}$ | Input query gene expression matrix, prior to scaling. |
| $\mathbf{G_{qs}} \in \mathbb{R}^{g \times m}$ | Query gene expression matrix, scaled by *reference* gene means $\mathbf{\mu}$ and standard deviations $\mathbf{\sigma}$. |
| $\mathbf{X_q} \in \{0,1\}^{c \times m}$ | Design matrix assigning query cells (cols) to query batches (rows). |
| $\mathbf{Z_q} = \mathbf{\Sigma_q} \mathbf{V_q^T} \in \mathbb{R}^{d \times m}$ | Query cell locations in original (pre-harmonized) reference PC embedding. |
| $\hat{\mathbf{Z}}_\mathbf{q} \in \mathbb{R}^{d \times m}$ | Approximate query cell locations in integrated embedding (hPC space). Output of Symphony reference mapping. |
| $\mathbf{R_q} \in [0,1]^{k \times m}$ | Soft cluster assignment of query cells (cols) to clusters (rows). Each column is a probability distribution that sums to 1. |
| $\mathbf{B_q} \in \mathbb{R}^{k \times (1+c) \times d}$ | 3D tensor of the estimated parameters (betas and intercepts) of the linear mixture model for each of $k$ clusters. |

**Symphony model and conditions for equivalence to Harmony integration**.
Symphony and Harmony both use a linear mixture model framework, but the two methods perform different tasks: Harmony integrates a reference, whereas Symphony compresses the reference and enables efficient query mapping. To motivate the Symphony model, it is helpful to first briefly review the mixture model, which serves as the basis. Harmony integrates scRNA-seq datasets across batches (e.g., multiple donors, technologies, studies) and projects the cells into a harmonized embedding where cells cluster by cell type rather than batch-specific effects. Harmony takes as input a low-dimensional embedding of cells (Z) and design matrix with assignments to batches (X) and outputs a harmonized embedding ($\hat{Z}$) with batch effects removed. Briefly, Harmony works by iterating between two subroutines—maximum diversity clustering and linear mixture model correction—until convergence. In the clustering step, cells are probabilistically assigned to soft clusters with a variant of soft k-means with a diversity penalty favoring clusters represented by multiple datasets rather than single datasets. In the correction step, each cluster learns a cluster-specific linear model that explains cell locations in PC space as a function of a cluster-specific intercept and batch membership. Then, cells are corrected by cell-specific linear factors weighted by cluster membership to remove batch-dependent effects. The full algorithm and implementation are detailed in Korsunsky et al.[17].

In the scenario of mapping $m$ query cells against $n$ reference cells, the de novo integration strategy would model all cells as in Eq. (1), where the $H$ subscript denotes the Harmony solution, in contrast to the Symphony model, which is presented in Eq. (2). Let $\mathbf{X_H} \in \{0,1\}^{(c+b) \times (m+n)}$ represent the one-hot encoded design matrix assigning all cells across batches. $\mathbf{X_H^*}$ denotes $\mathbf{X_H}$ augmented with a row of 1 s for the batch-independent intercept term: $\mathbf{X_H^*} = \mathbf{1} || \mathbf{X_H}$. The intercept terms represent cluster centroids (location of "experts" in the mixture of experts model). $\mathbf{Z_H}$ represents the low-dimensional PCA embedding of all cells. $\mathbf{R_H}$ represents the probabilistic assignment of cells across $k$ clusters, and $\text{diag}(\mathbf{R_{Hk}}) \in \mathbb{R}^{N \times N}$ denotes the diagonalized $k$th row of $\mathbf{R_H}$. For each cluster $k$, the parameters of the linear mixture model $\mathbf{B_k} \in \mathbb{R}^{(1+c+b) \times d}$ can therefore be solved for as in Eq. (1), using ridge regression with ridge penalty hyperparameter $\lambda$. Note that we do not penalize the batch-independent intercept term: $\lambda_0 = 0, \forall_{a \in [1:(c+b)]} \lambda_a = 1$.

De novo Harmony model:

$$\mathbf{B_k} = \left(\mathbf{X_H^*} \text{diag}(\mathbf{R_{Hk}}) \mathbf{X_H^{*T}} + \lambda \mathbf{I}\right)^{-1} \mathbf{X_H^*} \text{diag}(\mathbf{R_{Hk}}) \mathbf{Z_H^T} \tag{1}$$

The goal of Symphony mapping is to add new query cells to the model in order to estimate and remove the query batch effects. Symphony mapping approximates de novo Harmony integration on all cells, except the reference cell positions in the

---

**Table 2 Symphony minimal reference elements vs. additional components of Harmony reference.**

**Symphony minimal reference elements**

| | |
|---|---|
| $\boldsymbol{\mu} \in \mathbb{R}^{g \times 1}$ | Reference gene means used to center each gene for PCA. |
| $\boldsymbol{\sigma} \in \mathbb{R}^{g \times 1}$ | Reference gene standard deviations used to scale each gene for PCA. |
| $\mathbf{U} \in \mathbb{R}^{g \times d}$ | Gene loadings to project from expression to PCA (or CCA) space. |
| $\mathbf{Y}_{cos} \in \mathbb{R}^{d \times k}$ | Cluster centroid locations in harmonized PC space, L2-normalized. |
| $\mathbf{N_r} \in \mathbb{R}^{k \times 1}$ | First reference compression term. Vector containing the size of each of the $k$ clusters, effectively the number of reference cells contained within them. |
| $\mathbf{C} \in \mathbb{R}^{k \times d}$ | Second reference compression term. |

**Additional components of a full Harmony reference**

| | |
|---|---|
| $\mathbf{G_r} \in \mathbb{R}^{g \times n}$ | Input reference gene expression matrix, prior to scaling. |
| $\mathbf{X_r} \in \{0,1\}^{b \times n}$ | Design matrix assigning reference cells (cols) to reference batches (rows). |
| $\mathbf{B_r} \in \mathbb{R}^{k \times (1+b) \times d}$ | 3D tensor of the estimated parameters (betas and intercepts) of the linear mixture model for each of $k$ clusters for the reference cells. |
| $\hat{\mathbf{Z}}_r \in \mathbb{R}^{d \times n}$ | Integrated embedding for reference cells in harmonized PC (hPC) space, as output by Harmony. |
| $\mathbf{R_r} \in [0,1]^{k \times n}$ | Soft cluster assignment of reference cells (cols) to clusters (rows), as output by Harmony. Each column is a probability distribution that sums to 1. |

---

**Table 3 Components of Symphony query.**

| | |
|---|---|
| $\mathbf{G_q} \in \mathbb{R}^{g \times m}$ | Input query gene expression matrix, prior to scaling. |
| $\mathbf{X_q} \in \{0,1\}^{c \times m}$ | Design matrix assigning query cells (cols) to query batches (rows). |
| $\mathbf{Z_q} \in \mathbb{R}^{d \times m}$ | Query cell locations in original (pre-harmonized) PC embedding. |
| $\hat{\mathbf{Z}}_q \in \mathbb{R}^{d \times m}$ | Approximate query cell locations in integrated embedding (hPC space). |
| $\mathbf{R_q} \in [0,1]^{k \times m}$ | Soft cluster assignment of query cells (cols) to clusters (rows). Each column is a probability distribution that sums to 1. |
| $\mathbf{B_q} \in \mathbb{R}^{k \times (1+c) \times d}$ | 3D tensor of the estimated parameters (betas and intercepts) of the linear mixture model for each of $k$ clusters. |

---

harmonized embedding do not change. In order for Symphony mapping to be equivalent to de novo Harmony, several conditions must be met:

I. All cell states represented in the query dataset are captured by the reference datasets—i.e., there are no completely novel cell types in the query.
II. The number of reference cells is much larger than the query ($m \ll n$).
III. The query dataset is obtained independent of the reference datasets—i.e., the reference batch design matrix ($\mathbf{X_r}$) has no interaction with the query batch design matrix ($\mathbf{X_q}$).

We consider these to be fair assumptions for large-scale reference atlases, allowing Symphony to make three key approximations:

(1) With a large reference, the reference-only PCs approximate the PCs for the combined reference and query datasets. This allows us to project the query cells into the pre-harmonized reference PCA space using the reference gene loadings ($\mathbf{U}$).
(2) The cluster centroids ($\mathbf{Y}$) for the integrated reference cells approximate the cluster centroids from harmonizing all cells.
(3) The reference cell cluster assignments ($\mathbf{R_r}$) remains approximately stable with the addition of query cells.

Given these approximations, we can thereby harmonize the reference cells a priori and save the reference-dependent portions of the Harmony mixture model (Supplementary Methods). In Symphony, we model the reference cells as already harmonized with batch effects removed, so we can thereafter ignore the reference design matrix structure. The Symphony design matrix $\mathbf{X} \in [0,1]^{c \times N}$ assigns all cells (reference and query) to *query* batches only. $\mathbf{X}^*$ denotes $\mathbf{X}$ augmented with a row of 1 s ($\mathbf{X}_{[0,]}^*$) corresponding to the batch-independent intercepts (we model the intercepts for all cells). The remaining $c$ rows ($\mathbf{X}_{[1:c,]}^*$) represent the one-hot batch assignment of the cells among the $c$ query batches. Note that for the reference cell columns, these values are all 0 since the reference cells do not belong to any *query* batches. The parameters ($\mathbf{B_{qk}} \in \mathbb{R}^{(1+c) \times d}$) of the model for each cluster $k$ can then be solved for as in Eq. (2). Similar to Harmony, we use ridge regression penalizing the non-intercept terms, where $\lambda_0 = 0, \forall_{a \in [1:c]} \lambda_a = 1$.

Symphony model:

$$\mathbf{B_{qk}} \approx \left( \mathbf{X}^* \mathrm{diag}(\mathbf{R_k}) \mathbf{X}^{*T} + \lambda \mathbf{I} \right)^{-1} \mathbf{X}^* \mathrm{diag}(\mathbf{R_k}) \mathbf{Z}^T \quad (2)$$

The matrix $\mathbf{R} \in \mathbb{R}^{k \times N}$ denotes the assignment of query and reference cells (columns) to the reference clusters (rows). $\mathbf{Z} \in \mathbb{R}^{d \times N}$ denotes the horizontal matrix concatenation of the uncorrected query cells in original PC space ($\mathbf{Z_q}$) and corrected reference cells in harmonized space ($\hat{\mathbf{Z}}_r$). For each cluster $k$, let matrix $\mathbf{B_{qk}} \in \mathbb{R}^{(1+c) \times d}$ represent the query parameters to be estimated. The first row of $\mathbf{B_{qk}}$ represents the batch-independent intercept terms, and the remaining $c$ rows of

$\mathbf{B_{qk}}$ represent the query batch-dependent coefficients, which can be regressed out to harmonize the query cells with the reference. Note that the intercept terms from Symphony mapping should equal the cluster centroid locations from the integrated reference since the harmonized reference cells are modeled only by a weighted average of the centroid locations for the clusters over which it belongs (and a cell-specific residual). Hence, the reference cell positions should not change when removing query batch effects.

The matrices $\mathbf{X}^*$, $\mathbf{R_k}$, and $\mathbf{Z}$ in Eq. (2) can be partitioned into query and reference-dependent portions. In the Supplementary Methods, we show in detail how the reference-dependent portions can be further simplified into a $k \times 1$ vector and $k \times d$ matrix ($\mathbf{N_r}$ and $\mathbf{C}$), which we call the reference compression terms. Intuitively, the vector $\mathbf{N_r}$ contains the size (in cells) of each reference cluster. The matrix $\mathbf{C} = \mathbf{R_r} \hat{\mathbf{Z}}_r^T$ does not have as intuitive an explanation but follows from the derivation (Supplementary Methods). These terms can be computed at the time of reference building and saved as part of the minimal reference elements to reduce the necessary computations during mapping.

**Reference building and compression.** Reference compression is the key idea that allows for the efficient mapping of new query cells into the harmonized reference embedding without the need to reintegrate all cells. To construct a Symphony reference with minimal elements needed for mapping, reference cells are first harmonized in a low-dimensional space (e.g., PCs) to remove batch-dependent effects. Symphony then compresses the Harmony mixture model components to be saved for subsequent query mapping.

Symphony takes as input a gene expression matrix for reference cells ($\mathbf{G_r}$) and corresponding one-hot-encoded design matrix ($\mathbf{X_r}$) containing metadata about assignment of cells to batches. It outputs a set of data structures, referred to as the Symphony "minimal reference elements", that captures key information about the reference embedding that can be subsequently used to efficiently map previously unseen query cells (Algorithm 1). These components include the gene means ($\boldsymbol{\mu}$) and standard deviations ($\boldsymbol{\sigma}$) used to scale the genes, the PCA gene loadings ($\mathbf{U}$), the final L2-normalized cluster centroid locations ($\mathbf{Y}_{cos}$), and precomputed values which we call the "reference compression terms" ($\mathbf{N_r}$ and $\mathbf{C}$) that expedite the correction step of query mapping (Supplementary Methods). These elements are a subset of the components available once Harmony integration is applied to the reference cells. Note that other input embeddings, such as canonical correlation analysis (CCA), may be used in place of PCA as long as the gene loadings to perform query projection into those coordinates are saved.

Table 2 lists the Symphony minimal reference elements required to perform mapping. Table 2 also shows which additional components of a full Harmony reference are not included in the Symphony minimal reference elements. Importantly, the dimensions of the Symphony data structures do not require information on the $n$ individual reference cells and hence do not scale with the raw number of reference cells. Rather the components scale with the biological

complexity captured (i.e., number of clusters $k$ and dimensionality of embedding $d$). Conversely, the Harmony data structures store information on a per-cell basis ($n$). Note that in practice the integrated embedding of reference cells ($\hat{\mathbf{Z}}_r$) is saved to the reference because it is needed to perform downstream transfer of annotations from reference to query cells (e.g., k-NN), but it is not required during any computations of the mapping step.

Starting from reference cell gene expression, we first perform within-cell library size normalization (if not already done) and variable gene selection to obtain $\mathbf{G}_r$, scaling of the genes to have mean 0 and variance 1 (saving $\mu$ and $\sigma$ for each gene), and PCA to embed the reference cells in a low-dimensional space, saving the gene loadings ($\mathbf{U}$) (Implementation Details). Then, the PCA embedding ($\mathbf{Z}_r$) and batch design matrix ($\mathbf{X}_r$) are used as input to Harmony integration to harmonize over batch-dependent sources of variation. Given the resulting harmonized embedding ($\hat{\mathbf{Z}}_r$) and final soft assignment of reference cells to clusters ($\mathbf{R}_r$), the locations of the final reference cluster centroids $\mathbf{Y} \in \mathbb{R}^{d \times k}$ can be calculated as in Eq. (3) and saved.

$$\mathbf{Y} = \hat{\mathbf{Z}}_r \mathbf{R}_r^T \qquad (3)$$

Symphony then computes the reference compression terms $\mathbf{N}_r$ (intuitively, the number of cells per cluster) and $\mathbf{C}$, which does not have an intuitive explanation but can be directly computed as $\mathbf{C} = \mathbf{R}_r \hat{\mathbf{Z}}_r^T$. Refer to the Supplementary Methods for a complete mathematical derivation of the compression terms. Symphony reference building ultimately returns the minimal reference elements: $\mu$, $\sigma$, $\mathbf{U}$, $\mathbf{Y}_{cos}$, $\mathbf{N}_r$, and $\mathbf{C}$ (Supplementary Fig. 1a).

**Algorithm 1**. Build Symphony reference.

**function** BUILDREFERENCE($\mathbf{G}_r$, $\mathbf{X}_r$)

  $\mu, \sigma, \mathbf{G}_{rs} \leftarrow$ SCALE($\mathbf{G}_r$)

  $\mathbf{U}, \mathbf{Z}_r \leftarrow$ PCA($\mathbf{G}_{rs}$)

  $\hat{\mathbf{Z}}_r, \mathbf{R}_r \leftarrow$ HARMONIZE($\mathbf{Z}_r$, $\mathbf{X}_r$)

  $\mathbf{Y} \leftarrow \hat{\mathbf{Z}}_r \mathbf{R}_r^T$

  $\mathbf{Y}_{cos} \leftarrow \dfrac{\mathbf{Y}_{[\cdot,i]}}{\|\mathbf{Y}_{[\cdot,i]}\|_2}$     ▷ L2 normalize cluster centroids

  $\mathbf{N}_r \leftarrow$ rowSums($\mathbf{R}_r$)     ▷ First compression term

  $\mathbf{C} \leftarrow \mathbf{R}_r \hat{\mathbf{Z}}_r^T$     ▷ Second compression term

  $Ref \leftarrow \{\mu, \sigma, \mathbf{U}, \mathbf{Y}_{cos}, \mathbf{N}_r, \mathbf{C}\}$

  **return** $Ref$     ▷ Return minimal reference elements

**Query mapping**. The Symphony mapping algorithm localizes new query cells to their appropriate locations in the harmonized embedding without the need to run integration on the reference and query cells altogether. The joint embedding of reference and query cells can be used for downstream analyses, such as transferring cell type annotations from the reference cells to the query cells.

Symphony mapping takes as input the gene expression matrix for query cells ($\mathbf{G}_q$), query design matrix assigning query cells to batches ($\mathbf{X}_q$), and the precomputed minimal elements for a reference (Ref). It outputs a query object containing the locations of query cells in the integrated reference embedding ($\hat{\mathbf{Z}}_q$; Algorithm 2). Table 3 lists the components of the query object that is returned by Symphony.

The input to the query mapping procedure is a gene expression matrix ($\mathbf{G}_q$) and design matrix ($\mathbf{X}_q$) for query cells, and the output is the locations of the cells in the harmonized embedding ($\hat{\mathbf{Z}}_q$). At a high level, the mapping algorithm first projects the query cells into the original, pre-harmonized PC space as the reference cells using the reference gene loadings ($\mathbf{U}$) and assigns probabilistic cluster membership across the reference cluster centroid locations. Then, the query cells are modeled using the Symphony mixture model and corrected to their approximate locations in the integrated embedding by regressing out the query batch-dependent effects (Algorithm 2).

*Projection of query cells into pre-harmonized PC space.* Symphony projects the query cells into the same original PCs ($\mathbf{Z}_r$) as the reference. Symphony assumes that, given a much smaller query compared to the reference ($m \ll n$), the PCs will remain approximately stable with the addition of query cells. To project the query cells, we first subset the query expression data by the same variable genes used in reference building and scale the normalized expression of the genes by the same means ($\mu$) and standard deviations ($\sigma$) used to scale the reference cells. Let $\mathbf{G}_{qs}$ denote the query gene expression matrix scaled by the reference gene means and standard deviations. We can then use the reference gene loadings ($\mathbf{U}$) to project $\mathbf{G}_{qs}$ into reference PC space. In Eq. (4), $\mathbf{Z}_q \in \mathbb{R}^{d \times m}$ denotes the PC embedding for the query cells. Note that if an alternate starting embedding (e.g., CCA) is used instead

of PCA, the gene loadings must be saved to enable this query projection step.

$$\mathbf{Z}_q = \mathbf{U}^T \mathbf{G}_{qs} = \Sigma_q \mathbf{V}_q^T \qquad (4)$$

*Soft assignment across reference clusters.* Once the query cells are projected into PC space, we soft assign the cells to the reference clusters using the saved reference centroid locations ($\mathbf{Y}_{cos}$). Symphony assumes that the reference cluster centroid locations remain approximately stable with the addition of a much smaller query dataset since the query contains no novel cell types. Under these conditions, we use a previously published objective function for soft k-means clustering in Eq. (5), which includes a distance term and an entropy regularization term over $\mathbf{R}$ weighted by hyperparameter $s$. This is the same objective function as the clustering step of Harmony, except it does not include the diversity penalty term. In Harmony, the purpose of the diversity term is to penalize clusters that are only represented by one or a few datasets (suggesting they do not represent true cell types). In contrast, Symphony does not require the use of a diversity penalty because the reference centroids have already been established. Furthermore, the query cell types can comprise a subset of a larger set of reference cell types, and therefore not all clusters are necessarily expected to be represented in the query. We can solve for $\mathbf{R}_q$, the optimal probabilistic assignment for query cells across each of the $k$ reference clusters (see Query mapping implementation details).

$$\min_{\mathbf{R},\mathbf{Y}} \sum_{i,k} \mathbf{R}_{[k,i]} \|\mathbf{Z}_{[\cdot,i]} - \mathbf{Y}_{[\cdot,k]}\|^2 + s\mathbf{R}_{[k,i]} \log(\mathbf{R}_{[k,i]})$$
$$\text{s.t.} \ \forall_i \forall_k \mathbf{R}_{[k,i]} > 0, \ \forall_i \sum_k \mathbf{R}_{[k,i]} = 1 \qquad (5)$$

*Mixture of experts correction.* The final step in Symphony mapping is to model then remove the query batch effects to obtain $\hat{\mathbf{Z}}_q$, the approximate location of query cells in the harmonized reference embedding. In Eq. (2), we modeled the reference and query cells together and wish to solve for the query parameters $\mathbf{B}_{qk} \in \mathbb{R}^{(1+c) \times d}$ for each cluster $k$. The reference-dependent terms in Eq. (2) were previously computed and saved in compressed form ($\mathbf{N}_r$ and $\mathbf{C}$). With $\mathbf{R}_q$ and $\mathbf{Z}_q$ calculated from query cell projection and clustering, we can finally solve for $\mathbf{B}_{qk}$. Similar to the correction step of Harmony, we obtain cell-specific correction values for the query cells by removing the batch-dependent terms captured in $\mathbf{B}_{qk[1:c,\cdot]}$. Note that the reference batch terms are neither modeled nor corrected during reference mapping, so the harmonized reference cells do not move.

The final locations of the query cells in the harmonized embedding are estimated by iterating over all $k$ clusters and subtracting out the non-intercept batch terms for each cell weighted by cluster membership, as in Eq. (6). Intuitively, the query centroids are moved so that they overlap perfectly with the reference centroids in the harmonized embedding. The vector $\hat{\mathbf{Z}}_{q[i]}$ denotes the approximate location in harmonized PC space for query cell $i$. Note that mapping results may slightly differ based on whether one maps query cells all together (correcting for query batches) or maps each query batch separately. As all query cells play a role in parameter estimation if mapped altogether, the batches are technically not independent.

$$\mathbf{Z}_{q[i]} = \sum_k \mathbf{R}_{q[k,i]} [\mathbf{B}_{qk[0,\cdot]}^T + \mathbf{B}_{qk[1:c,i]}^T \mathbf{X}_q] + \varepsilon$$
$$\hat{\mathbf{Z}}_{q[i]} = \mathbf{Z}_{q[i]} - \sum_k \mathbf{R}_{q[k,i]} \mathbf{B}_{qk[1:c,\cdot]}^T \mathbf{X}_q \qquad (6)$$
$$\hat{\mathbf{Z}}_{q[i]} = \sum_k \mathbf{R}_{q[k,i]} \mathbf{B}_{qk[0,\cdot]}^T + \varepsilon$$

**Algorithm 2**. Map query cells onto reference

**function** QUERYMAPPING($\mathbf{G}_q$, $\mathbf{X}_q$, Ref)

  $\mathbf{G}_{qs} \leftarrow$ SCALE($\mathbf{G}_q$, Ref\$$\mu$, Ref\$$\sigma$)     ▷ \$ denotes accessing a component of Ref

  $\mathbf{Z}_q \leftarrow$ PCAPROJECTION($\mathbf{G}_{qs}$, Ref\$$\mathbf{U}$)

  $\mathbf{R}_q \leftarrow$ CLUSTER($\mathbf{Z}_q$, Ref\$$\mathbf{Y}_{cos}$)

  $\hat{\mathbf{Z}}_q \leftarrow \mathbf{Z}_q$

  **for** $k \leftarrow 1 \dots k$ **do**

    $\mathbf{E} \leftarrow \mathbf{X}_q^* \mathbf{R}_q^{(k)} \mathbf{X}_q^{*T}$     ▷ $\mathbf{X}_q^*$ denotes query design matrix augmented with row of 1s

    $\mathbf{E}_{[0,0]} \leftarrow \mathbf{E}_{[0,0]} +$ Ref\$$\mathbf{N}_{r[k]}$

    $\mathbf{F} \leftarrow \mathbf{X}_q^* \mathbf{R}_q^{(k)} \mathbf{Z}_q^T$

    $\mathbf{F}_{[0,\cdot]} \leftarrow \mathbf{F}_{[0,\cdot]} +$ Ref\$$\mathbf{C}_{[k,\cdot]}$

    $\mathbf{B}_{qk} \leftarrow (\mathbf{E} + \lambda \mathbf{I})^{-1}(\mathbf{F})$

    $\mathbf{B}_{qk[0,\cdot]} \leftarrow 0$     ▷ Do not correct the intercept terms

    $\hat{\mathbf{Z}}_q \leftarrow \hat{\mathbf{Z}}_q - \mathbf{B}_{qk}^T \mathbf{X}_q^* \mathbf{R}_q^{(k)}$

  **return** $\hat{\mathbf{Z}}_q$     ▷ Return query locations in hPC space

### Reference building implementation details

*Normalization.* Starting with the gene expression matrix for reference cells, we perform $\log(CP10K + 1)$ library size normalization of the cells (if not already

done). Log-normalization is recommended and performed by default (and used in all scRNA-seq analyses in the manuscript). However, Symphony can be used with other normalization methods, such as SCTransform[63] or TF-IDF (see scATAC-seq analysis). The only requirement is that reference and query datasets are normalized in the same manner.

*Variable gene selection and scaling.* We subset by the top $g$ variable genes by the variance stabilizing transform (VST) method (as provided in Seurat[18]), which fits a line to the log(variance) and log(mean) relationship using local polynomial regression, then standardizes the features by observed mean and expected variance, calculating gene variance on the standardized values, which is re-implemented as a standalone function at https://github.com/immunogenomics/singlecellmethods. The data is scaled such that the expression of each gene has a mean expression of 0 and variance of 1 across all cells.

*Principal component analysis (PCA).* We perform dimensionality reduction on the scaled gene expression $\mathbf{G_{rs}}$ using principal component analysis (PCA). PCA projects the data a low-dimensional, orthonormal embedding that retains most of the variation of gene expression in the dataset. Singular value decomposition (SVD) is a matrix factorization method that can calculate the PCs for a dataset. Here, we use SVD ("irlba" R package[64]) to perform PCA. SVD states that matrix $\mathbf{G_{rs}}$ with dimensions $g \times n$ can be factorized as:

$$\mathbf{G_{rs}} = \mathbf{U\Sigma V^T} \tag{7}$$

In Eq. (7), $\mathbf{\Sigma V^T} = \mathbf{Z_r}$ (dimensions $d \times n$) represents the embedding of reference cells in PC space, after truncating the matrix on the first $d$ (by default, $d = 20$) PCs. The gene loadings ($\mathbf{U} \in \mathbb{R}^{g \times d}$) are saved. Note that an alternative embedding, such as canonical correlation analysis (CCA) may be used in place of PCA, as long as the gene loadings are saved.

*Harmony integration.* The PCA embedding ($\mathbf{Z_r}$) is then input to Harmony for dataset integration. By default, Symphony uses the default parameters for the cluster diversity enforcement ($\theta = 2$), the entropy regularization hyperparameter for soft k-means ($s = 0.1$), and the number of clusters $k = \min\left(100, \frac{n}{30}\right)$. We save the L2-normalized cluster centroid locations $\mathbf{Y_{cos}}$ to the reference object since query mapping employs a cosine distance metric. If the reference has a single-level batch structure, no integration is performed, and the clusters are defined using soft k-means.

**Query mapping implementation details**

*Normalization and scaling.* The gene expression for query cells are assumed to be library size normalized in the same manner that was used to normalize the reference cells, e.g., $\log(CP10K + 1)$. During scaling, the query data is subset by the same variable genes from the reference datasets, and query gene expression is scaled by the *reference* gene means and standard deviations. Any genes present in the query but not the reference are ignored, and any genes present in the reference but not the query have scaled expression set to 0.

*Clustering step uses cosine distance.* As in Harmony, in practice we use cosine distance rather than Euclidean distance in the clustering step. For the computation of the distance term, we L2-normalize the columns (cells) of $\mathbf{Z}$ and columns (centroids) of $\mathbf{Y_k}$ such that the squared values sum to 1 across each column. Let the terms $\mathbf{Z_{q\_cos[\cdot,i]}}$ and $\mathbf{Y_{cos[\cdot,k]}}$ represent the L2-normalized locations of query cell $i$ and the reference centroid for cluster $k$ in PC space, respectively. We compute the cosine distance between the cells and centroids. Since all $\mathbf{Z_{q\_cos[\cdot,i]}}$ and $\mathbf{Y_{cos[\cdot,k]}}$ each have unity norm, the squared Euclidean distance $\|\mathbf{Z_{q\_cos[\cdot,i]}} - \mathbf{Y_{cos[\cdot,k]}}\|^2$ is equivalent to the cosine distance $2\left(1 - \cos(\mathbf{Y_{cos[\cdot,k]}}, \mathbf{Z_{q\_cos[\cdot,i]}})\right) = 2(1 - \mathbf{Y_{cos[k,\cdot]}^T} \mathbf{Z_{q\_cos[\cdot,i]}})$. Therefore, the objective function for query assignment to centroids becomes:

$$\min_{\mathbf{R,Y}} \sum_{i,k} 2\mathbf{R_{q[k,i]}}(1 - \mathbf{Y_{cos[k,\cdot]}^T} \mathbf{Z_{q\_cos[\cdot,i]}}) + s\mathbf{R_{q[k,i]}} \log(\mathbf{R_{q[k,i]}})$$
$$\text{s.t. } \forall_i \forall_k \mathbf{R_{[k,i]}} > 0, \ \forall_i \sum_k \mathbf{R_{[k,i]}} = 1 \tag{8}$$

We can solve the optimization problem using an expectation-maximization framework. Following the same strategy as Korsunsky et al.[17], we calculate $\mathbf{R_q}$, the optimal probabilistic assignments for each query cell $i$ across each of the $k$ reference clusters. In Eq. (9), we can interpret $\mathbf{R_{q[k,i]}}$ as the probability that query cell $i$ belongs to cluster $k$. The denominator term simply ensures that for any given cell $i$, the probabilities across all $k$ clusters sum to one. By default, $s = 0.1$.

$$\mathbf{R_{q(k,i)}} = \frac{\exp\left(-\frac{2}{s}(1 - \mathbf{Y_{cos[k,\bullet]}^T} \mathbf{Z_{q\_cos[\bullet,i]}})\right)}{\sum_k \exp\left(-\frac{2}{s}(1 - \mathbf{Y_{cos[k,\bullet]}^T} \mathbf{Z_{q\_cos[\bullet,i]}})\right)} \tag{9}$$

*Query label prediction and prediction confidence score.* Once query cells are embedded in the same low-dimensional feature space as the reference, reference labels can be transferred to the query using any downstream model (e.g., k-NN, SVM, logistic regression) using the harmonized PCs as input. See the analysis of

benchmarking against automatic cell type classifiers for examples of using different downstream inference methods with Symphony.

For most analyses presented, we use a simple and intuitive k-NN classifier (as implemented in the "class" R package), which uses majority vote with ties broken randomly. We provide a convenient wrapper function in the Symphony package to do this (knnPredict), which optionally returns the prediction confidence, measuring the proportion of reference neighbors contributing to the winning vote. For k-NN prediction, we would recommend that users alter the $k$ parameter so that it is ideally no larger than the number of cells in the rarest cell type of the reference. For example, if the reference contains only 10 cells of a rare cell type, then we recommend the user set $k$ no higher than 10, to ensure that rare cell types in the reference have the chance of being predicted given a majority vote k-NN classifier.

**Mapping confidence metrics.** Symphony offers two scores that measure the confidence in query mapping and helps to identify query cell states missing in the reference. We recommend that users try both metrics and further investigate any query cells/clusters that appear to map poorly.

*Background: Mahalanobis distance.* Mahalanobis distance is a multivariate metric that measures the distance from a point to a distribution. It can be thought of as analogous to the univariate $Z$-score. We use Mahalanobis distance rather than Euclidean distance since Euclidean distance assumes uncorrelated features, whereas Mahalanobis distance accounts for potentially correlated features. PCA technically returns uncorrelated variables (which would have a covariance matrix containing zeros in all non-diagonal positions); however, when considering the distribution of cells surrounding each soft-cluster individually (rather than all cells altogether), the covariance matrices have non-zero values. Mahalanobis distance ($D$) from a point $\mathbf{x}$ to a distribution with mean $\mathbf{\mu}$ and covariance matrix $\mathbf{\Sigma}$ in $d$-dimensional space is defined as:

$$D^2 = (\mathbf{x} - \mathbf{\mu})^T \mathbf{\Sigma}^{-1} (\mathbf{x} - \mathbf{\mu}) \tag{10}$$

*Per-cell mapping metric.* The per-cell mapping metric measures the weighted Mahalanobis distance between each query cell and the distribution of reference cells they map nearest to, weighted by soft-cluster membership. In Eq. (9), $\mathbf{x}$ is the query cell position ($d$-dimensional vector), and $\mathbf{\mu}$ and $\mathbf{\Sigma}$ are the weighted mean ($\mathbf{\mu}$) and covariance matrix ($\mathbf{\Sigma}$) for each reference Harmony soft-cluster centroid in pre-Harmonized PC space, weighted according to the reference cells belonging to that cluster ($\mathbf{R_r}$). For each query cell, we calculate its Mahalanobis distance ($D$) to each reference centroid then take the weighted average across all centroids the query cell belongs to (defined using $\mathbf{R_q}$). As the metric is a distance measure, it ranges from 0 to infinity. In practice, we have noticed that cell states well-represented in the reference tend to have values <10.

*Per-cluster mapping metric.* The per-cluster mapping metric takes in a user-defined set of query cluster labels, e.g., putative cell types from running a de novo PCA pipeline on the query followed by graph-based clustering. User-defined clusters are likely to represent unique cell types within the query data. The intuition behind this metric is that if a query cell type is well-represented by the reference PC structure, then it should map closely to a reference centroid. We first project the query into reference pre-Harmony PCs, then calculate the Mahalanobis distance between the query cluster and its nearest reference centroid, where the covariance is defined using the query cells in reference PC space. All cells in a given cluster receive the same score. By aggregating signal using multiple query cells per cluster rather than each cell individually, this metric potentially offers greater discriminatory ability than the per-cell metric. A disadvantage of the metric is that it is sometimes difficult to anticipate what the covariance of the query will be upon projection into reference PCs. Additionally, if a query cluster has very few cells, the estimation of its covariance matrix becomes numerically unstable; in practice, we return NAs for clusters smaller than $2d$, where $d$ is the dimensionality of the embedding.

**Analysis of 10x PBMCs**

*Preprocessing scRNA-seq data.* The three 10x PBMCs datasets were previously preprocessed by our group as part of the Harmony publication. We used the same $\log(CP10K + 1)$ normalized expression data, filtered as described in Korsunsky et al.[17]. The datasets consist of PBMCs sequenced using three technologies: 3'v1 ($n = 4809$ cells), 3'v2 (8380 cells), and 5' (7697 cells).

*Symphony mapping experiments.* To construct each of three references for subsequent mapping, we aggregated two reference datasets into a single normalized expression matrix and identified the top 1000 reference variable genes across each technology batch (then pooled them) using the variance stabilizing transformation (VST) procedure[18]. We ran Harmony on the top 20 PCs, harmonizing over "technology" with default parameters. For Symphony mapping, we specified query "technology" covariate.

*Constructing gold standard embedding.* To construct the gold standard de novo Harmony embedding, we concatenated all three datasets together into a single expression matrix, subset by the top 1000 variable genes calculated within each of

three batches in the dataset then pooled, and ran Harmony integration on the top 20 PCs, harmonizing over "technology" with default parameters.

*Assigning ground truth cell types.* We clustered the cells in the gold standard embedding using the Louvain algorithm as implemented in the single-cellmethods:::buildSNN_fromFeatures function and Seurat:::RunModularityClustering function[18]. For PBMCs, we used nn_k = 5 (to capture rare HSCs), nn_eps = 0.5, and resolution = 0.8. We labeled clusters with ground truth cell types according to expression of canonical lineage marker genes (Supplementary Table 2). PBMCs were assigned across 9 types: T (CD4: *CD3D, IL7R, CD4*; CD8: *CD3D, CD8A*), NK (*GNLY*), B (*MS4A1*), Monocytes (CD14: *CD14, LYZ*; CD16: *FCGR3A, MS4A7*), DCs (*FCER1A*), Megakaryocytes (*PPBP*), and HSCs (*CD59*). Clusters were labeled if the AUC (calculated using presto[65]) for the corresponding lineage marker was >0.7. For clusters that did not express specific lineage markers or were ambiguous between multiple lineages, we manually assigned a cell type based on the top differentially expressed genes (Supplementary Table 3) and comparing to the cluster annotation from Korsunsky et al. (2019). PBMCs cluster 14 was identified as low-quality cells (high in mitochondrial genes). We removed all cells in this cluster (n = 315) from further analyses, leaving 20,571 cells total. The final ground truth labels were used in downstream analyses and cell type classification accuracy evaluation.

*Evaluation of mixing and cell type classification accuracy.* To compare dataset mixing between de novo integration and mapping, we calculated Local Inverse Simpson Index (LISI) using the compute_lisi function from https://github.com/immunogenomics/LISI with default parameters (perplexity = 30). Perplexity represents the effective number of each cell's neighbors. For each mapping experiment, we calculated dataset LISI on all cells, then subset the results for query cell neighborhoods only to measure the effective number of datasets in the local neighborhood of each query cell.

We predicted query cell types by transferring reference cell type annotations using the knn function in the "class" R package (k = 5). We calculated overall accuracy across all query cells and cell type F1-scores (the harmonic mean of precision and recall, ranging from 0–1). Precision = TP/(TP+FP), recall = TP/(TP + FN), F1 = (2\*precision\*recall)/(precision+recall). Cell type F1 was the metric Abdelaal et al. used to benchmark automated cell type classifiers[35]. We used their provided evaluate.R script (https://github.com/tabdelaal/scRNAseq_Benchmark/blob/master/evaluate.R) to calculate confusion matrices and F1-scores by cell type.

*Quantifying local similarity between two embeddings.* k-NN-correlation (k-NN-corr) is a new metric that quantifies how well a given alternative embedding preserves the local neighborhood structure with respect to a gold standard embedding. Anchoring on each query cell, we calculate (1) the pairwise distances to its $k$ nearest reference neighbors in the gold standard embedding and (2) the distances between the same query-reference neighbor pairs in an alternate embedding (Methods), then calculate the Spearman (rank-based) correlation between (1) and (2). For each query cell, we obtain a single k-NN-corr value capturing how well the relative distances to its $k$ nearest reference neighbors are preserved. Note that k-NN-corr is asymmetric with respect to which embedding is selected as the gold standard and which is selected as the alternative because the nearest neighbor pairs are fixed based on how they were defined in the gold standard. The distribution of k-NN-corr scores for all query cells can measure the embedding quality, where higher k-NN-corr indicates greater recapitulation of the gold standard. Lower values for $k$ assess more local neighborhoods, whereas higher $k$ assesses more global structure.

We calculated k-NN-corr between the gold standard Harmony embedding and two alternative embeddings: (1) the full Symphony mapping algorithm (projection, clustering, and correction) and (2) PCA-projection only as a comparison to a batch-naïve mapping. PCA-projection refers to the first step of Symphony mapping, where query cells are projected from gene expression to pre-harmonized PC space: $\mathbf{Z_q} = \mathbf{U^T G_q}$.

**Benchmarking against automatic cell type classifiers**. We downloaded the PbmcBench benchmarking dataset used by a recent comparison of automatic cell type identification methods[35,39]. For each of 48 train-test experiments described in Abdelaal et al.[35] (see below for details), we used the same evaluation metrics (median cell type F1-score) to evaluate Symphony in comparison to the 22 other classifiers. We obtained the numerical F1-score results for the other classifiers for all 48 experiments directly from the authors in order to determine Symphony's place within the rank ordering of classifier performance.

During reference building, we explored two different gene selection methods: (1) unsupervised (top 2000 variable genes) and (2) supervised based on identifying the top 20 differentially expressed (DE) genes per cell type. Option (2) was included to give Symphony the same information as prior-knowledge classifiers (e.g., SCINA with 20 marker genes per cell type). We used the "presto" package[65] for DE analysis. No integration was performed because the reference had a single-level batch structure (clusters were simply assigned using soft k-means). Onto each of seven references (each representing one protocol for donor pbmc1), we mapped either a second protocol for donor pbmc1 (six experiments) or the same protocol for donor pbmc2 (six experiment). Given the resulting Symphony joint feature embeddings, we used three downstream classifiers to predict query cell types: 5-

NN, SVM with a radial kernel, and multinomial logistic regression with ridge ("glmnet" package in R)[66]. We note that other methods in the original benchmark were permitted to have a "rejection option" (leave uncertain cells as "unclassified" and not included in F1-score calculation). Hence, we also added a version for each of the two Symphony 5-NN versions that only assigned a label if the cell had >0.6 prediction confidence ( ≥ 4 out of 5 neighbors with the winning vote). A total of 8 Symphony-based classifiers were tested (2 gene selection methods \* 3 downstream classifiers + 2 rejection option versions).

**Pancreas benchmark**

*Constructing the pancreas query with mouse and human cells.* The pancreas query dataset (Baron et al.[46]; inDrop, n = 8569 human, 1886 mouse cells) along with author-defined cell type labels were downloaded from https://hemberg-lab.github.io/scRNA.seq.datasets/human/pancreas/. In order to combine the human and mouse matrices into a single aggregated query, we "humanized" the mouse expression matrix by mapping mouse genes to their orthologous human genes. This mapping was computed using the "biomaRt" R package[67], mapping "mgi_symbol" from the "mmusculus_gene_ensembl" database to "hgnc_symbol" from the "hsapien_gene_ensembl" database. We added additional ortholog pairs from HomoloGene (https://ftp.ncbi.nih.gov/pub/HomoloGene/build37.1/homologene.data) to obtain a total of 22,578 human to mouse gene ortholog pairs. We represented this map as a matrix, with mouse genes as rows, human genes as columns, and values in {0, 1} assigned to denote whether a mouse gene maps to a human gene. We then normalized the matrix to have each column sum to one, effectively creating a count-preserving probabilistic map from $h$ mouse to $H$ human genes $\mathbf{M} \in \mathbb{R}^{H \times h}$. Mapping from mouse to human genes is then performed with matrix multiplication: $\mathbf{E_{human}} = \mathbf{M E_{mouse}}$. Note that while the mouse gene expression matrix $\mathbf{E_{mouse}}$ contains only integers ($\mathbf{E_{mouse}} \in \mathbb{Z}^{h \times N}$), the many-to-many mapping means that the mapped human gene expression matrix $\mathbf{E_{human}}$ may contain non-integers ($\mathbf{E_{human}} \in \mathbb{R}^{H \times N}$). For any human orthologs that were missing in the mouse expression data, we filled in the expression with zeroes. We then log(CP10K + 1) normalized the query cells.

*Preprocessing reference scRNA-seq data.* The pancreas reference datasets were each sequenced with a different technology: Fluidigm C1 ($n = 638$ cells), CEL-seq (946 cells), CEL-seq2 (2238 cells), Smart-seq2 (2355 cells). We obtained the log(CP10K + 1) normalized data from the Harmony publication[17]. The pancreas cell type labels were obtained from Korsunsky et al.[17], which assigned cells across 9 types within each dataset individually according to cluster-specific expression of marker genes: alpha (*GCG*), beta (*MAFA*), gamma (*PPY*), delta (*SST*), acinar (*PRSS1*), ductal (*KRT19*), endothelial (*CDH5*), stellate (*COL1A2*), and immune (*PTPRC*). We removed 290 cells that were left unassigned as part of ambiguous or outlier clusters during within-dataset annotation, leaving 5887 reference cells.

We benchmarked three reference mapping methods as follows:

*Symphony mapping onto a Harmony reference.* We calculated the top 1000 variable genes within each of the four reference dataset separately using VST then pooled them (total 2,236 variable genes) for PCA. For reference integration, we ran Harmony on the top 20 PCs, harmonizing over "donor" (theta = 2) and "technology" (theta = 4), with tau = 5. For Symphony mapping, we specified query "donor", "species", and "technology" covariates.

As a comparison with de novo integration, we ran Harmony integration on all five datasets together. We pooled the top 1000 variable genes within each dataset (total 2650 genes), calculated the top 20 PCs, and harmonized over "species" (theta = 2), "donor" (theta = 2), and "technology" (theta = 2), with tau = 5.

*Seurat mapping onto a Seurat reference.* We ran Seurat version 4[30] (Seurat_4.0.2) and followed the steps from the author's tutorial (https://satijalab.org/seurat/v3.2/integration.html) to integrate the reference datasets, given that the FindIntegrationAnchors and IntegrateData functions for de novo integration are equivalent between Seurat v3 and v4. We used the same 2,236 variable genes as above and 20 PCs. We followed the tutorial (https://satijalab.org/seurat/v4.0/reference_mapping.html) to map each donor dataset from the query individually. We used the FindTransferAnchors function with reduction = "pcaproject" and MapQuery function with reference.reduction = "pca" (as the documentation recommends for unimodal analysis).

As a comparison with de novo integration, we ran Seurat integration (FindIntegrationAnchors and IntegrateData) on all five datasets (integrating over plate-based technologies and Baron donors as batches) with the same 2650 variable genes as above and 20 PCs.

*scArches mapping onto a trVAE reference.* We ran scArches[28] version 0.3 with trVAE[33] using default parameters provided in the authors' notebooks (https://github.com/theislab/scarches/tree/master/notebooks). For the pancreas analysis, we only had access to normalized expression data and therefore ran scArches with trVAE using the "mse" reconstruction loss function. We included query batch information in the condition_key parameter.

As a comparison with de novo integration, we ran trVAE on all 5 datasets with default parameters, specifying batch as "dataset" for the 4 plate-based datasets and "donor" for the Baron et al. dataset.

*Evaluation metrics.* We used the resulting joint (reference and query) cell embedding to predict query cell types from reference cells using a 5-NN classifier and calculated cell type prediction F1-scores, as described above. For Seurat, we additionally compared another set of predicted labels using Seurat's TransferData function. Note that for the cell type prediction and cell type F1-score calculation, we excluded query Schwann cells from the accuracy metrics because that cell type is not present in the reference.

To assess degree of mixing, we calculated ref_query LISI and query_donors LISI on query cell neighborhoods using the *compute_lisi* function as above. ref_query LISI measures how well the reference and query datasets are mixed (max ref_query LISI of 2), whereas query_donors LISI measures how well the individual donors within the query dataset are mixed (max of 6).

To assess how well the query low-dimensional structure is preserved in the mapped embedding, we developed a new metric called within-query k-NN-correlation (wiq-kNN-corr). For each query batch (here, donor), we run a standard PCA pipeline on the cells (using 20 dimensions and selecting 2000 variable genes per batch using VST). Then, anchoring on each query cell, we calculate it's (1) distances to the $k$ nearest neighbors in the query PCA embedding and (2) the distances to those same $k$ cells after reference mapping. Then, wiq-kNN-corr is the Spearman correlation between (1) and (2), ranging between –1 and 1, where higher values represent better retention of the sorted original neighbor ordering. The calculation is similar to k-NN correlation described above, except instead of measuring the sorted ordering of *reference* neighbors in a de novo integration embedding, we measure the sorted ordering of *query* neighbors in the query PCA embedding.

**Fetal liver hematopoiesis trajectory inference example.** We obtained post-filtered, post-doublet removal data directly from the authors[47] along with author-defined cell type annotations for 113,063 cells (14 donors) sequenced with 10x 3' end bias (reference) and a separate 25,367 cells sequenced with 10x 5' end bias (query). For building the harmonized reference from all reference (3') cells, we followed the same variable gene selection procedures as the original authors, using the Seurat variance/mean ratio (VMR) method with parameters min_expr = 0.0125, max_expr = 3, and min_dispersion = 0.625 (resulting in 1917 variable genes). We integrated the reference with Harmony over "donor" (theta = 3). To map query (5') cells against the reference, we removed two donors (F2 and F5, $n = 3953$) from the query based on low library complexity (Supplementary Fig. 5b), leaving $n = 21,414$ cells from 5 donors. During mapping, we specified both "donor" and "technology" as covariates. We predicted query cell types by transferring reference cell type annotations using the knn function in the "class" R package ($k = 30$). We visualized the confusion matrix for the query (5')-to-reference (3') experiment using the "ComplexHeatmap" R package[68].

For the trajectory inference analysis, we obtained trajectory coordinates from the force directed graph (FDG) embedding of all reference cells from the original authors[47], forming a reference trajectory. We restricted the trajectory to immune cell types only (excluding hepatocytes, fibroblasts, and endothelial). We then mapped a subset of the query cells belonging to the MEM lineage (MEMPs, megakaryocytes, mast cells, early-late erythroid; $n = 5141$) to the reference-defined trajectory by averaging the FDG coordinates of the 10 reference immune cell neighbors in the Symphony embedding.

**Evaluating performance of Symphony mapping metrics**

*Simulating missing cell type scenarios in fetal liver hematopoiesis dataset.* Using the 3' fetal liver dataset described above ($n = 113,063$ cells), we held out one random donor (F8, 16,945 cells) as the query and used the remaining 13 donors as the reference dataset. We constructed 3 increasingly difficult scenarios where the reference is missing cell types present in the query by artificially removing cell types from the reference:

1. Removing all non-immune cell types: endothelial cells ($n = 321$ cells), fibroblasts ($n = 361$ cells), hepatocytes ($n = 306$ cells).
2. Removing all myeloid cells: Kupffer cells ($n = 6022$ cells), Mono-Mac ($n = 1035$), Monocyte ($n = 375$), Monocyte precursor ($n = 44$), DC1 ($n = 56$), DC2 ($n = 292$), VCAM1+ EI Macro. ($n = 52$), Neut-myeloid prog. ($n = 91$), DC precursor ($n = 14$), pDC precursor ($n = 9$)
3. Removing Kupffer cells ($n = 6022$ cells)

For each of the three scenarios, we built a Symphony reference using the same variable gene selection and reference building parameters as in the fetal liver hematopoiesis example, then mapped the query containing all 27 cell types onto the reference. We calculated both per-cell mapping and per-cluster mapping metrics for the query cells. To plot ROC curves and calculate AUC values for each metric, we used the "pROC" R package (roc and auc functions), using a binary label of missing vs. present in the reference as the ground truth for prediction. Note that for the per-cluster metric, the pDC precursor ($n = 9$ cells), DC precursor ($n = 14$), and Pre-pro B cell ($n = 12$) clusters were too small to calculate a per-cluster score and were assigned a value of 0 for the per-cluster metric in all scenarios (unable to be flagged as novel) for inclusion in AUC calculations.

*Comparison of Symphony mapping metrics to Seurat mapping score.* Using the 10x PBMCs dataset described above, we designated the 3'v2 and 5' data as the reference

and held out the 3'v1 data as a query. For each of the major cell types (B, DC, HSC, MK, Mono, NK, or T), we artificially removed all reference cells of that type and built a Symphony reference with that type missing (total 7 references/scenarios). We used the same reference building parameters as the original 10x PBMCs analysis. We then mapped all query cells onto each reference, simulating seven scenarios where the query contains a different novel unseen population, and calculated Symphony per-cell and per-cluster mapping metrics for the query cells in each scenario.

For each scenario, we also built a reference using Seurat (v4.0.2), integrating the reference dataset with FindIntegrationAnchors with 20 dimensions and mapping the query with FindTransferAnchors and MapQuery. We calculated query mapping scores with the MappingScore function. The Seurat mapping score is based on projecting the query into the reference space, then projecting back into the query and finding cells whose local neighborhoods are most altered by the transformation (see https://rdrr.io/cran/Seurat/man/MappingScore.html).

We generated ROC curves and calculated AUCs across all query cells, using a binary label of missing vs. present in the reference as the ground truth for prediction. For the Symphony per-cell metric and Seurat mapping score, each query cell was assigned its own value for the mapping metric, whereas for the Symphony per-cluster metric, all cells from the same cluster were assigned the same value. The HSC cluster ($n = 21$ cells) was too small to calculate a per-cluster score and all HSCs were assigned a value of 0 for the per-cluster metric in all scenarios (unable to be flagged as novel) for inclusion in AUC calculations. We calculated AUCs for each metric in two ways: (1) considering each scenario separately (threshold values independent across scenarios) and (2) aggregating cells across all 7 scenarios together into a single AUC calculation.

**Mapping tumor cells against healthy reference.** We mapped a renal cell carcinoma dataset onto a reference of healthy fetal kidney cells (datasets in Supplementary Table 1).

**Building the healthy kidney reference.** We found that the reference dataset gene names were previously assigned by the original authors using Gencode v24, whereas the query dataset gene names were assigned by the original authors using Gencode v30 liftover37 (query dataset.gtf file was provided by Bi et al.). Gencode.gtf files for versions 4-38 were downloaded from: http://ftp.ebi.ac.uk/pub/databases/gencode/Gencode_human. For many genes, the names were mismatched between the reference and query Gencode versions (different synonyms for the same gene). Therefore, to sync the two datasets, we used the Ensembl IDs of the reference genes to convert them to Gencode v30 gene names. We used the top 2000 variable genes across all cells to build the reference with 15 PCs, integrating over "Experiment" with theta = 0.5. Note that this reference building procedure is different from the original study[48], which did not use Harmony. For improved readability, we collapsed cell type labels for immune and stromal cells (e.g., "Proliferating monocyte" and "Monocyte" were collapsed into "Monocyte").

**Mapping the renal cell carcinoma dataset.** We mapped the query dataset ($n = 34,326$ cells from eight donors) starting from expression using default Symphony parameters, correcting for query "donor_id". As some gene names remained discordant between reference and query datasets, the mapping was based on the 1723 (out of 2000) reference variable genes shared. We used 10-NN to transfer reference cell type labels to the query. We calculated the per-cell mapping confidence (excluding the "Misc/Undetermined" cells, $n = 278$).

**Extending Symphony to scATAC-seq.** scATAC-seq is different from scRNA-seq in that open chromatin peaks are typically defined in a dataset-specific manner (i.e., rather than a pre-specified list of genes that apply to all datasets). Hence, this proof-of-concept analysis was run on peaks called on all cells as defined by the benchmarking paper by Chen et al.[52], obtained from the Pinello Lab GitHub: https://github.com/pinellolab/scATAC-benchmarking/blob/master/Real_Data/Buenrostro_2018/input/combined.sorted.merged.bed. In this dataset, peaks were called on each cell type aggregated separately then merged. The full peaks by cells matrix was calculated using chromVAR's getCounts function as demonstrated in their notebook (https://github.com/pinellolab/scATAC-benchmarking/blob/master/Real_Data/Buenrostro_2018/run_methods/chromVAR/chromVAR_buenrostro2018_kmers.ipynb), and subsequently binarized. The cell type information was also gathered from the Pinello Lab GitHub (https://github.com/pinellolab/scATAC-benchmarking/blob/master/Real_Data/Buenrostro_2018/input/metadata.tsv) while the donor information was inferred from the cell name.

We defined the query cells ($n = 298$) as those that belong to donor BM1214 while the remaining cells ($n = 1736$) were assigned as reference. BM1214 had cells corresponding to CMPs, GMPs, and pDCs, whose cell types all had cells from other donors also in the reference set. Since scATAC-seq is sparse and zero-inflated, the mean-scaling approach used for genes was changed to TF-IDF normalization on the binarized peaks by cells matrices. Seurat's TF-IDF function was modified to allow for an IDF vector as input and outputted the TF matrix, IDF vector, and normalized peaks by cells matrix. Following the TF-IDF implementation in Stuart & Butler et al.[18], we computed log(TFxIDF). The inverse-document frequency

(IDF) vector was calculated on the reference cells only and then used in the query cell normalization to get all the cells in the same space before mapping. With only this change to the Symphony methods, scATAC-seq query cells were mapped to a comparable reference. Feature selection, SVD, and Harmony were done as in the 10x PBMC analysis. Predicted query cell types were calculated using 5-NN. For plotting, we used the same cell type colors primarily defined from the Supplemental Data Table 1 of the original Buenrostro et al. (2018) paper with GMPs changed to a darker orange to better distinguish them visually from the CMPs and the "unknown" cells changed from gray to black, allowing gray to be used as the null color to better emphasize the other cell type colors.

**Memory T-cell surface protein inference example**. We used a memory T-cell CITE-seq dataset collected from a tuberculosis disease progression cohort of 259 individuals of admixed Peruvian ancestry[40]. The dataset includes expression of the whole-transcriptome (33,538 genes) and 30 surface protein markers from 500,089 memory T cells isolated from PBMCs. Including technical replicates, 271 samples were processed across 46 batches.

To assess protein prediction accuracy using Symphony embeddings, we randomly selected 217 samples (411,004 cells), normalized the expression of each gene (log2(CP10K+1)) and built a Symphony reference based on mRNA expression, correcting for donor and batch. The held-out 54 samples ($n = 89,085$ cells) comprised the query that we mapped onto the reference. We predicted the expression of each of the 30 surface proteins in each of the query cells by averaging the protein's expression across the cell's 50 nearest reference neighbors. Nearest neighbors were defined based on Euclidean distance in the batch-corrected low-dimensional embedding. As a ground truth for each protein in each query cell, we computed a smoothed estimate of the cells' measured protein expression by averaging the protein's expression across the cell's 50 nearest neighbors in the batch-corrected complete PCA embedding of all 259 donors. We did not use the cells' raw measured protein expression due to dropout. We computed the Pearson correlation coefficient between our predicted expression and the ground truth expression across all cells per donor for each marker.

To assess protein prediction accuracy based on mapping to a joint mRNA and protein-based Symphony reference, we first built an integrated reference by using canonical correlation analysis (CCA) to project cells into a low-dimensional embedding maximizing correlation between mRNA and protein features. We randomly selected 217 samples (395,373 cells) to comprise this reference, and normalized the expression of each gene (log2(CP10K + 1)), selected the top 2865 most variable genes, and scaled (mean = 0, variance = 1) all mRNA and protein features. We computed 20 canonical variates (CVs) with the cc function in the "CCA" R package[69] and corrected the mRNA CVs for donor and batch effects with Harmony. Then, we used Symphony to construct a reference based on the batch-corrected CVs, gene loadings on each CV, and mean and standard deviation used to scale each gene prior to CCA. The held-out 54 samples ($n = 104,716$ cells) comprised the query that we mapped onto the reference. As described above, we predicted the expression of each of the 30 surface proteins in each of the query cells based on the cell's 5, 10, or 50 nearest neighbors in the reference, estimated the smoothed ground truth expression of each protein in each query cell (now based on the batch-corrected CCA embedding of all 259 donors) and computed the per-donor Pearson correlation coefficient for each protein marker.

**Visualization**. For visualizing the embeddings using UMAP[70] (and included as the default in Symphony), we used the "uwot" R package with the following parameters: n_neighbors = 30, learning_rate = 0.5, init = "laplacian", metric = "cosine", and min_dist = 0.1 or 0.3. For each Symphony reference, we saved the uwot model at the time of UMAP using the uwot::save_uwot function and saved the path to the model file as part of the Symphony reference object. Saving the reference UMAP model allows for the fast projection of new query cells into reference UMAP space from the query embedding from Symphony mapping using the uwot::transform function.

For the pancreas benchmarking, we computed a de novo UMAP embedding on the joint reference and query embedding because a UMAP projection can potentially obscure differences between the projected data and dataset used to construct the UMAP model. For general purposes, we recommend UMAP projection when the reference cell UMAP coordinates are desired to remain stable.

To distinguish the reference plots from query plots, we visually present the reference embedding as a contour density instead of individual cells. The density plots were generated using "ggplot2" function stat_density_2d with geom = "polygon" and contour_var = "ndensity". We provide a custom function to generate these plots as part of the Symphony package (plotReference function).

**Runtime scalability analysis**. We downsampled a large memory T-cell dataset[40] to create benchmark reference datasets with 20,000, 50,000, 100,000, 250,000, and 500,000 cells. For each, we built a Symphony reference (20 PCs, 100 centroids) integrating over "donor" and mapped three different-sized queries: 1000, 10,000, and 100,000 cells. To isolate the separate effects of number of query cells and number of query batches on mapping time, we mapped against the 50,000-cell reference: (1) varying the number of query cells (from 1000 to 10,000 cells) while

keeping the number of donors constant and (2) varying the number of query donors (6 to 120 donors) while keeping the number of cells constant (randomly sampling 10,000 cells). We also performed separate experiments varying the number of reference centroids (25 to 400) and number of dimensions (10 to 320 PCs) while keeping all other parameters constant. We ran all jobs on Linux servers allotted 4 cores and 64 GB of memory (Intel Xeon E5-2690 v.3 processors) and used the system.time R function to measure elapsed time.

To compare runtime against Seurat and scArches, we used the same different-sized benchmark datasets and ran reference building and mapping or the corresponding de novo integration method (anchor-based integration for Seurat or trVAE for scArches). All jobs were allocated a maximum of 120 GB of memory and 24 h of runtime (and automatically terminated if memory or runtime were exceeded). We measured reference building and mapping runtime and corresponding de novo integration runtime for each method as elapsed time starting from gene expression. All jobs were run on a Linux server: Symphony and Seurat were allotted 4 CPU cores each, whereas scArches/trVAE was allotted 48 CPU cores to speed up runtime as it is a neural-net-based method.

**Constructing and mapping to multimillion cell atlas**. We obtained the AnnData file for the dataset (GSE158055_covid19.h5ad) from https://drive.google.com/file/d/1TXDJqOvFkJxbcm2u2-_bM5RBdTOqv56w/view. The link to the AnnData object was obtained from the following GitHub issue (response from user saketkc): https://github.com/satijalab/seurat/issues/4030. Owing to a limitation on the 32-bit sparse matrices in R (the maximum number of non-zero values in a sparse matrix currently cannot exceed $>2^{31} - 1$ for the "Matrix" R package), the gene expression matrix (1,462,702 cells by 27,943 genes) was preprocessed using the Python "scanpy" package. We log(CP10k + 1) normalized the data and subset to 1301 variable genes (list of variable genes was obtained from contacting the original authors). The remainder of the analysis was performed in R (v4.0). We held out a random 5% of samples (14 samples, 72,781 cells) as the query and built a Symphony reference using the other 95% of samples (270 samples, 1,389,921 cells), integrating over "Sample.name" and "dataset" with theta = 2.5 and 1.5, respectively, following the original publication. Reference building and mapping procedures were run on a Linux cluster with 4 cores and timed using the system.time function in R. UMAP steps were excluded from runtime as these are not inherent to the Symphony algorithm.

**Building Tabula Muris Senis (FACS) Symphony reference**. As a comprehensive mouse atlas, we built a Symphony reference using the Tabula Muris Senis dataset (not directly featured in the paper but included as a pre-built reference on Zenodo). We downloaded the FACS.h5ad file provided by the original authors on Figshare (see Supplementary Table 1). We extracted the expression matrix (counts), metadata, and highly variable genes using Python then read the data into R. For reference building, we used log(CP10k + 1) normalization, subset by the same variable genes as the original authors, then ran PCA and Harmony with 50 dimensions, nclust = 300, integrating over "mouse.id" with theta = 2.

**Reporting summary**. Further information on research design is available in the Nature Research Reporting Summary linked to this article.

# Data availability

Datasets for all analyses were obtained from publicly available sources, for which the specific links are listed in Supplementary Table 1. Additionally, we provide a compendium of 8 pre-built Symphony references available for download on Zenodo (see Table 1 for links). The 10x PBMCs data matrices were obtained from Korsunsky et al.[17]: https://github.com/immunogenomics/harmony2019/tree/master/data/figure4; original files from 10x Genomics: https://support.10xgenomics.com/single-cell-gene-expression/datasets. The pancreas reference data matrices were obtained from Korsunsky et al.[17]: https://github.com/immunogenomics/harmony2019/tree/master/data/figure5; original data is located on GEO (GSE81076[44], GSE85241[45], GSE86469[43]) and EMBL-EBI (E-MTAB-5061[42]). The human and mouse pancreas query data (Baron et al., 2016)[46] was downloaded from https://hemberg-lab.github.io/scRNA.seq.datasets/human/pancreas. The fetal liver hematopoiesis data from Popescu et al. (2019) is located on EMBL-EBI (E-MTAB-7407[47]), and post-doublet removal data was kindly provided by the authors. The PbmcBench data were obtained from the Zenodo repository for Abdelaal et al.[35]: https://zenodo.org/record/3357167#.YSL8p9NKhTY. The memory T-cell CITE-seq dataset from Nathan et al.[40] is available on GEO (GSE158769). The healthy fetal kidney data (Stewart et al., 2019)[48] was obtained from https://www.kidneycellatlas.org/. The renal cell carcinoma data (Bi et al.)[49] was obtained from the Broad Institute Single-Cell Portal (SCP1288). The 1.46 million cell COVID-19 dataset (Ren et al.)[41] is available on GEO (GSE158055), and.h5ad file was obtained from https://drive.google.com/file/d/1TXDJqOvFkJxbcm2u2-_bM5RBdTOqv56w/view. The scATAC-seq hematopoiesis dataset (Buenrostro et al.)[51] was downloaded from the Pinello Lab GitHub: https://github.com/pinellolab/scATAC-benchmarking/blob/master/Real_Data/Buenrostro_2018/input/combined.sorted.merged.bed. Gencode.gtf files for versions 4-38 (used for determining gene name synonyms in cancer analysis) were downloaded from: http://ftp.ebi.ac.uk/pub/databases/gencode/Gencode_human.

## Code availability

We provide an efficient implementation of Symphony at https://github.com/immunogenomics/symphony along with documentation, tutorials, and pre-built references. Symphony is also available for download as an R package on CRAN: https://cran.r-project.org/web/packages/symphony/index.html. The version of Symphony code used for the study is CRAN version 0.1.0. Jupyter notebooks and scripts to reproduce figures for the analyses in the manuscript are available at https://github.com/immunogenomics/symphony_reproducibility.

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

## Acknowledgements

We thank members of the Raychaudhuri Lab, in particular Yang Luo, for helpful feedback, comments, and fruitful discussion. We thank members of the Tuberculosis Research Unit (TBRU) LIMAA and Socios En Salud, in particular Megan Murray, Jessica Beynor, Yuriy Baglaenko, Sara Suliman, Ildiko van Rhijn, and Leonid Lecca, for their contributions to generating the memory T-cell dataset. We would also like to thank Issac Goh, Muzlifah Haniffa, and other members of the Haniffa Lab for graciously providing preprocessed datasets from their fetal liver hematopoiesis study. Additionally, we would like thank Ahmed Mahfouz, Kevin Bi and colleagues of the Van Allen Lab, Wenhong Hou, and Zemin Zhang for their kind assistance regarding their published datasets. We thank Jason Ku Wang for technical assistance. This work is supported in part by funding from the National Institutes of Health (1UH2AR067677, U19 AI111224, U01 HG009379, 1R01AR073833, and R01AR063759). The project described was supported by award Numbers T32GM007753 from the National Institute of General Medical Sciences (J.B.K.), T32AR007530 from the National Institute of Arthritis and Musculoskeletal and Skin Diseases (A.N.), and 5T32HG002295-17 from the National Human Genome Research Institute (K.W.). The content is solely the responsibility of the authors and does not necessarily represent the official views of the National Institutes of Health.

## Author contributions

J.B.K., I.K., and S.R. conceived the project. J.B.K. and I.K. designed and implemented the method under the guidance of S.R. J.B.K., I.K., K.W., F.Z., and A.N. contributed to the analysis of real datasets. S.R., N.M., and L.R. helped interpret data and analyses. S.R., A.N., and D.B.M. contributed to generating the memory T-cell CITE-seq dataset. I.K. and S.R. jointly supervised the work. J.B.K., I.K., and S.R. composed the initial manuscript draft. All authors provided critical intellectual feedback and participated in interpreting the data and revising the manuscript.

## Competing interests

S.R. receives research support from Biogen. I.K. does bioinformatics consulting for Brilyant Inc. No other authors have competing interests.
