## [Peer Review File · Nature Communications]

Efficient and precise single-cell reference atlas mapping with SymphonyREVIEWER COMMENTS

Reviewer #1 (Expertise: multimodal single-cell data analysis):

Joyce Kang et al. introduce Symphony, a method for constructing scRNA-seq reference datasets and rapidly mapping query datasets to these references. The method builds on the earlier Harmony method developed by the same group. As large, high-quality scRNA-seq reference datasets are now able to be assembled, and their proper use can greatly improve the analysis of new query datasets, there is an urgent need for such “reference mapping” methods in the community and I am quite excited by the Symphony method developed here. However, I have several comments about the manuscript in its current form, which I hope can help to further improve the work.

The authors should explore what would happen if the query contains cell states that are not present in the reference. Although the authors state this is a condition for query mapping (first condition for Harmony de novo equivalence), users may not always know whether their dataset contains a new cell state before mapping, and so it is worth exploring what results are expected in this case and how users may identify the presence of novel states in a reference-mapping workflow. For example, Seurat generates a mapping score and a prediction score for each query cell, which can be used to assess the mapping confidence. Can Symphony provide any metric that could help identify poorly-mapped query cells?

Can other normalisation methods be incorporated into the Symphony reference-construction and mapping workflow? log-normalisation may not be the best normalisation method for scRNA-seq data, and more recent methods like GLM-PCA or SCTransform could give better results.

Can Symphony be extended to allow the construction and mapping of other data modalities, particularly scATAC-seq? Since the method works in low-dimensional space, I’d imagine this is possible with scATAC-seq data processed using LSI (which uses the SVD), although some aspects would need to be modified (storing reference mean and SD not required, you could perhaps store the inverse document frequency for each peak in the reference instead?).

Under the section “Symphony maps against a large reference within seconds”, the authors claim to have shown that Symphony scales efficiently to map against multimillion-cell references. However, the authors did not actually demonstrate construction of a multimillion-cell reference, and there are other limitations that may become relevant as the datasets become much larger (the use of 32-bit sparse matrices, for example). To support their claim, the authors should actually demonstrate the construction of such a reference, and the mapping of queries to it.

In Figure 4, the immune cells in the Symphony reference seem to be split into two separate clusters of cells. Are these cells truly distinct (different immune cell types) or is this an artefact of the reference building? This also appears to be the case, although less extreme, for the beta and alpha cells.

Figure 4e: as well as reference-query mixing, it's important to measure how well preserved the original query low-dimensional structure is in the mapped embedding. This could be done per-batch in the query, using kNN-corr or a similar metric.

My understanding is that CCA captures shared sources of variation across two matrices. Since the authors use CCA for defining a joint embedding for the CITE-seq reference, would this embedding be biased to only capture sources of variation that were present in both the RNA and protein assays? What would happen in the case where one modality captures variation that is not shared in the second modality (for example, protein separates CD4 and CD8 T cells whereas this separation is very difficult to detect in the RNA modality)? The authors should also compare with the multimodal reference construction method in Seurat v4 (WNN followed by supervised PCA).

The explanation of how cell type labels are transferred from reference to query is unclear. Is it a simple majority vote using the label of the 5 nearest neighbours, or is the distance to each neighbour also considered? Also, how sensitive are the label predictions to the choice of k, and when should users alter the k parameter?

Minor comments and suggestions:

The overlapping density plots shown in Figure 2C at first glance appear to show the Harmony methods with a density peak at 1, but these in fact are from the PCA plots below. An alternative visualisation could be used that would avoid this problem (violin plot or boxplot for example).

The Seurat functions BuildSNN and RunModularityClustering aren't part of v3/v4. They were replaced by FindNeighbors and FindClusters. Which functions and Seurat version were used for clustering?

The GitHub repository containing the code to reproduce the analysis is not accessible, so I was unable to review the code used.

The authors should make the R package available on CRAN on Bioconductor.

Reviewer #2 (Expertise: Methods for the integration of single-cell data):

In this paper by Kang et al, entitled "Efficient and precise single-cell reference atlas mapping with Symphony", the authors present an algorithm to build integrated atlases and rapidly mapping query datasets. This mapping is much faster than performing de novo integration of the reference and the query and yields similar results. Moreover, it performs batch correction simultaneously to mapping, if necessary. As such, the reference atlas is frozen and not influenced by the query dataset. These are all useful and desirable functionalities in scRNAseq data analysis. The authors demonstrate Symphony capabilities on several datasets with complex experimental designs and compare Symphony performances with Seurat and scArches. The manuscript is well written, the methods section is accurate, and the description of the analyses is in general sound and convincing. I also congratulate the authors for the optimal implementation of the github page, with clear and documented tutorials, a rarity when reviewing yet unpublished tools.

In my opinion, the main limitation of this tool is the first condition that must be met for its use: the fact that all cell states in the query data set are captured by the reference dataset. Although "reasonable", it is not easy to satisfy this requirement. Often the user does not know a priori the composition of its dataset. In fact, the entire operation of mapping it to an atlas is performed to answer this very question. A dataset, even if obtained through cell sorting, could contain contaminant cells or unknown populations. When comparing organisms, as performed in the manuscript for pancreas populations in human and mouse, the comparability of cell populations is also an issue. Seurat, for example, assigns cells two different scores: mapping score and prediction score. The first reflect confidence that the cell is well represented in the reference, the second reflect confidence in the associated annotation. Is it possible to provide a similar mapping score for Symphony? E.g. in the Pancreas dataset analysis, Schwann cells are present in the query but not in the reference. These cells are mapped (to the most transcriptionally similar cell types I assume) but are not considered in accuracy estimation. Is there any way to assign a score that flag these cells as not represented in the reference? For example, are they far away from most centroids? A mapping score is important because a user could mislabel cells and misuse them without realizing it.

Note that this could also be used on purpose to force the positioning of cells on a reference or on a trajectory. For example, the authors in lines 354/356 discuss about healthy and diseased samples. It would then be interesting to see what happens if we map tumor cells to an atlas that contain the same normal tissue. Can we discriminate cancer cells with stem or differentiated features?

Finally, is it possible to provide a "prediction" score for the 5-NN classifier, since this classifier is implemented in the symphony package?

Minor comments:

- Is the mapping of each query batch independent from the rest of the query? i.e. if I map once a query composed of multiple batches or if I perform several mappings, one for each batch, do I get the same results? This would also be an advantage over performing de novo integration or using other tools since the inferred label would not change.
- Seurat is sometimes referred to as Seurat, Seurat v4, Seurat 3, Seurat 3 / 4. Also, at line 79 authors say that Seurat v4 is “compatible” with Seurat integration. This is a bit confusing (and not clear). Seurat 3 and 4 adopts the same exact anchor-based integration strategy, both for de novo integration and label transfer. The only difference (for what concerns the topic of this work) is that Seurat v4 introduces the mapQuery function that allows query projection onto the reference UMAP structure. Therefore the authors can simply refer to “Seurat” and specify the used version only in the method section.
- In figure S1C Zr_corr should be replaced with Zr as written in table 2.
- The calculation of LISI is not clear; from what I understood, the value should range between the minimum and maximum number of categories. Why then in figure 2c values seem to go below 1 and above 3? How many neighbours are used for LISI calculation?
- Line 157: the use of “similarity” here is not intuitive. Authors should say that it is an elaboration of distance (see line 156). Moreover, the checkmark and x mark in figure S2 are misleading since they evoke “correct” and “wrong” but instead it is a matter of good and bad mapping.
- What are the dashed lines in figures 2d and S2f?
- Figure 2 refers to “harmony” embedding whereas the text talks mainly about gold standard embedding. I would uniformise for clarity.
- The 5-NN classifier is not described in the methods. How does it work? Sometimes the authors use different numbers of neighbors. How can a user tune this number?
- Schwann cells are shown in figure S4a but not in figure 4b. how are they classified after symphony mapping? De novo integration appears to locate them close to stellate cells.
- When comparing Symphony with Seurat in the Pancreas dataset, it is not clear if the authors used the labels predicted by the Seurat TransferData function.
- Line 243-248. Does this refer to figure S5C and S5D? If yes, please insert ref to the figure.
- Here the use of 3' and 5' is a bit confusing. Figure 5b even names 3' cells. I would use the same nomenclature adopted in figure 1: reference (3') and query (5').
- Line 267-270 and figure S7 description are complex and should be rephrased. Please avoid using expressions such as “5'-to-3' experiment”, use query-to-reference instead.
- Line 272: here figure S6C should be cited.
- In the description of CITE-seq dataset analysis, it should be made clear that ground truth protein values are derived by smoothing of the measured expression.
- Line 771: V is not defined in the glossary.

- Line 873-875. Here it is stated that the top 2,000 variable genes across all cells were selected. But in the pbmc tutorial and also in the pbmc pre-built reference (both in the github rep), there are more than 2,000 variable genes. It looks like the function for variable genes selection performs a union of the variable genes identified in each batch. Is this the case?

- Line 957: I would not use U to indicate the matrixes since U is already used to indicate gene loadings.

Reviewer #3 (Expertise: single-cell data integration):

Summary

The manuscript presents a new pipeline, Symphony, to accelerate the mapping of the new query cells with a minimal change to the reference embeddings. Symphony compresses reference via building a linear mixture model first introduced by Harmony and assigns labels iteratively to the query cells in a low dimension embedding based on similarity. Symphony can efficiently store the reference data to allow the mapping of new cells. The potential reduction of training time to compress reference datasets and increase consistency in data visualization would be of interest to the general scRNA-seq community.

Major comments

1. The promise of fast mapping of new query cells can only be achieved with a comprehensive and ready-to-go reference dataset. It would be important to provide pre-built atlas level reference embeddings for 1) adult mouse from Tabula Muris, Tabula Muris Senis, Microwellseq, 2) adult human from Human Cell Atlas, and demonstrate their usability.

2. First assumption of Symphony is that "all cell states represented in the query data set are captured by the reference dataset". However, in practice, it is hard to know a priori all cell types in the query dataset. Thus, it still would be important for potential users to know how Symphony would handle novel query cell types or query cell types that do not have corresponding cell types in the training set.

3. The mapping time scales well with reference cell size. How well does it scale well with the number of cell types in the reference datasets?

4. The runtime analysis only included Symphony reference building, query mapping, and Harmony de novo. How does this runtime fare against that of other methods (Seurat v4, scArches, SCN, scmap-cell, scmap-cluster, and SCINA)?

5.It wasn't clear how the protein expression was inferred in Figure 6 and how the parameters (50-NN) were chosen.

6.What are some of the guiding principles for selecting a reference dataset?

Minor comments

1.To better demonstrate the accuracy of cell-type annotation, Fig.S3a should be included in the main figures.

2.After line 102 "Symphony builds upon the linear mixture model framework first introduced by Harmony.", the authors should emphasize the differences between the two algorithms.

Reviewer #1

Joyce Kang et al. introduce Symphony, a method for constructing scRNA-seq reference datasets and rapidly mapping query datasets to these references. The method builds on the earlier Harmony method developed by the same group. As large, high-quality scRNA-seq reference datasets are now able to be assembled, and their proper use can greatly improve the analysis of new query datasets, there is an urgent need for such “reference mapping” methods in the community and I am quite excited by the Symphony method developed here. However, I have several comments about the manuscript in its current form, which I hope can help to further improve the work.

We thank the reviewer for their enthusiasm about our manuscript.

Reviewer #1, Comment #1

The authors should explore what would happen if the query contains cell states that are not present in the reference. Although the authors state this is a condition for query mapping (first condition for Harmony de novo equivalence), users may not always know whether their dataset contains a new cell state before mapping, and so it is worth exploring what results are expected in this case and how users may identify the presence of novel states in a reference-mapping workflow. For example, Seurat generates a mapping score and a prediction score for each query cell, which can be used to assess the mapping confidence. Can Symphony provide any metric that could help identify poorly-mapped query cells?

We thank the reviewer for focusing our attention to the situation where query cell types are not represented in the reference dataset. It is true that in practice, this basic assumption for Symphony might be violated by users. Below, we describe our analyses where we assess the consequence of violating this assumption. As we (and the reviewer) expected, mapped cells may be assigned inappropriate identities. We agreed with the reviewer that there would be a benefit to have “mapping confidence” and “prediction confidence” scores that can indicate potential mismatches between the query and the reference, and flag situations where a cell type missing in the reference dataset is present.

To address the problem of *mapping* confidence, we have developed two new metrics: (1) **per-cell mapping metric** and (2) **per-cluster mapping metric**. The first metric gives a score to each query cell, whereas the second metric gives a score to user-defined *groups* of query cells representing putative query cell types (e.g. derived from clustering within the query internally). These metrics are based on Mahalanobis distance, which can be thought of as a multidimensional Z-score that measures how far away query cells/clusters are from the reference clusters in the low-dimensional harmonized embedding. These metrics can be used to identify individual query cells or cell clusters that are poorly represented by the reference. Higher distance metrics indicate lower confidence in the mapping. We have included new analyses demonstrating how these scores can be used (described below). To address the separate but

related topic of *prediction* confidence, we have added a **prediction confidence score** to the k-NN prediction function, which provides the proportion of neighbors supporting the predicted label (see **Reviewer #1, Comment #8** for more details on prediction confidence).

In new analyses, we show what happens when we map novel cell types in increasingly difficult scenarios using the fetal liver hematopoiesis dataset (**Supplementary Figs. 9-11**). In some scenarios, such as when the novel cell types are completely distinct from the reference (e.g. mapping non-immune cell types onto an immune-only reference, **Supplementary Fig. 9**), the per-cell and per-cluster metrics are able to clearly distinguish the poorly mapping cells (per-cell AUC=0.997, per-cluster AUC=1.0). In other scenarios, where the novel cell types are more similar to an existing reference cell state, the metrics may have more difficulty in identifying the novel cell type. For example, when Kupffer cells (specialized tissue-resident macrophages in the liver) are missing in the reference (**Supplementary Fig. 11**), they map onto the very closely related (direct precursor) “Monocyte-Macrophage” reference cell state (per-cell AUC=0.633, per-cluster AUC=0.963). Another example of a difficult situation is the novel Schwann cells in the pancreas analysis (see **Reviewer #2, Comment #1** for details). Currently, Symphony will tend to map novel cell states to the most similar reference state. In some cases, this behavior can actually be useful and biologically informative, as we demonstrate in a new analysis of mapping tumor cells onto the corresponding healthy tissue (see **Reviewer #2, Comment #2**).

We also compared the performance of our mapping confidence score metrics against the Seurat mapping confidence score, which was brought up by **Reviewers #1 and #2**. Note that because Seurat does not scale efficiently to datasets >100,000 cells (see **Reviewer #3, Comment #4**), we performed this comparison using the smaller 10x PBMCs dataset of ~25,000 cells for faster runtimes. For each of the 7 major cell types, we constructed a “missing cell type” scenario where the reference (3’v2 and 5’) is missing the cell type, and the query (3’v1) contains the cell type (simulating a “novel” query type) along with all other cell types. We then assessed how well each mapping metric could distinguish the missing type, as defined by AUC (which measures the ability to rank cells according to their probability of class membership, here missing vs. present in the reference). We find that the Symphony per-cell metric, per-cluster metric, and Seurat mapping scores offer comparable performance (AUCs below). Furthermore, the ability for mapping metrics to detect novel populations depends on the identity of the missing cell type (**Supplementary Fig. 12, 13a**), which is consistent with our observations from the fetal liver missing cell types examples.

Table of AUC values for each “missing cell type” experiment:

Missing population	Symphony per-cell metric	Symphony per-cluster metric	Seurat mapping score
B	1	1	0.99
DC	0.84	1	0.94
HSC	0.99	0	0.89
MK	0.69	1	0.55
Mono	0.92	1	0.95
NK	0.79	1	0.74
T	0.94	1	0.98

AUC values when combining all 7 experiments (same cutoff threshold across all experiments):

- Symphony per-cell AUC = 0.926
- Symphony per-cluster AUC = 0.994
- Seurat mapping score AUC = 0.961

In our thorough exploration of mapping metrics, we have learned that identifying poorly mapped cells may be a nontrivial problem. Mapping algorithms are specifically designed to map cells accurately despite the presence of potentially large batch effects and variable sequencing data. They are intentionally forgiving algorithms. Indeed, our colleagues have also noted that a single per-cell metric may be insufficient to fully capture the mapping confidence, as Satija and colleagues noted on the FAQ page for Azimuth: “Azimuth computes a series of metrics that relate to QC for the mapping procedure. We’ve found that a single metric is insufficient to describe the quality of mapping, and therefore compute [several metrics]”. Identification of poorly mapped cells may require additional investigation by users if these metrics flag problematic cells. We note some of the challenges around this issue in the updated manuscript in the **Discussion**, and highlight it as a potentially important future direction that the single cell analysis community should invest further effort in.

Added to Main Text:

Symphony helps identify query cell types missing in the reference

Although the first assumption of Symphony is that the reference is comprehensive, users may not always be aware if their query contains new “unseen” cell states prior to mapping. Symphony will typically map missing query states onto their most similar reference state(s) in these situations. To help users flag unseen cell states, we developed two metrics that help users detect and remove poorly mapping cells (**Methods**): (1) *per-cell* mapping metric and (2) *per-cluster* mapping metric. These metrics are based on Mahalanobis distance, a multivariate distance metric analogous to the univariate Z-score. They measure how far away query cells (1) or user-defined query clusters (2) are from the reference cell states in the low-dimensional embedding, where higher metrics indicate worse mapping.

In general, we found that these metrics were potentially useful for flagging novel cell types (**Supplementary Note 1**). For example, we tested the metric using the fetal liver hematopoiesis dataset described above. We found that the ability to call out a query cell type as novel depends on the cell type as well as what is present in the reference (**Supplementary Figs. 9-11**). For example, when the “missing” cell types are very different from the reference (mapping non-immune cell types like fibroblasts, endothelial cells, and hepatocytes onto an immune-only reference), the mapping metrics are able to clearly distinguish the missing cell states as novel (per-cell AUC=0.997, per-cluster AUC=1.0, **Supplementary Fig. 9**). In situations where the novel cell types are very similar to an existing reference cell state, the metrics may have more difficulty in identifying them. For example, when Kupffer cells (specialized tissue-resident liver macrophages) are missing in the reference (**Supplementary Fig. 11**), they map onto the closely related (immediate precursor) “Monocyte-Macrophage” reference cell state (per-cell AUC=0.633, per-cluster AUC=0.963). Our metrics are in general comparable to the Seurat mapping score, though different metrics offer the strongest performance under different scenarios (**Supplementary Note 1, Supplementary Fig. 12-13**).

Add to Discussion:

Identification of novel cell-types that have failed to map is an important future direction for mapping algorithms. To identify potentially novel cell-types, we provide two mapping metrics and a prediction confidence score to aid users in flagging and removing poorly mapping cells. We recognize that these metrics may be less informative in cases where the novel population is very similar to an existing reference population. Hence, Symphony does not entirely supplant the need for users interested in novel cell type discovery to conduct *de novo* analyses of the query alone.

Added as Supplementary Note 1: Mapping confidence metrics

Development of Symphony mapping metrics

Although Symphony inherently assumes that all query cell types are present in the reference, users may not always know whether their data contains novel (“unseen”) query cell types. To help identify these situations, Symphony provides two metrics that quantify how well query cells are represented by the reference: *per-cell* mapping metric and *per-cluster* mapping metric. Both metrics are based on the Mahalanobis distance, a multivariate distance metric which measures the distance from a point (vector in multidimensional space) to a distribution (**Methods**). The *per-cell* metric gives a value to each query cell, whereas the *per-cluster* metric gives a value to each (user-defined) query cluster. Because the metric measures distance, higher values indicate a greater difference between the query and reference and therefore a worse mapping. In order to handle a large range of potential query-to-reference dataset differences, we do not prescribe specific cutoff values to use in all situations. Rather, users can select a threshold above which to flag query cells/clusters warranting further investigation or removal from the mapping. We explored Symphony’s behavior when the query contains unseen cell types as well as the performance of the mapping metrics in the analyses below.

Mapping confidence vs. prediction confidence

As a point of clarification, we note that mapping confidence is separate but related to the concept of prediction confidence. Symphony’s prediction confidence score reflects certainty in the annotation transfer step when the reference is assumed to contain the query cell state. It assigns lower confidence to query cells that lie “on the border” between two reference states (**Methods**).

Testing mapping metrics in different missing cell type scenarios

We first tested the metrics using the fetal liver hematopoiesis dataset, in three increasingly difficult scenarios. In each scenario, we artificially remove cell type(s) from the reference dataset prior to reference building, then mapped a held-out query donor containing all 27 cell types, including the now “unseen” types. We assessed how well each mapping metric could distinguish the missing type, as defined by AUC (which measures the ability to rank cells according to their probability of class membership, here missing vs. present in the reference). In aggregate, these case study scenarios show that the Symphony mapping metrics can be extremely useful in identifying novel cell states. However, the metrics may lack sensitivity in detecting very fine-grained cell state missing in the reference. Symphony typically maps these query states to the most similar reference state.

Scenario 1: Reference missing non-immune cells

In the easiest scenario, the reference did not contain hepatocytes, fibroblasts, and endothelial cells (the non-immune cell types). After mapping the query containing all cell types (**Supplementary Fig. 9a**), we found that the unseen non-immune query cell types were clearly distinguishable as having worse per-cell and per-cluster mapping metrics compared to the cell types captured in the reference (per-cell AUC=0.997 and per-cluster AUC=1.0) (**Supplementary Fig. 9b-d**).

Supplementary Figure 9: Scenario where reference is missing non-immune cells. Unseen query cell types: endothelial cells (n=321 cells), fibroblasts (n=361), hepatocytes (n=306). **(a)** UMAP of harmonized embedding, with reference (n=89,566) shown as density colored by cell type and query cells (n=16,945) plotted with unseen states colored (and present states in gray), to highlight where the unseen cells map to. **(b)** Symphony *per-cell* mapping metrics calculated on the query cells, colored by whether cell types are unseen vs. seen, plotted by individual cell types as a boxplot (left, in descending order by mean) or aggregating all the unseen vs. seen cell types together in a violin plot (right). **(c)** AUC for the per-cell metric, measuring how distinguishable seen vs. unseen cells are. **(d)** Symphony *per-cluster* mapping metrics for each query cell type, with x-axis ordered the same as in **(b)**, colored by unseen vs. seen. Light gray shading indicates clusters too small to calculate the metric (num cells < 2 x dimensionality, **Methods**). **(e)** AUC for per-cluster metric across all query cells (all cells of the same cluster receive the same metric).

Scenario 2: Reference missing myeloid cells

In a more difficult example, we built a reference missing a subset of immune cells: all cells of the myeloid lineage. Upon mapping the query (**Supplementary Fig. 10a**), we found that the distance metrics for the unseen myeloid cell types are generally higher than for the seen cells (per-cell AUC=0.996, per-cluster AUC=0.996) **Supplementary Fig. 10b-d**). However, the distinguishability was somewhat lower compared to the first scenario, since the missing cell types are biologically more similar to the cell types in the reference. The distinguishability also varied by cell type along the myeloid lineage, where more differentiated myeloid cells (Kupffer cells and Mono-Mac) had the highest per-cell metrics (worst mapping). For unseen cell types that had the lowest metrics (better mapping), we found that they mapped onto biologically similar cell states in the reference. For example, the neutrophil-myeloid progenitor cells mapped onto reference hematopoietic stem cells (**Supplementary Fig. 10a**), which likely reflects their similar, less differentiated state. VCAM1+ erythroblastic island macrophages (VCAM1+ EI Macro.) cells are transcriptionally similar to both macrophages and erythroid cells [1]; supporting their mapping onto reference erythroid cells (**Supplementary Fig. 10a**).

Supplementary Figure 10: Scenario where reference is missing myeloid lineage cells. Unseen query cell types: Kupffer cells (n=6,022 cells), Mono-Mac (n=1,035), Monocyte (n=375), Monocyte precursor (n=44), DC1 (n=56), DC2 (n=292), VCAM1+ EI Macro. (n=52), Neut-myeloid prog. (n=91), DC precursor

(n=14), pDC precursor (n=9). **(a)** UMAP of harmonized embedding, with reference (n=64,049) shown as density colored by cell type and query cells (n=16,945) plotted with unseen states colored (and present states in gray), to highlight where the unseen cells map to. **(b)** Symphony *per-cell* mapping metrics calculated on the query cells, colored by whether cell types are unseen vs. seen, plotted by individual cell types as a boxplot (left, in descending order by mean) or aggregating all the unseen vs. seen cell types together in a violin plot (right). **(c)** AUC for the per-cell metric, measuring how distinguishable seen vs. unseen cells are. **(d)** Symphony *per-cluster* mapping metrics for each query cell type, with x-axis ordered the same as in **(b)**, colored by unseen vs. seen. Light gray shading indicates clusters too small to calculate the metric (num cells < 2 x dimensionality, **Methods**). **(e)** AUC for per-cluster metric across all query cells (all cells of the same cluster receive the same metric).

Scenario 3: Reference missing Kupffer cells

In the most difficult scenario, we built a reference missing Kupffer cells, which are liver tissue-resident macrophages. This scenario is especially difficult because the reference contains biologically similar macrophage and monocyte states. In this case, Symphony maps the unseen query Kupffer cells onto their immediate precursor (Monocyte-Macrophage) state in the reference (**Supplementary Fig. 11a**). The mapping metrics are not able to clearly distinguish the Kupffer cells as novel (per-cell AUC=0.633, per-cluster AUC=0.963) (**Supplementary Fig. 11b-d**).

Supplementary Figure 11: Scenario where reference is missing Kupffer cells. Unseen query cell type: Kupffer cells (n=6,022 cells). **(a)** UMAP of harmonized embedding, with reference (n=77,299) shown as density colored by cell type and query cells (n=16,945) plotted with unseen states colored (and present states in gray), to highlight where the unseen cells map to. **(b)** Symphony *per-cell* mapping metrics calculated on the query cells, colored by whether cell types are unseen vs. seen, plotted by individual cell types as a boxplot (left, in descending order by mean) or aggregating all the unseen vs. seen cell types together in a violin plot (right). **(c)** AUC for the per-cell metric, measuring how distinguishable seen vs. unseen cells are. **(d)** Symphony *per-cluster* mapping metrics for each query cell type, with x-axis ordered the same as in **(b)**, colored by unseen vs. seen. Light gray shading indicates clusters too small to calculate the metric (num cells < 2 x dimensionality, **Methods**). **(e)** AUC for per-cluster metric across all query cells (all cells of the same cluster receive the same metric).

Comparison of Symphony mapping metrics to Seurat mapping score

Next, we sought to systematically compare the performance of the Symphony mapping metrics to the Seurat mapping score, using the 10x PBMCs dataset described previously. Using reference datasets (5' and 3'v2), we iteratively removed one broad cell type from the reference prior to reference building, representing 7 different “missing cell type” scenarios: B, DC, HSC, Mono, MK, NK, and T. We built references using Symphony and Seurat for each scenario, mapped the query (3'v1) containing all cell types onto each reference, and then calculated the Symphony per-cell metric, Symphony per-cluster metric, and Seurat mapping score for each scenario (**Supplementary Fig. 12**).

Supplementary Figure 12: Symphony mapping metrics and Seurat mapping score across PBMCs missing cell type scenarios. We mapped a query (3'v1) containing all cell types onto references built with datasets (3'v2 and 5') with one major cell type artificially removed: B, DC, HSC, MK, Mono, NK, and T (total 7 “missing cell type” scenarios). **(a)** Symphony *per-cell* metrics for query cells across the scenarios (title of boxplot shows the missing type). Query cells are grouped by cell type and colored by seen (green) vs.

unseen (orange) in the reference for that scenario. Higher values indicate worse mapping. **(b)** Symphony *per-cluster* metrics for each scenario (1 value assigned to each query cluster), colored by seen (green) vs. unseen (orange). Higher values indicate worse mapping. Light gray “too few cells” bar indicates that the HSC cluster was too small ($n=21$ cells) to calculate the per-cluster metric (**Methods**). **(c)** Seurat mapping confidence scores for the same scenarios with Seurat reference mapping pipeline. Lower values indicate worse mapping.

When each method was permitted to select a unique cutoff value for each scenario to flag unseen cells, all three metrics performed comparably well (Symphony mean per-cell AUC=0.88, per-cluster AUC=0.86, Seurat AUC=0.86; **Supplementary Fig. 13a**). Consistent with our observations in the fetal liver scenarios, the ability for mapping scores to detect novel populations highly depends on the identity of the missing cell type (**Supplementary Fig. 13a**). For example, it is easier for all three methods to call out missing B or T cells as novel than it is to identify NK cells or MKs as novel. We next calculated the AUCs for each method by aggregating all cells from all 7 scenarios together and using “seen” vs. “unseen” as the label to predict for each cell. When methods were made to choose the same cutoff values across all 7 scenarios, the AUCs are also highly similar across the three metrics (Symphony per-cell AUC=0.926, Symphony per-cluster AUC=0.994, Seurat mapping score AUC=0.961; **Supplementary Fig. 13b**).

Supplementary Figure 13: ROC curves for Symphony metrics and Seurat mapping score across PBMCs missing cell type scenarios. AUCs were calculated across all query cells in each scenario using a binary label of missing vs. present in the reference as the ground truth for prediction. We generated ROCs for each metric in two ways: **(a)** considering each scenario separately (threshold values independent across scenarios) and **(b)** aggregating cells across all 7 scenarios together for a single calculation. For the Symphony per-cell metric and Seurat mapping score, each query cell is assigned its own value, whereas for the Symphony per-cluster metric, all cells from the same cluster are assigned the same value. The HSC

cluster (21 cells) was too small to calculate a per-cluster score and all HSCs were assigned a distance of 0 in all scenarios (unable to be flagged as novel) for inclusion in AUC calculations.

Add to Methods: Mapping confidence metrics

Symphony offers two scores that measure the confidence in query mapping. We recommend that users try both metrics and further investigate any query cells/clusters that appear to map poorly.

Background: Mahalanobis distance

Mahalanobis distance is a multivariate metric that measures the distance from a point to a distribution. It can be thought of as analogous to the univariate Z-score. We use Mahalanobis distance rather than Euclidean distance since Euclidean distance assumes uncorrelated features, whereas Mahalanobis distance accounts for potentially correlated features. PCA technically returns uncorrelated variables (which would have a covariance matrix containing zeros in all non-diagonal positions); however, when considering the distribution of cells surrounding each soft cluster individually (rather than all cells altogether), the covariance matrices have non-zero values. Mahalanobis distance (D) from a point x to a distribution with mean i and covariance matrix E in d -dimensional space is defined as:

$$D^2 = (x - i)' E^{-1} (x - i)$$

(1) Per-cell mapping metric. This metric measures the weighted Mahalanobis distance between each query cell and the distribution of reference cells they map nearest to, weighted by cluster membership. In the formula above, x is the query cell position (d -dimensional vector), and i and E are the weighted mean (i) and covariance matrix (E) for each reference Harmony soft cluster centroid in pre-Harmonized PC space, weighted according to the reference cells belonging to that cluster. For each query cell, we calculate its Mahalanobis distance (D) to each reference centroid then take the weighted average across all centroids the query cell belongs to (defined using R). Because the metric is a distance measure, it ranges from 0 to infinity. In practice, we have noticed that cell states well-represented in the reference tend to have values less than 10.

(2) Per-cluster mapping metric. This metric takes in a user-defined set of query cluster labels (e.g. putative cell types from running a *de novo* PCA pipeline on the query followed by graph-based clustering). User-defined clusters are likely to represent unique cell types within the query data. The intuition behind this metric is that if a query cell type is well-represented by the reference PC structure, then it should map closely to a reference centroid. We first project the query into reference pre-Harmony PCs, then calculate the Mahalanobis distance between the query cluster and its nearest reference centroid, where the covariance is defined using the query cluster in reference PC space. All cells in a given cluster receive the same score. By aggregating signal using multiple query cells per cluster rather than each cell individually, this metric potentially offers greater discriminatory ability than the per-cell metric. A disadvantage of the metric is that it is sometimes difficult to anticipate what the covariance of the query will be upon projection into reference PCs. Additionally, if a query cluster has very few cells, the estimation of its covariance matrix becomes numerically unstable; in practice, we return NAs for clusters smaller than $2d$, where d is the dimensionality of the embedding.

Add to Methods: Evaluating performance of Symphony mapping metrics

Simulating missing cell type scenarios in fetal liver hematopoiesis dataset

Using the 3' fetal liver dataset described above (n=113,063 cells), we held out one random donor (F8, 16,945 cells) as the query and used the remaining 13 donors as the reference dataset. We constructed 3 increasingly difficult scenarios where the reference is missing cell types present in the query by artificially removing cell types from the reference (cell types defined by original annotations):

1. Removing all non-immune cell types: endothelial cells (n=321 cells), fibroblasts (n=361), hepatocytes (n=306)
2. Removing all myeloid cells: Kupffer cells (n=6,022 cells), Mono-Mac (n=1,035), Monocyte (n=375), Monocyte precursor (n=44), DC1 (n=56), DC2 (n=292), VCAM1+ EI Macro. (n=52), Neut-myeloid prog. (n=91), DC precursor (n=14), pDC precursor (n=9)
3. Removing Kupffer cells (n=6,022 cells)

For each of the three scenarios, we built a Symphony reference using the same variable gene selection and reference building parameters as in the previous section ("Fetal liver hematopoiesis trajectory inference example"), then mapped the query containing all 27 cell types onto the reference. We calculated both per-cell mapping and per-cluster mapping metrics for the query cells. To plot ROC curves and calculate AUC values for each metric, we used the 'pROC' package in R (*roc* and *auc* functions), using a binary label of missing vs. present in the reference as the ground truth for prediction. Note that for the per-cluster metric, the pDC precursor (n=9 cells), DC precursor (n=14), and Pre-pro B cell (n=12) clusters were too small to calculate a per-cluster score and were assigned a value of 0 for the per-cluster metric in all scenarios (unable to be flagged as novel) for inclusion in AUC calculations.

Comparison of Symphony mapping metrics to Seurat mapping score

Using the 10x PBMCs dataset described above, we designated the 3'v2 and 5' data as the reference and held out the 3'v1 data as a query. For each of the major cell types (B, DC, HSC, MK, Mono, NK, or T), we artificially removed all reference cells of that type and built a Symphony reference with that type missing (total 7 references/scenarios). We used the same reference building parameters as the original 10x PBMCs analysis. We then mapped all query cells onto each reference, simulating 7 scenarios where the query contains a different novel unseen population, and calculated Symphony per-cell and per-cluster mapping metrics for the query cells in each scenario.

For each scenario, we also built a reference using Seurat (v4.0.2), integrating the reference dataset with *FindIntegrationAnchors* with 20 dimensions and mapping the query with *FindTransferAnchors* and *MapQuery*. We calculated query mapping scores with the *MappingScore* function. The Seurat mapping score is based on projecting the query into the reference space, then projecting back into the query and finding cells whose local neighborhoods are most altered by the transformation (see documentation).

We generated ROC curves and calculated AUCs across all query cells with the 'pROC' package in R (*roc* and *auc* functions), using a binary label of missing vs. present in the reference as the ground truth for prediction. For the Symphony per-cell metric and Seurat mapping score, each query cell was assigned its own value for the mapping metric, whereas for the Symphony per-cluster metric, all cells from the same cluster were assigned the same value. The HSC

cluster (21 cells) was too small to calculate a per-cluster score and all HSCs were assigned a value of 0 for the per-cluster metric in all scenarios (unable to be flagged as novel) for inclusion in AUC calculations. We calculated AUCs for each metric in two ways: (1) considering each scenario separately (threshold values independent across scenarios) and (2) aggregating cells across all 7 scenarios together into a single AUC calculation.

Reviewer #1, Comment #2:

Can other normalisation methods be incorporated into the Symphony reference-construction and mapping workflow? log-normalisation may not be the best normalisation method for scRNA-seq data, and more recent methods like GLM-PCA or SCTransform could give better results.

We thank the reviewer for this comment and agree that in some cases other normalization methods can produce different or potentially better results. Symphony is not tied to the log(CP10K+1) normalization. In designing Symphony, we made the core method agnostic to the specific approach used to normalize the data, as long as the reference and query datasets are normalized in the same way. Users can use the normalization method of their choice by using the `buildReferenceFromHarmonyObject` reference building function (where all preprocessing steps prior to the Harmony integration step are customizable). We have added text about alternative normalization strategies in the **Methods**.

We have also made the Symphony code compatible with the Seurat-based workflow (see new vignette at <https://github.com/immunogenomics/symphony/blob/main/vignettes/Seurat.ipynb>, which demonstrates how a user can use SCTransform normalization with Symphony). As an example of using alternative normalization strategies, we used the 'hcabm40k' dataset which is packaged with SeuratData, and performed two Symphony analyses using different normalization methods: SCTransform and log(CP10K+1). We found the resulting embeddings were highly concordant (see UMAP plots from vignette, reproduced below). As an additional example of using a different normalization strategy with Symphony, see **Reviewer #1, Comment #3** (below) which uses TF-IDF weighting/normalization.

Using log(CP10k+1) normalization

Reference consists of donors MantonBM1-4

Using SCTransform normalization

Added to Methods:

Normalization

Starting with the gene expression matrix for reference cells, we perform log(CP10K+1) library size normalization of the cells (if not already done). Log-normalization is recommended and performed by default (and used in all scRNA-seq analyses in the manuscript). However, Symphony can be used with other normalization methods, such as SCTransform⁶³ or TF-IDF (see scATAC-seq analysis). The only requirement is that reference and query datasets are normalized in the same manner.

Reviewer #1, Comment #3:

Can Symphony be extended to allow the construction and mapping of other data modalities, particularly scATAC-seq? Since the method works in low-dimensional space, I'd imagine this is possible with scATAC-seq data processed using LSI (which uses the SVD), although some aspects would need to be modified (storing reference mean and SD not required, you could perhaps store the inverse document frequency for each peak in the reference instead?)

We thank the reviewer for this creative idea. We agree that it would be useful to extend Symphony to be able to construct references and map queries for other single-cell modalities besides RNA (such as open chromatin regions via scATAC-seq). Encouragingly, the SnapATAC pipeline, recently developed by Fang et al. for the analysis of scATAC-seq data [2], uses Harmony for batch integration, so we felt that it might be possible to make a Symphony reference for this modality. As we mentioned above, Symphony is able to accommodate different normalization strategies, and we felt that TF/IDF weighting, commonly used in scATAC-seq analysis, might also be easily compatible with Symphony. We have explored the reviewer's suggestion in a new analysis. As suggested, we modified Symphony to be able to store the

inverse document frequency calculated on the reference cells rather than gene means and standard deviations.

To demonstrate functionality, we used an existing scATAC-seq dataset of hematopoiesis differentiation from Buenrostro et al. [3]. We chose this dataset because it is a popular published benchmark and has “known” ground truth cell types as defined by FACS [4]. We note that this is also an early dataset and has some challenges too. It has a small number of cells by current standards, lack of overlap between cell types represented across donors, and closely related cell types along differentiation. Nonetheless, with minimal modification to the existing Symphony pipeline, we were able to show that Symphony can build a reasonable scATAC-seq reference embedding which distinguishes cell types along different differentiation pathways similar to the original tSNE in Buenrostro et al. [3] and map a query dataset such that the query cells preferentially map to their corresponding reference cell types (**Supplementary Fig. 14**).

This analysis shows promise, and we plan to pursue scATAC-seq reference mapping as a future direction. scATAC-seq differs from scRNA-seq in that open chromatin regions (“peaks”) are typically defined within datasets, rather than having a set list of genes as features. In our analysis, we bypassed this issue by using shared peaks between reference and query for mapping. An important open problem is how to optimally select features to perform reference mapping on reference and query datasets that potentially derive from separate studies. One approach may be to remap query reads to the reference open chromatin regions or binning the genome into small (e.g. 500 or 1k bp) regions (added to **Discussion**).

ding Symphony to scATAC-seq data. We built a reference using a scATAC-seq dataset (Buenrostro et al.), then mapped a held-out donor as the query. **(a)** Diagram of the differentiation pathway of flow-sorted (“known”) cell types present in the reference (reproduced from Buenrostro et al. [3]) **(b)** Symphony reference embedding (n=1,736) built from all donors except BM1214, colored by “known” cell type. UMAP shows regions of related cell types along Lymphoid, Myeloid, and Erythroid differentiation pathways as in Buenrostro et al.[2]. **(c, d)** Symphony mapping embedding, colored by **(c)** reference or query or **(d)** “known” cell type. **(e)** Barplot showing, for each of the 3 “known” cell types present in the query (CMP, GMP, and pDC), the number of query cells predicted across each of the cell types by Symphony (5-NN). **(f)** Prediction confidence scores for the query cells, measuring the proportion of 5 nearest reference neighbors supporting the predicted cell type label, colored by whether the query was ultimately predicted correctly or not.

Added to Main Text:

Extension of Symphony to scATAC-seq data

We next wondered whether Symphony may be extended to other single-cell modalities, especially scATAC-seq. As a proof-of-concept analysis, we built a reference (n=1,736 cells) using a published scATAC-seq dataset of flow-sorted cells capturing hematopoietic differentiation [3,4], leaving out one donor (n=298 cells) to map as a query (**Supplementary Fig. 11**). We modified Symphony to use the

shared open chromatin peaks as input features rather than genes (**Methods**) and were able to map the query cells such that 84% of cells were assigned their “known” cell type or the immediate precursor type (**Supplementary Fig. 11d-e**).

Added to Discussion:

Single-cell reference mapping using modalities beyond scRNA-seq poses unique challenges. For example, in scATAC-seq, peaks are not standardized and are typically redefined by peak calling algorithms in each analysis. Hence, it is not immediately clear how to optimally select the best peak features to perform reference mapping when reference and query datasets have been analyzed with different peak sets. One approach may be to remap query reads to the reference open chromatin regions or binning the genome into small (e.g. 500 or 1k bp) regions.

Added to Methods:

Extending Symphony to scATAC-seq

scATAC-seq is different from scRNA-seq in that open chromatin peaks are typically defined in a dataset-specific manner (i.e. rather than a pre-specified list of genes that apply to all datasets). Hence, this proof-of-concept analysis was run on peaks called on all cells as defined by the benchmarking paper by Chen et al.⁵², obtained from the Pinello Lab Github: https://github.com/pinellolab/scATAC-benchmarking/blob/master/Real_Data/Buenrostro_2018/input/combined.sorted.merged.bed. In this dataset, peaks were called on each cell type aggregated separately then merged. The full peaks x cells matrix was calculated using chromVAR's getCounts function as demonstrated in https://github.com/pinellolab/scATAC-benchmarking/blob/master/Real_Data/Buenrostro_2018/run_methods/chromVAR/chromVAR_buenrostro2018_kmers.ipynb, and subsequently binarized. The cell type information was also gathered from the Pinello Lab Github https://github.com/pinellolab/scATAC-benchmarking/blob/master/Real_Data/Buenrostro_2018/input/metadata.tsv while the donor information was inferred from the cell name.

We defined the query cells (n=298) as those that belong to donor BM1214 while the remaining cells (n=1,736) were assigned as reference. BM1214 had cells corresponding to CMPs, GMPs, and pDCs, whose cell types all had cells from other donors also in the reference set. Since scATAC-seq is sparse and zero-inflated, the mean-scaling approach used for genes was changed to TF-IDF normalization on the binarized peaks x cells matrices. Seurat's TF-IDF function was modified to allow for an IDF vector as input and outputted the TF matrix, IDF vector, and normalized peaks x cells matrix. Following the TF-IDF implementation in Stuart & Butler et al. (2019)¹⁸, we computed $\log(\text{TF} \times \text{IDF})$. The inverse-document frequency (IDF) vector was calculated on the reference cells only and then used in the query cell normalization to get all the cells in the same space before mapping. With only this change to the Symphony methods, scATAC-seq query cells were mapped to a comparable reference. Feature selection, SVD, and Harmony were done as in the 10x PBMC analysis. Predicted query cell types were calculated using 5-NN. For plotting, we used the same cell type colors primarily defined from the Supplemental Data Table 1 of the original Buenrostro et al. (2018) paper with GMPs changed to a darker orange to better distinguish them visually from the CMPs and the 'unknown' cells changed from grey to black, allowing grey to be used as the 'null' color to better emphasize the other cell type colors.

Reviewer #1, Comment #4:

Under the section “Symphony maps against a large reference within seconds”, the authors claim to have shown that Symphony scales efficiently to map against multimillion-cell references. However, the authors did not actually demonstrate construction of a multimillion-cell reference, and there are other limitations that may become relevant as the datasets become much larger (the use of 32-bit sparse matrices, for example). To support their claim, the authors should actually demonstrate the construction of such a reference, and the mapping of queries to it.

We thank the reviewer for this comment and agree that to support our claim that Symphony supports multi-million cell reference, we should actually demonstrate it directly. In a new analysis, we used the recent Ren et al. (*Cell*, 2021) COVID-19 PBMC dataset which contains 1.46 million cells comprising 284 samples from 196 individuals (GSE158055). We split the dataset into a random 270 samples (1.39 million cells) for reference construction and held-out 14 samples (72,781 cells) for query mapping (**Methods**). Reference building took 17.7 hours elapsed time (18.8 hrs total if including UMAP step) using 48.5 GB of memory. Mapping the query took 11.0 seconds (62.6 seconds including UMAP projection). We have included this result in the text under the runtime section “Symphony maps against a large reference within seconds” and added **Supplementary Fig. 4** showing the mapping embedding.

Supplementary Figure 4: Symphony constructs and maps to a multi-million cell atlas. To demonstrate scalability to multimillion cell atlases, we used a large-scale scRNA-seq dataset (Ren et al., *Cell*, 2021). We built a Symphony reference of 1.39 million cells from 270 samples and mapped a held-out set of 14 samples (n=72,781 cells) as the query. UMAP plots show the resulting embeddings of reference and query cells, colored by author-defined major cell type.

As a technical point, we note per the reviewer’s point that the published gene expression matrix (from GSE158055) was indeed too large to directly read into R due to a limitation on 32-bit sparse matrices in R. The maximum number of non-zero values in a sparse matrix currently cannot exceed $2^{31}-1$ for the `Matrix` package. The `spam64` package in theory is able to handle larger matrices, but as of writing this, it currently does not support the `readMM()` function

to read in the matrix. Therefore, to bypass this issue, we performed all pre-processing for this dataset in python using `scanpy` before loading the expression matrix (subset by variable genes) into R for further processing and reference building with Symphony.

Added to Main Text:

To directly test Symphony's scalability to multi-million cell atlases, we built a reference of 1.39 million cells (270 samples) from a recent COVID-19 dataset [5] in 17.7 hours and mapped a held-out query of 72,781 (14 samples) in 11.0 seconds (**Methods, Supplementary Fig. 4**).

Added to Methods:

Constructing and mapping to multi-million cell atlas. We obtained the AnnData file for the dataset (GSE158055_covid19.h5ad) from <https://drive.google.com/file/d/1TXDJqOvFkJxbcm2u2-bM5RBdTQqv56w/view>. The link to the AnnData object was obtained from the following GitHub issue (response from user saketkc): <https://github.com/satijalab/seurat/issues/4030>. Due to a limitation on the 32-bit sparse matrices in R (the maximum number of non-zero values in a sparse matrix currently cannot exceed $>2^{31}-1$ for the `Matrix` package), the gene expression matrix (1,462,702 cells \times 27,943 genes) was preprocessed using the Python `scanpy` package. We $\log(\text{CP}10\text{k}+1)$ normalized the data and subset to 1,301 variable genes (list of variable genes was obtained from contacting the original authors). The remainder of the analysis was performed in R. We held out a random 5% of samples (14 samples, 72,781 cells) as the query and built a Symphony reference using the other 95% of samples (270 samples, 1,389,921 cells), integrating over 'Sample.name' and 'dataset' with $\theta = 2.5$ and 1.5, respectively, following the original publication. Reference building and mapping procedures were run on a Linux cluster with 4 cores and timed using the `system.time` function in R. UMAP steps were excluded from runtime as these are not inherent to the Symphony algorithm.

Reviewer #1, Comment #5:

In Figure 4, the immune cells in the Symphony reference seem to be split into two separate clusters of cells. Are these cells truly distinct (different immune cell types) or is this an artefact of the reference building? This also appears to be the case, although less extreme, for the beta and alpha cells.

We thank the reviewer for noting that the reference immune cells segregate into two distinct clusters. We have further examined the immune clusters, denoted `immune_1` (20 cells) and `immune_2` (7 cells) using `presto` [6] to test for differential expression between them. The left table below shows the top 5 differentially expressed genes per group, and the right table shows output for several marker genes. We found that indeed there are biologically meaningful differences between the two clusters: `immune_1` is likely macrophage (LYZ+, CD14+), and `immune_2` is likely mast cells (KIT+). However, given the extremely small number of cells in each cluster we decided to just aggregate them together under a single "immune" label for presentation.

immune_1	immune_2	non-immune	feature	group	avgExpr	logFC	statistic	auc	pval	padj
<chr>	<chr>	<chr>	<chr>	<chr>	<dbl>	<dbl>	<dbl>	<dbl>	<dbl>	<dbl>
ITGB2	RAC2	EPCAM	CD14	immune_1	0.8906322	0.7628579	90209.5	0.7687873	1.262186e-08	3.598987e-07
TYROBP	CPA3	PERP	LYZ	immune_1	3.3979756	2.7848067	108780.5	0.9270539	2.876471e-12	1.191603e-10
IFI30	GATA2	DSP	KIT	immune_2	2.9112932	2.8951355	41118.0	0.9989796	1.511501e-71	6.452851e-69
HCK	CD22	TUJSC3								
LAPTMS	TPSAB1	DSTN								

Reviewer #1, Comment #6:

Figure 4e: as well as reference-query mixing, it's important to measure how well preserved the original query low-dimensional structure is in the mapped embedding. This could be done per-batch in the query, using kNN-corr or a similar metric.

We thank the reviewer for the important point. The goal of reference mapping is to map query cells into the reference embedding, but not at the expense of disrupting the original query low-dimensional structure (i.e. as defined from PCA on each batch of query cells). Following the reviewer's suggestion, we have developed a new metric called "within-query k-NN correlation" (wiq-kNN-corr), described below in the updated **Methods**. We observe that Symphony and Seurat exhibit nearly identical wiq-kNN-corr, whereas scArches performs more poorly on this metric (**Fig. 4g**). We have also updated the LISI plots in **Fig. 4** and **Supplementary Fig. 6** to be boxplots rather than density plots (as was suggested in **Reviewer #1, Minor Comment #1**).

Added to Legend: (g) Degree to which the query low-dimensional structure is preserved after mapping, as measured by within-query k-NN correlation (wiq-kNN-corr, with k=500) calculated across all query cells, within each query donor. Vertical lines indicate the mean wiq-kNN-corr.

Added to Main Text:

Reference mapping should place query cells into the reference embedding, but not at the expense of disrupting the query's original low-dimensional structure. Therefore, we developed a new metric called within-query k-NN correlation (wiq-kNN-corr), which is similar to the k-NN-corr metric but instead measures how well the original query low-dimensional structure is preserved after mapping. Anchoring on each query cell, we calculate it's (1) distances to the k nearest neighbors in the original query PCA embedding within each query batch (in this case, donor) and (2) the distances to those same k cells after reference mapping. Then, wiq-kNN-corr is the Spearman correlation between (1) and (2), ranging between -1 and 1 where higher values

represent better retention of the sorted ordering of original neighbors. We observe that for $k=500$ Symphony and Seurat exhibit nearly identical wiq-kNN-corr (mean wiq-kNN-corr=0.59 in human, 0.55 in mouse for Symphony; 0.6 in human, 0.57 in mouse for Seurat), whereas scArches performs more poorly on this metric (0.19 in human, 0.13 in mouse) (**Fig. 4g**).

Added to Methods:

To assess how well the query low-dimensional structure is preserved in the mapped embedding, we developed a new metric called within-query k-NN-correlation (wiq-kNN-corr). For each query batch (here, donor), we run a standard PCA pipeline on the cells (using 20 dimensions and selecting 2,000 variable genes per batch using vst). Then, anchoring on each query cell, we calculate it's (1) distances to the k nearest neighbors in the query PCA embedding and (2) the distances to those same k cells after reference mapping. Then, wiq-kNN-corr is the Spearman correlation between (1) and (2), ranging between -1 and 1 where higher values represent better retention of the sorted original neighbor ordering. The calculation is similar to k-NN correlation described above, except instead of measuring the sorted ordering of reference neighbors in a *de novo* integration embedding, we measure the sorted ordering of *query* neighbors.

Reviewer #1, Comment #7:

My understanding is that CCA captures shared sources of variation across two matrices. Since the authors use CCA for defining a joint embedding for the CITE-seq reference, would this embedding be biased to only capture sources of variation that were present in both the RNA and protein assays? What would happen in the case where one modality captures variation that is not shared in the second modality (for example, protein separates CD4 and CD8 T cells whereas this separation is very difficult to detect in the RNA modality)? The authors should also compare with the multimodal reference construction method in Seurat v4 (WNN followed by supervised PCA).

We appreciate the reviewer's concern that CCA may be biased to capturing sources of variation present in both RNA and protein assays. We believe that RNA-protein integration is in itself an extremely interesting and challenging area, and is indeed a topic of great interest in the single cell analysis field [7]. The dataset used in the analysis is from a study that we have recently published, which used canonical correlation analysis (CCA) to integrate memory T cells assayed with CITE-seq [8]. In our experience, CCA works well for T cells and is able to distinguish populations that are blurred in scRNA-seq only data (e.g. better separation of CD4+ and CD8+ T cells; **Supplementary Fig. 15**). We agree that benchmarking against the recently published Seurat WNN approach [9] would be informative and represents a fruitful direction for future work, especially once the field has converged upon optimal metrics by which to compare multimodal embeddings.

However, the main point of the multimodal T cell analysis is to demonstrate that Symphony can be used with other linear starting embeddings beyond PCA (in this case, CCA) which has been used to integrate multimodal datasets [8]. We have modified the **Discussion** to clarify that there are several strategies to build multimodal references, including Seurat v4 WNN. Symphony should theoretically work on all approaches that are based on an initial linear projection. Since

multimodal analysis is relevant to only one of our benchmarks, we felt that a complete investigation is beyond the scope of this manuscript.

Added to Discussion:

As another example, multimodal single-cell integration is an important area of active research. For the CITE-seq analysis, we used one strategy (CCA) based on finding shared variation between modalities [8], but alternative approaches have been proposed [9,10] that may be optimal for specific applications.

Reviewer #1, Comment #8:

The explanation of how cell type labels are transferred from reference to query is unclear. Is it a simple majority vote using the label of the 5 nearest neighbours, or is the distance to each neighbour also considered? Also, how sensitive are the label predictions to the choice of k , and when should users alter the k parameter?

We thank the reviewer for pointing out that the cell type label transfer step is unclear. For the k -NN label transfer, we use a simple majority vote using the label of the k nearest neighbors; no distance information is used. Specifically, we use the k -NN classification function implemented in the R `class` package, which provides both a prediction and a probability (proportion of the votes for the winning class), which serves as the prediction confidence score. We have added details to the **Methods** and have modified the main text to make this more intuitive:

Added to Methods (under “Query label prediction and prediction confidence”): Once query cells are embedded in the same low-dimensional feature space as the reference, reference labels can be transferred to the query using any downstream model (e.g. k -NN, SVM, logistic regression) using the harmonized PCs as input. See the Pbmcbench benchmarking analysis which compares multiple downstream methods.

For most analyses presented, we use a simple and intuitive k -NN classifier (as implemented in the ‘class’ package in R), which uses majority vote with ties broken randomly. We provide a convenient wrapper function in the Symphony package (‘knnPredict’), which can optionally return the prediction confidence measuring the proportion of reference neighbors contributing to the winning vote. For k -NN prediction, we would recommend that users alter the k parameter so that it is ideally no larger than the number of cells in the rarest cell type of the reference. For example, if the reference contains only 10 cells of a rare cell type, then we recommend the user set k no higher than 10, to ensure that rare cell types in the reference have the chance of being predicted given a majority vote k -NN classifier.

Modified Main Text:

Once query cells are mapped into the reference low-dimensional feature embedding, users can choose any downstream model to predict query labels from the reference cells using their shared harmonized features as input (**Methods**). To demonstrate this, we used a simple and intuitive k -NN classifier to annotate query cells across 9 cell types based on the majority vote of each query cell’s 5 nearest reference cells in the harmonized embedding [...]

For the Symphony-based k -NN model, we also enabled the option for Symphony to leave cells as unclassified based on a “prediction confidence score” (**Methods**), which measures the proportion of reference neighbors with the winning vote.

To investigate how sensitive the label transfer is to the choice of k , we used the fetal liver hematopoiesis example to test values of k ranging from 5-50. We found that the median cell type F1 score (ranging from 0.82 to 0.84) and overall classification accuracy (ranging from 0.846 to 0.850) for the query were both highly stable over values of k . We have added this result to **Supplementary Fig. 8d** (below). We would recommend that users alter the k parameter so that it is ideally no larger than the number of cells in the rarest cell type of the reference. For example, if the reference contains only 10 cells of a rare cell type, then we recommend the user set k no higher than 10, to ensure that rare cell types in the reference have the chance of being predicted given a majority vote k-NN classifier. In general, if the cell types in the reference are clearly well-separated, k should not make a big difference; if the cell types are more mixed, then k might be more important.

To address the concept of a *prediction confidence score* for the k-NN classifier (brought up in **Reviewer #1, Comment #1**), we have augmented the k-NN prediction function in our package to return the proportion of reference neighbors used to make the assignment. This reflects the confidence in cell type assignment and can be used to identify cells that lie “on the border” between two annotated reference cell states. For the fetal liver example, we find that cells that are incorrectly predicted by 30-NN have lower confidence than cells that are correctly predicted (added as **Supplementary Fig. 8b**). In fact, prediction accuracy tracks closely with prediction confidence (**Supplementary Fig. 8c**).

Added to Legend: **(b)** Boxplots showing prediction confidence (measured as the proportion of nearest reference neighbors with the winning vote) across query cells for 30-NN, colored by correct vs. incorrect prediction. **(c)** Relationship between prediction confidence score (x-axis; proportion of 30-NN with winning vote) and prediction accuracy (y-axis; proportion of correctly classified cells), showing that the two measures track closely. Point size is the number of cells with a given prediction confidence score. Error bars show 95% C.I. using the binomial proportion confidence interval. **(d)** Median cell type F1 and overall classification accuracy across varying values of $k=5, 10, 30, 50$ used for query cell type prediction.

Added to Main Text (fetal liver section):

We first inferred query cell types with k-NN classification (**Methods**) and confirmed accurate cell type assignment based on the authors' independent query annotations⁴⁷, achieving median cell type F1 score of 0.83 and overall accuracy of 85.0% for $k=30$ (**Supplementary Fig. 8a, Supplementary Table 9**). Correctly predicted cells generally had a higher proportion of reference neighbors supporting the predicted label (**Supplementary Fig. 8b-c**). To assess sensitivity to the

parameter of k for inference, we tested values of k ranging from 5 to 50 and found that median F1 remained highly stable (0.82-0.84) across choices of k (**Supplementary Fig. 8d**).

As another example of using the prediction score, we use the example of predicting ground truth (as obtained by flow sorting) cell types in the Buenrostro scATAC-seq dataset (see **Reviewer #1, Comment #3**), which contains a continuum of closely related cell states undergoing differentiation (**Supplementary Fig. 14**) rather than clearly segregating cell types. We built a reference from all donors except one (BM1214), then mapped the query donor and inferred its cell types using 5-NN. Cells with incorrectly predicted cell types tended to have a lower proportion of neighbors supporting their prediction due to falling on the “border” between two reference cell types, showing the utility of the metric.

Additionally, we would like to emphasize that Symphony embeddings can be used to predict query labels from reference cells using additional strategies beyond a k -NN classifier (linear/logistic regression, SVMs, etc.), as demonstrated in **Fig. 3a** in the comparison to supervised classification methods. For most analyses in the manuscript, we used k -NN since it is simple and intuitive. Part of the **Discussion** (reproduced below) acknowledges other strategies for predicting cell labels:

Discussion:

We approach annotation transfer in two steps. We first learn a predictive model in the reference embedding, and then map query cells and use their reference coordinates to predict query annotations. In this two-step approach, Symphony mapping provides a feature space but is otherwise independent from the choice of downstream inference model. In PBMC type prediction (**Fig. 3a**), we used Symphony embeddings to train multiple competitive classifiers: k -NN, SVM, and logistic regression. In our specific analyses, we found that a simple k -NN classifier can achieve high performance with only 5-10 neighbors, and modestly outperformed SVM and logistic regression (**Fig. 3a**). In practice, users can choose more complex inference models if it is warranted for certain annotations.

Minor comments

Reviewer #1, Minor Comment #1:

The overlapping density plots shown in Figure 2C at first glance appear to show the Harmony methods with a density peak at 1, but these in fact are from the PCA plots below. An alternative visualisation could be used that would avoid this problem (violin plot or box plot for example).

We thank the reviewer for pointing this out. We have now updated Fig. 2c to use a boxplot instead of a density plot, as suggested.

Reviewer #1, Minor Comment #2:

The Seurat functions BuildSNN and RunModularityClustering aren't part of v3/v4. They were replaced by FindNeighbors and FindClusters. Which functions and Seurat version were used for clustering?

We thank the reviewer for catching this discrepancy. The function used for clustering was the internal (non-exported) Seurat function `Seurat:::RunModularityClustering` (Louvain implementation). The function used for building the nearest neighbor graph is actually not from the Seurat package but was actually the

`singlecellmethods:::buildSNN_fromFeatures` function from the `singlecellmethods` package (available on GitHub):

<https://github.com/immunogenomics/singlecellmethods/blob/master/R/buildSNN.R>. We

apologize for the incorrect attribution and have updated the **Methods** text to reflect this distinction and thank the reviewer for pointing this out.

Reviewer #1, Minor Comment #3:

The GitHub repository containing the code to reproduce the analysis is not accessible, so I was unable to review the code used.

We apologize for this inadvertent oversight. The code to reproduce the analyses is now available and accessible at https://github.com/immunogenomics/symphony_reproducibility.

Added to code availability:

Jupyter notebooks and scripts to reproduce figures are available at

https://github.com/immunogenomics/symphony_reproducibility.

Reviewer #1, Minor Comment #4:

The authors should make the R package available on CRAN on Bioconductor.

We are actively working on making the R package available on CRAN and it will be fully available once the manuscript is accepted for publication.

Reviewer #2

In this paper by Kang et al, entitled "Efficient and precise single-cell reference atlas mapping with Symphony", the authors present an algorithm to build integrated atlases and rapidly mapping query datasets. This mapping is much faster than performing de novo integration of the reference and the query and yields similar results. Moreover, it performs batch correction simultaneously to mapping, if necessary. As such, the reference atlas is frozen and not influenced by the query dataset. These are all useful and desirable functionalities in scRNAseq data analysis. The authors demonstrate Symphony capabilities on several datasets with complex experimental designs and compare Symphony performances with Seurat and scArches. The manuscript is well written, the methods section is accurate, and the description of the analyses is in general sound and convincing. I also congratulate the authors for the optimal implementation of the github page, with clear and documented tutorials, a rarity when reviewing yet unpublished tools.

We thank the reviewer for their positive and enthusiastic comments.

Reviewer #2, Comment #1:

In my opinion, the main limitation of this tool is the first condition that must be met for its use: the fact that all cell states in the query data set are captured by the reference dataset. Although "reasonable", it is not easy to satisfy this requirement. Often the user does not know a priori the composition of its dataset. In fact, the entire operation of mapping it to an atlas is performed to answer this very question. A dataset, even if obtained through cell sorting, could contain contaminant cells or unknown populations. When comparing organisms, as performed in the manuscript for pancreas populations in human and mouse, the comparability of cell populations is also an issue. Seurat, for example, assigns cells two different scores: mapping score and prediction score. The first reflect confidence that the cell is well represented in the reference, the second reflect confidence in the associated annotation. Is it possible to provide a similar mapping score for Symphony? E.g. in the Pancreas dataset analysis, Schwann cells are present in the query but not in the reference. These cells are mapped (to the most transcriptionally similar cell types I assume) but are not considered in accuracy estimation. Is there any way to assign a score that flag these cells as not represented in the reference? For example, are they far away from most centroids? A mapping score is important because a user could mislabel cells and the misuse them without realizing it.

We thank the reviewer for this helpful feedback and comments. We now offer two new metrics that help reflect the confidence in mapping, as well as one new metric for prediction confidence. Ultimately, we recommend that users try several metrics to evaluate their mapping and perform a

manual review for any suspicious query cells or cell clusters. For full details, please see Reviewer #1, Comment #1.

Using these metrics, we see that the mouse cells in the pancreas example appear to be comparable (slightly higher metrics) to the human cells. This is perhaps expected given that in order to perform the mapping, we “humanized” the mouse genes by using the corresponding orthologs to translate between the gene names of the two species; we do not see any evidence that there is a novel cell type in the mouse cells based on the metrics that we developed.

We find that the Schwann cells are a particularly difficult example, and neither Symphony nor Seurat’s mapping metrics (shown below) are able to distinguish them as potentially novel. They are a very small population in the query (<20 cells). The Symphony per-cluster metric generally works well (see Reviewer #1, Comment #1), but because it requires the calculation of query cluster covariance in low-dimensional space, it is challenging to define parameters accurately for rare populations; this makes identification of mismapping difficult. As described in Reviewer #2, Minor Comment #9,

the Schwann cells are mapped to stellate cells (the most transcriptionally similar type) by both Symphony and Seurat mapping as well as *de novo* integration methods.

We further note that the identity of these cells is somewhat uncertain. While they are labelled as Schwann cells by the original authors, the authors also noted that some of the markers were not consistent with conventional Schwann cells (quoted from Baron et al., *Cell Systems*, 2016):

“We hypothesize that this population of 13 cells represents pancreatic Schwann cells responding to injury. These cells express known Schwann cell markers such as SOX10, S100B, CRYAB, NGFR, PLP1, and PMP22. **However, components of the myelin sheath are lowly expressed or absent, and several genes shown previously to be upregulated in the Schwann cell response to nerve injury mark the population.** These genes include SOX2, ID4, and FOXD3, which are transcription factors associated with Schwann cell dedifferentiation and repression of myelin sheath component expression... Based on this profile of expression of known Schwann cell and injury markers, we characterize this cell population as Schwann cells that dedifferentiated under extraction and culture conditions.”

Hence, given the somewhat ambiguous identity of Schwann cells and extremely low population size in this example, we focus the discussion of confidence metrics in the new **Supplementary Note 1** on the fetal liver hematopoiesis and 10x PBMCs examples.

Reviewer #2, Comment #2:

Note that this could also be used on purpose to force the positioning of cells on a reference or on a trajectory. For example, the authors in lines 354/356 discuss about healthy and diseased samples. It would then be interesting to see what happens if we map tumor cells to an atlas that contain the same normal tissue. Can we discriminate cancer cells with stem or differentiated features?

We thank the reviewer for the suggestion and agree that it would be interesting to map tumor cells onto the corresponding healthy tissue. By “forcing” the positioning of tumor cells onto a healthy reference embedding, we may potentially find normal cell types that the tumor cells are most similar to. In a new analysis, we built a reference atlas of healthy fetal kidney cells from Stewart et al. (n=27,203 cells) [11]. For the query, we used a renal cell carcinoma (RCC) dataset (n=34,326) from Bi et al. [12], which includes both tumor cells and tumor-associated immune/stromal compartments. We find that tumor cells have higher per-cell metrics (indicating worse mapping) compared to the immune and stromal cells. Interestingly, the tumor cells map primarily to the reference “proximal tubule” and its immediate precursor state (“Medial S shaped body”), which is consistent with prior literature that RCC derives from proximal tubule cells [13]. We refer the reviewer to the new **Fig. 6** and additions to the Main Text (below) for more details.

Added to Main Text:

Symphony maps tumor-derived cells onto a healthy atlas

Given that Symphony maps unseen query cells to their most similar reference type, we hypothesized that Symphony may be able to map tumor-derived cells onto an atlas of corresponding healthy tissue. As an exploratory analysis, we built a reference (n=27,203 cells) of healthy fetal kidney [11] and mapped a renal cell carcinoma (RCC) dataset (n=34,326 cells) [12], transferring reference cell type labels to the query using 10-NN and comparing the predicted labels to the original annotations from Bi et al. (**Methods, Fig. 6**). As a sanity check, we observed excellent correspondence between the original and predicted annotations for immune and stromal cell types (**Fig. 6c**). We next examined the mapping results for the cells from the three tumor programs (TP1, TP2, and Cycling Tumor) originally defined by Bi et al. We found that TP1 and TP2 both primarily mapped to the reference “Proximal tubule” cell type and its direct precursor (“Medial S shaped body”); Cycling Tumor primarily mapped to “Medial S shaped body”,

“Proximal tubule”, and “Proliferating distal renal vesicle,” concordant with a more actively proliferating phenotype (**Fig. 6d**). These results are consistent with prior literature, as RCC has been thought to arise from proximal tubule cells [13]. Compared to the immune/stromal compartments, the tumor cells exhibited higher per-cell mapping metrics, indicating that they are less well-represented by the reference (**Fig. 6e**). This example demonstrates how intentionally mapping novel cell types, such as cancer cells onto a healthy atlas, can potentially provide biologically informative results.

Figure 6. Mapping tumor cells onto an atlas of healthy tissue. We built a reference of healthy fetal kidney (Stewart et al., 2019) and mapped a renal cell carcinoma dataset (Bi et al., 2021). **(a)** UMAP of healthy fetal kidney reference (n=27,203), colored by cell type as defined by the original publication. **(b)** Mapping tumor query dataset (which contains myeloid, lymphoid, stromal, and tumor compartments) onto the reference. Cells colored by reference (gray) or query compartment (as defined by original authors). **(c, d)** Heatmaps comparing original query cell types (rows), as defined by Bi et al., to the predicted reference cell types from Symphony (columns) for **(c)** immune and stromal compartments and **(d)** tumor cells. Color bar indicates the proportion of query cells per original cell type that were predicted to be of each reference type (rows sum to 1). Columns sorted by hierarchical clustering on the average gene expression (all genes) for the cell types to order similar types together. **(e)** Boxplot of per-cell mapping metric per query cell type (higher values indicate less confidence in the mapping), colored by tumor cells (orange) or immune/stromal (green) as defined in Bi et al.

Added to Methods:

We mapped a renal cell carcinoma dataset onto a reference of healthy fetal kidney cells (datasets in **Supplementary Table 1**).

Building the healthy kidney reference

We found that the reference dataset gene names were assigned using Gencode v24, whereas the query dataset gene names were assigned using Gencode v30 liftover37 (query dataset .gtf file was provided by Bi et al.). For many genes, the names were mismatched between the two versions (different synonyms for the same gene). Therefore, to sync the two datasets, we used the Ensembl IDs of the reference genes to "convert" them to Gencode v30 gene names. We used the top 2,000 variable genes across all cells to build the reference with 15 PCs, integrating over "Experiment" with $\theta=0.5$. Note that this reference building procedure is different from the original study (Stewart et al. 2019), which did not use Harmony. For improved readability, we collapsed cell type labels for immune and stromal cells (e.g. 'Proliferating monocyte' and 'Monocyte' were collapsed into 'Monocyte').

Mapping the renal cell carcinoma dataset

We mapped the query dataset starting from expression using default Symphony parameters, correcting for query 'donor_id'. Because some gene names remained discordant between reference and query datasets, the mapping was based on the 1,723 (out of 2,000) reference variable genes shared. We used 10-NN to transfer reference cell type labels to the query.

In lines 354-356 of the original submission (referenced by the reviewer), we wrote that: "...a reference with only healthy individuals is useful for annotation of cell types, while a reference with both healthy and diseased individuals is useful for annotation of cell types and pathological cell states." If a query diseased cell does not map well to a healthy reference, it is difficult to attribute the difference to a healthy vs. disease difference, or a batch difference between reference and query. Therefore, in general, we recommend that if the biological question is about healthy vs. diseased states, then ideally a reference containing both healthy and diseased cells is used. Users can map cases and controls onto a common reference, for example, and then quantify abundance differences across reference-defined cell states.

Reviewer #2, Comment #3:

Finally, is it possible to provide a "prediction" score for the 5-NN classifier, since this classifier is implemented in the symphony package?

We have added a prediction score to the k-NN prediction classifier that reports the proportion of neighbor reference cells that have the predicted label (ranging from 0 to 1). This score is helpful to identify cells that may fall "on the boundary" between two annotated clusters (see Reviewer #1, Comment #8 for details).

Minor comments

Reviewer #2, Minor Comment #1:

Is the mapping of each query batch independent from the rest of the query? i.e. if I map once a query composed of multiple batches or if I perform several mappings, one for each batch, do I get the same results? This would also be an advantage over performing de novo integration or using other tools since the inferred label would not change.

We thank the reviewer for this insightful question. In the Symphony model, all query cells from all batches play a role in parameter estimation. Hence, each query batch is technically not independent. We have updated the **Methods** to inform users of this. We believe the lack of independence can actually serve as a helpful feature in Symphony, especially for complex query datasets that may contain complex batch structure where batch correcting within the query is advantageous. In practice, for more simple queries, the effect is minimal. For example, for the fetal liver example, regardless of whether one maps all five query donors together vs. individually, we find that the overall cell type prediction accuracy is the same (0.85 for both cases).

Added to Methods (under “Mixture of experts correction”):

Note that mapping results may slightly differ based on whether one maps query cells all together (correcting for query batches) or maps each query batch separately. Because all query cells play a role in parameter estimation if mapped altogether, the batches are technically not independent.

Reviewer #2, Minor Comment #2:

Seurat is sometimes referred to as Seurat, Seurat v4, Seurat 3, Seurat 3 / 4. Also, at line 79 authors say that Seurat v4 is “compatible” with Seurat integration. This is a bit confusing (and not clear). Seurat 3 and 4 adopts the same exact anchor-based integration strategy, both for de novo integration and label transfer. The only difference (for what concerns the topic of this work) is that Seurat v4 introduces the mapQuery function that allows query projection onto the reference UMAP structure. Therefore the authors can simply refer to “Seurat” and specify the used version only in the method section.

We thank the reviewer for clarifying this confusing terminology. We have edited the manuscript text and figures to refer to Seurat as simply “Seurat” and have moved version details to the Methods for clarity. For the initial submission, we had used Seurat v4 beta (since Seurat v4 was still under development), but we have now installed Seurat 4.0.2 and reran all the analyses with the most updated version to remake **Fig. 4** and **Supplementary Figs. 5-6**. We now use Seurat v4.0.2 for all analyses in the manuscript.

Reviewer #2, Minor Comment #3:

In figure S1C Zr_corr should be replaced with Zr as written in table 2.

We thank the reviewer for catching the mismatch. We have made the change to **Supplementary Fig. 1** and legend.

Reviewer #2, Minor Comment #4:

The calculation of LISI is not clear; from what I understood, the value should range between the minimum and maximum number of categories. Why then in figure 2c values seem to go below 1 and above 3? How many neighbours are used for LISI calculation?

We thank the reviewer for pointing this out. The reviewer’s understanding is correct; in this example, LISI should range between 1 and 3. The values that seem to go below 1 and above 3 in

Fig. 3c were an artifact of the `geom_density_ridges` function in the `ggridges` package used for plotting. We have updated the figure to use a boxplot instead, which clarifies that the values range from 1 to 3 as expected (see **Reviewer #1, Comment #9** for updated plot). For the LISI calculations, we used the default parameters for `compute_lisi` (with perplexity = 30). Perplexity refers to the effective number of each cell's neighbors. We have added this to the **Methods**.

Added to Methods:

To compare dataset mixing between *de novo* integration and mapping, we calculated Local Inverse Simpson Index (LISI) using the `compute_lisi` function from <https://github.com/immunogenomics/LISI> with default parameters (perplexity = 30). Perplexity represents the effective number of each cell's neighbors.

Reviewer #2, Minor Comment #5:

Line 157: the use of "similarity" here is not intuitive. Authors should say that it is an elaboration of distance (see line 156). Moreover, the checkmark and x mark in figure S2 are misleading since they evocate "correct" and "wrong" but instead it is a matter of good and bad mapping.

We thank the reviewer for noting this confusing terminology. To clarify, we used the radial basis function (RBF) kernel to measure similarity as a function of squared Euclidean distance: $similarity(x,y) = \exp(-||x-y||^2/(2\sigma^2))$. Technically, since we are using Spearman correlation (rank-based), we can actually use distance directly rather than similarity and obtain the exact same correlation rho values. Therefore, for the sake of clarity and intuitiveness, we have rewritten this section to simply use the word "distance" rather than similarity, as the Spearman rho values for k-NN-corr remain the same either way. We have updated **Supplementary Fig. 2d-e** to reflect this change. We have also updated **Supplementary Fig. 2b-c** by replacing the checkmark and x-mark with the phrases "Good mapping" and "Bad Mapping".

Reviewer #2, Minor Comment #6:

What are the dashed lines in figures 2d and S2f?

We thank the reviewer for noting the confusing lines. The dashed lines in **Fig. 2d** and **Supplementary Fig. 2f** denote the mean value of k-NN-correlation across all query cells for a given mapping approach. In the previous version of the manuscript, there was a plotting error in **Supplementary Fig. 2f** where the mean was plotted as the mean across the entire dataset (rather than within each query facet separately). We have since fixed the error in the updated figure below and thank the reviewer for noting it. We have also modified the legends for both **Fig. 2d** and **Supplementary Fig. 2f** to clarify.

Correlation between query cell distance to 500 nearest gold standard reference neighbors in gold standard vs. alternative mapping embedding

Reviewer #2, Minor Comment #7:

Figure 2 refers to “harmony” embedding whereas the text talks mainly about gold standard embedding. I would uniformise for clarity.

We thank the reviewer for pointing out this confusing terminology. We have updated the text and legend of **Fig. 2** to uniformise to “gold standard” embedding (see **Reviewer #2, Minor Comment #17** for updated **Fig. 2**).

Reviewer #2, Minor Comment #8:

The 5-NN classifier is not described in the methods. How does it work? Sometimes the authors use different numbers of neighbors. How can a user tune this number?

We thank the reviewer for this question regarding the classification step. The k-NN classifier works as a simple weighted vote of the k nearest neighbors. In our package, users can modify the value of k . To help users interpret their results, we have added a prediction confidence score. We find that the prediction labels are quite stable regardless of choice of k (see **Reviewer #1, Comment #8** for details).

Reviewer #2, Minor Comment #9:

Schwann cells are shown in figure S4a but not in figure 4b. How are they classified after symphony mapping? De novo integration appears to locate them close to stellate cells.

We thank the reviewer for this insightful comment. Schwann cells are a small population of 19 cells in the query that are missing in the reference. They map closest to the reference stellate cells, and all 19 cells are predicted to be “stellate” by 5-NN. To explore whether these two cell types are transcriptionally similar, we applied a simple hierarchical clustering to the average expression profile for each query cell type (assigned in the Baron et al. human dataset) that shows that Schwann cells are indeed transcriptionally similar to stellate cells. To further quantify this, we calculated the correlation between each pair of cell types (based on average expression of the 2,236 reference variable genes across all cells belonging to that type in the Baron et al. human data). We see that stellate is the cell type with the highest correlation to Schwann cells (Pearson $r=0.68$). We now show Schwann cells in **Fig. 4b-d**. Furthermore, as we mentioned in **Reviewer #2, Comment #1**, the identity of these cells is somewhat uncertain given the original author’s observation that these cells lack expression of myelin sheath (which is characteristic of Schwann cells) and are hypothesized to represent a dedifferentiated population.

Encouragingly, the mapping result is consistent with the *de novo* integration results. As the reviewer noted, *de novo* integration locates the Schwann cells near stellate cells. Hence, we note that identifying Schwann cells as novel may be a difficult task: if *de novo* integration does not distinguish them as clearly distinct, then it will be hard for reference mapping to do so too. The *de novo* integration results for Symphony (left), Seurat (middle), and trVAE (right) are reproduced from **Supplementary Fig. 6**:

Reviewer #2, Minor Comment #10:

When comparing Symphony with Seurat in the Pancreas dataset, it is not clear if the authors used the labels predicted by the Seurat TransferData function.

In the original submission, we used 5-NN for all 3 of Symphony, Seurat, and scArches for comparability. For Seurat, this was based on Seurat's embedding of query cells in reference PCA space, as calculated in MapQuery. To address this comment, we reran the Seurat experiments using `TransferData` as well, and found the cell type classification results to be overall highly concordant: interestingly, 5-NN actually performed slightly better for predicting epsilon cells compared to the `TransferData` function. We added both sets of results (Seurat 5-NN and Seurat `TransferData`) to an updated **Supplementary Fig. 5c** and updated the **Methods**.

Reviewer #2, Minor Comment #11:

Line 243-248. Does this refer to figure S5C and S5D? If yes, please insert ref to the figure.

Yes, those lines refer to (now renumbered) **Supplementary Figs. 6c-d**. We have inserted a reference to those figures into the text.

Reviewer #2, Minor Comment #12:

Here the use of 3' and 5' is a bit confusing. Figure 5b even names 3' cells. I would use the same nomenclature adopted in figure 1: reference (3') and query (5').

We fully agree with the reviewer that the use of 3' and 5' is confusing. We have updated **Fig. 5b-d** and **Supplementary Fig. 7-8**, main text, and legends with the reviewer's suggested phrasing (also see next comment below).

Reviewer #2, Minor Comment #13:

Line 267-270 and figure S7 description are complex and should be rephrased. Please avoid using expressions such as "5'-to-3' experiment", use query-to-reference instead.

We thank the reviewer for noting the convoluted phrasing. We have rephrased the lines noted and **Supplementary Fig. 8** (renumbered from 7) legend to improve clarity. Part of the reason why the wording was complicated is that we initially presented results for “held-out donor” experiments (within the 3’ dataset only) in addition to the main query-to-reference experiment. For the sake of clarity and readability, we have since removed the held-out donor analysis and focus solely on the main query-to-reference experiment in both text and figures for this section.

Modified Supplementary Fig. 8a Legend: Fetal liver hematopoiesis cell type classification. We mapped the query (5’, n=21,414, n=5 donors) dataset onto the reference (3’, n=113,063 cells, 14 donors) and assessed cell type classification accuracy across 27 fine-grained cell types: **(a)** Cell type confusion matrix for 30-NN cell type classification, colored by the proportion of query cells in a given true cell type that was classified to each reference label (rows sum to 1).

Modified Main Text:

We first inferred query cell types with k-NN classification (**Methods**) and confirmed accurate cell type assignment based on the authors’ independent query annotations⁴⁷, achieving median cell type F1 score of 0.83 and overall accuracy of 85.0% for k=30 (**Supplementary Fig. 8, Supplementary Table 7**).

Reviewer #2, Minor Comment #14:

Line 272: here figure S6C should be cited.

We have added the citation to the figure (now renumbered **Supplementary Fig. 7**) in the text.

Modified Main Text:

To evaluate query trajectory inference, we used the Symphony joint embedding to position query cells from the MEM lineage (n=5,141) in the reference-defined trajectory by averaging the FDG coordinates of the 10 nearest reference cells (**Supplementary Fig. 7c**).

Reviewer #2, Minor Comment #15:

In the description of CITE-seq dataset analysis, it should be made clear that ground truth protein values are derived by smoothing of the measured expression.

We thank the reviewer for this comment. To make it clear that the measured protein expression was smoothed for the ground truth values, we have modified the Main text as follows (see Reviewer #3, Comment #5 for additional details).

Modified Main Text:

Then, we mapped the held-out query using only mRNA expression to mimic a unimodal scRNA-seq experiment, reserving the measured query surface protein expression for validation. To mitigate sparsity and variability in detection, we defined ground truth protein values using 50-NN smoothing of the measured values from CITE-seq (i.e. averaging the expression of 50 nearest neighbors in the embedding, **Methods**). We accurately predicted the surface protein expression of each query cell using the 50-NN average from the nearest reference cells in the harmonized embedding. For all proteins, we found strong concordance between predicted and ground truth expression (Pearson r: 0.88-0.99, **Fig. 7c-d**).

Reviewer #2, Minor Comment #16:

Line 771: V is not defined in the glossary.

We have added V to the glossary, clarifying that reference PC embedding $Z_r = \sum_r V_r r$ and that query PCA projection embedding $Z_q = \sum_q V_q q^T$.

Reviewer #2, Minor Comment #17:

Line 873-875. Here it is stated that the top 2,000 variable genes across all cells were selected. But in the pbmc tutorial and also in the pbmc pre-built reference (both in the github rep), there are more than 2,000 variable genes. It looks like the function for variable genes selection performs a union of the variable genes identified in each batch. Is this the case?

We thank the reviewer for recognizing this potentially confusing discrepancy. When generating **Fig. 2** (PBMCs analysis) included in the original submission version of the manuscript, we did select the top 2,000 variable genes across all cells, rather than within each batch (i.e. 3pv1, 3pv2, or 5p) separately. However, we have since moved to calculating variable genes within each batch separately then pooling them as the recommended variable gene selection approach. To make things consistent we have remade **Fig. 2** and have redone the 10x PBMCs analysis using top 1,000 variable genes within each reference donor then pooling them, which kept the total number of genes near (but not exactly) 2,000 as in the original version. The updated **Fig. 2** is reproduced below, and we have updated the **Methods** text for the analysis.

Reviewer #2, Minor Comment #18:

Line 957: I would not use U to indicate the matrices since U is already used to indicate gene loadings.

We thank the reviewer for catching the overloaded use of the symbol U. We have replaced the symbol “U” with “E” (for expression) in this section to improve clarity.

Modified Methods:

Mapping from mouse to human genes is then performed with matrix multiplication: $E_{\text{human}} = ME_{\text{mouse}}$. Note that while the mouse gene expression matrix $E_{\text{mouse}} \in \mathbb{Z}^{d \times N}$, the many-to-many mapping means that the mapped human gene expression matrix E_{human} may contain non-integers ($E_{\text{human}} \in \mathbb{R}^{D \times N}$).

Reviewer #3

Summary

The manuscript presents a new pipeline, Symphony, to accelerate the mapping of the new query cells with a minimal change to the reference embeddings. Symphony compresses reference via building a linear mixture model first introduced by Harmony and assigns labels iteratively to the query cells in a low dimension embedding based on similarity. Symphony can efficiently store the reference data to allow the mapping of new cells. The potential reduction of training time to compress reference datasets and increase consistency in data visualization would be of interest to the general scRNA-seq community.

We thank the reviewer for their overall positive comments. We agree that mapping to stable references would aid the interpretation and reproducible annotations for new query datasets.

Major comments

Reviewer #3, Comment #1:

The promise of fast mapping of new query cells can only be achieved with a comprehensive and ready-to-go reference dataset. It would be important to provide pre-built atlas level reference embeddings for 1) adult mouse from Tabula Muris, Tabula Muris Senis, Microwellseq, 2) adult human from Human Cell Atlas, and demonstrate their usability.

We thank the reviewer for this helpful comment. We agree that the value of single-cell reference mapping will be realized with high-quality, comprehensive references, and that providing pre-built reference atlases would greatly increase the resource value of Symphony.

Based on the reviewer's comment, we have added the Tabula Muris Senis (FACS) atlas (n=110,824 cells) as a comprehensive mouse atlas, as well as a mapping example in the latest version of the pre-built references tutorial on GitHub: https://github.com/immunogenomics/symphony/blob/main/vignettes/prebuilt_references_tutorial.ipynb. We agree that the adult human atlas from Human Cell Atlas (Tabula Sapiens) would also be useful; however, given that it is not yet peer-reviewed, we will defer building a Symphony reference from this dataset until cell type annotations are finalized. In the interim, we have focused on generating other (more focused) human atlases that we believe will be useful in many applications (**Table 1**). In total, we have generated a compendium of 8 pre-built references readily available with this initial release of Symphony (see **Table 1** for Zenodo links to download each reference). These include 10x PBMCs, pancreatic islet cells, fetal liver hematopoiesis, healthy fetal kidney, multimodal memory T cells, cross-tissue fibroblast atlas, cross-tissue immune atlas, in addition to the mouse atlas Tabula Muris Senis. In particular, the cell-type-focused fibroblast and T cell atlases have already proved useful in other studies from our group [14,15]. In the near future, our group has plans to provide additional references to the community as part of the Accelerating Medicines Partnership (AMP) Consortium, which aims to build large-scale, clinically

actionable atlases from healthy and diseased tissue samples of synovium in rheumatoid arthritis synovium and kidney in systemic lupus erythematosus.

The Symphony pipeline is able to efficiently construct large-scale references within hours, which will enable the greater single-cell community to incorporate reference construction and sharing as part of routine data sharing practices. We encourage users who use Harmony integration for integrative analyses to publish their harmonized atlas as a mappable object using Symphony in a public open-access repository such as Zenodo. Our package includes a user-friendly `buildReferenceFromHarmonyObj` function to help facilitate this, and we have provided many examples with this manuscript.

Table 1. A Compendium of Pre-built Symphony Reference Atlases

	Name	Description	Zenodo Link	Data source
1	10x PBMCs Atlas	Healthy human PBMCs (n=20,571) sequenced using three 10x protocols (3'v1, 3'v2, 5')	Link	10x Genomics
2	Pancreatic Islet Cells Atlas	Pancreatic islet cells (n=5,887) from 32 human donors; from 4 separate studies	Link	Segerstolpe et al. (2016) ⁴² Lawlor et al. (2017) ⁴³ Grun et al. (2016) ⁴⁴ Muraro et al. (2016) ⁴⁵
3	Fetal Liver Hematopoiesis Atlas	Human fetal liver cells (n=113,063) from 14 donors, sequenced with 10x (3')	Link	Popescu et al. (2019) ⁴⁷
4	Healthy Fetal Kidney Atlas	Human fetal kidney cells (n=27,203) from 6 samples	Link	Stewart et al. (2019) ⁶⁰
5	Memory T Cell (CITE-seq) Atlas	Human memory T cells (n=500,089) from a tuberculosis cohort (259 donors) assayed with CITE-seq	Link	Nathan et al. (2021) ⁵⁶
6	Cross-tissue Fibroblast Atlas	Human fibroblasts (n=79,148) from 74 samples spanning 4 inflammatory tissues and corresponding controls	Link	Korsunsky et al. (2021) ²⁵
7	Cross-tissue Inflammatory Immune Atlas	Immune cells (n=307,084) from 125 healthy or disease-affected donors across 6 inflammatory diseases	Link	Zhang et al. (2021) ⁶¹
8	Tabula Muris Senis (FACS) Atlas	Mouse cells from 23 tissues and organs (n=110,824 cells) across the lifespan.	Link	The Tabula Muris Consortium (2020) ⁶²

Symphony reference UMAP for Tabula Muris Senis (added to GitHub tutorial):

Added to Data Availability Statement:

Additionally, we provide a compendium of 8 pre-built Symphony references available for download on Zenodo (see **Table 1**).

Added to Discussion:

Instead of a single monolithic reference for all cell types across all tissues and disease, we expect the proliferation of multiple, well-annotated specialized references that focus on fine-grained modeling of diverse biological systems. In this initial release of Symphony, we provide eight pre-built reference atlases (**Table 1**) and an efficient, user-friendly pipeline to facilitate community expansion of high-quality references for the single-cell community. We encourage atlas builders to share their datasets as a mappable reference on open-access data repositories, such as Zenodo.

Reviewer #3, Comment #2:

First assumption of Symphony is that “all cell states represented in the query data set are captured by the reference dataset”. However, in practice, it is hard to know a priori all cell types in the query dataset. Thus, it still would be important for potential users to know how Symphony would handle novel query cell types or query cell types that do not have corresponding cell types in the training set.

We thank the reviewer for the helpful feedback. We agree that it is difficult to know whether the first assumption of Symphony is violated *a priori*. We have added new analyses that address this and now offer several confidence metrics that can aid users in identifying cells that may not be well-represented in the reference. For details, please see **Reviewer #1, Comment #1** for an extensive discussion and **Reviewer #2, Comment #2**.

Reviewer #3, Comment #3:

The mapping time scales well with reference cell size. How well does it scale well with the number of cell types in the reference datasets?

We thank the reviewer for this question. This is difficult to directly assess, since cell types can be defined at varying resolutions (e.g. fine-grained T cell subsets, rather than coarse-grained major cell types). For example, in the COVID-19 atlas (**Supplementary Fig. 4**), the authors defined 12 broad cell types which consist of 64 fine-grained types. Rather than directly measuring runtime as a function of number of author-defined cell types, we tested the number of principal components (d) and the number of centroids (k), as these measure the biological complexity captured in the reference, and may be what the reviewer is interested in interrogating.

To address this question, we kept the number of reference and query cells constant (at 50,000 and 10,000, respectively) and varied the number of reference soft clusters used in the mixture model (k) and number of dimensions in the embedding (d). We find that the query mapping time scales well as we increase the complexity of the reference. Results are summarized in **Supplementary Table 4** (reproduced below).

Main Text:

Importantly, Symphony mapping time does not depend on the number of cells or batches in the reference since the reference cells are modeled post-batch correction (**Methods**); however, it does depend on the reference complexity (number of centroids k and dimensions d) and number of query cells and batches (**Supplementary Table 4**) since the query mapping algorithm solves for the query batch coefficients for each of the reference-defined clusters.

Effects of # reference centroids and # dimensions			
k (# centroids)	d (# dimensions)	Reference building elapsed time (s)	Query mapping elapsed time (s)
Effect of # centroids (keep everything else constant)			
25	20	58.94	0.693
50	20	70.89	0.741
100	20	139.38	0.805
200	20	275.89	0.983
400	20	1781.44	5.106
Effect of # dimensions (keep everything else constant)			
100	10		219.04
100	20		142.43
100	40		270.97
100	80		176.37
100	160		300.96
100	320	567.17	1.36
50,000-cell reference (30 donors), 10,000-cell query (6 donors) for all experiments			

Reviewer #3, Comment #4:

The runtime analysis only included Symphony reference building, query mapping, and Harmony de novo. How does this runtime fare against that of other methods (Seurat v4, scArches, SCN, scmap-cell, scmap-cluster, and SCINA)?

We agree with the reviewer that benchmarking runtime against other methods would be informative. For this analysis, we benchmarked the Symphony reference building and mapping pipeline against scArches and Seurat v4, given that these methods (in contrast to SCN, scmap, and SCINA) fall under “true” reference mapping, which we define as placing query cells within the reference embedding (rather than assigning a hard label using supervised classification). We found that Symphony was the only method out of the three that was able to build references of >100,000 cells without excessive memory (>120 GB) or runtime (>24 hr) requirements. Note that in this analysis, we used Linux CPUs to run all jobs, whereas in practice, it would be preferable to use a GPU for neural-network-based methods such as scArches. We ran all three methods using CPUs (Linux machine) for comparability and because not all labs will necessarily have access to GPUs. We ran Symphony and Seurat using 4 cores. We recognized that scArches/trVAE would be better suited to have more computing, so we allotted it 48 cores instead of 4 cores.

C**F i g . 3 c . L e g e n d :**

Reference mapping runtime comparison between Symphony, Seurat, and scArches, for building different sized references (runtime measured in mins) and mapping different sized queries onto a 50,000-cell reference (runtime measured in secs, plotted on log scale). Note that all methods were run on Linux CPUs (Symphony and Seurat were each allotted 4 cores, scArches was allotted 48 cores). All jobs were allocated a maximum of 120 GB of memory and 24 hrs of runtime.

Added to Main Text:

Compared to alternative reference mapping approaches Seurat and scArches, Symphony was the only method to scale to large datasets (>100,000 cells) without requiring prohibitive memory (>120 GB) or runtime (>24 hr) requirements (**Fig. 3c**).

Added to Methods:

To compare runtime against Seurat and scArches, we used the same different-sized benchmark datasets and ran reference building and mapping or the corresponding *de novo* integration method (anchor-based integration for Seurat or trVAE for scArches). All jobs were allocated a maximum of 120 GB of memory and 24 hours of runtime (and automatically terminated if memory or runtime were exceeded). We measured reference building and mapping runtime and corresponding *de novo* integration runtime for each method as elapsed time starting from gene expression. All jobs were run on a Linux server: Symphony and Seurat were allotted 4 CPU cores, whereas scArches/trVAE was allotted 48 CPU cores to speed up runtime as it is a neural-net-based method.

Reviewer #3, Comment #5:

It wasn't clear how the protein expression was inferred in Figure 6 and how the parameters (50-NN) were chosen.

We thank the reviewer for bringing up this confusing part. We tried three different values of k (5, 10, 50; results shown in **Supplementary Fig. 12**, reproduced below) and $k=50$ performed the best. We have clarified that the query protein expression was inferred using the protein measurements averaged across the 50 nearest reference neighbors (see **Reviewer #2, Minor Comment #15** for revised Main Text).

Reviewer #3, Comment #6:

What are some of the guiding principles for selecting a reference dataset?

We thank the reviewer for this great question. We have added the following text to the **Discussion** to help frame some guiding principles for reference selection.

Added to Discussion:

Choosing which reference(s) to use is a key question in a reference-based analysis. When selecting a reference, one should consider (1) the relevance and comprehensiveness of the reference relative to the biological question of interest, (2) similarity of the cell-types being queried, (3) similarity of the technology used to assay the reference versus the query, (4) quality and resolution of cell-level annotations and any associated metadata, including the availability of additional modalities (e.g. CITE-seq), and (5) reference size (number of cells and samples included). For instance, a cell-type-specific embedding like the memory T cell reference (**Fig. 7**) may be able to capture more variability within a given cell type compared to an unsorted PBMCs reference (**Fig. 2**), which may better capture variability across multiple immune populations. Similarly, a reference with only healthy individuals is useful for annotating normal cell types, while a reference with both healthy and diseased individuals is useful for annotating both physiologic and pathologic cell states. It may also be useful to map the query to several references and consider the results in aggregate. For example, one may first map cells to a comprehensive atlas for the tissue or context of interest for coarse-grained annotations, then remap cells from certain cell types onto cell-type-specific references (e.g. T cell-only) for more fine-grained annotations.

Minor comments

Reviewer #3, Minor Comment #1:

To better demonstrate the accuracy of cell-type annotation, Fig.S3a should be included in the main figures.

We thank the reviewer for the suggestion. We have now included this figure panel into the main figures as **Fig. 3a**. An update is that in the original analysis, we used Symphony to assign a label to every query cell, whereas the other methods were allowed to have a “rejection option” leaving cells unclassified. Some methods that performed well on this benchmark had a nontrivial number of unlabeled cells (including the previous top performer, SCINA), see Figure S9 reproduced from Abdelaal et al. [16], below. Hence, to make the comparison more fair, we added new results for

the k-NN classifier versions Symphony, in which Symphony did not assign labels to query cells predicted with 60% confidence or less (only assigned a label to cells with >3 of 5 reference neighbors with the winning vote). We found that, across 48 experiments, this led to unlabeled percentages of: 13.5% of query cells for the variable genes version, and 13.9% of query cells for the differentially expressed genes version. Both of these versions (Symphony_vargenes_kNN_predconf>0.6 and Symphony_DEGs_kNN_predconf>0.6) outperformed all other methods on this benchmark. We have also added the F1 scores for these 2 new versions to **Supplementary Fig. 3**.

Figure S9. Percentage of unlabeled cells across the Pbmcbench datasets. (A) Heatmap showing the

Fig. S9 from Abdelaal et al. (“A comparison of automatic cell identification methods for single-cell RNA-sequencing data”), showing that other methods left some percentage of cells as unlabeled (i.e. not included in F1 score calculation).

Updated Fig. 3a:

Figure 3. Symphony matches performance of top supervised classifiers and scales mapping to large references within seconds. (a) Following the cross-technology PBMC benchmarking experiment from Abdelaal et al. (2019)³⁵, we ran a total of 48 train-test experiments per Symphony-based classifier. Two different versions of the Symphony feature embeddings were generated depending on variable gene selection method: top 2,000 variable genes (vargenes) or top 20 differentially genes (DEGs) expressed per cell type. Symphony embeddings were used to train 3 downstream classifiers: k-NN (k=5), SVM with radial kernel, and multinomial logistic regression (glmnet) with ridge. Symphony (orange) median cell-type F1 score across 48 train-test experiments compared to supervised methods (green), demonstrating comparability to top supervised methods and stable performance regardless of downstream classification method. “predconf>0.6” indicates option where only cells with >60% prediction confidence were included (4 or 5 out of 5 reference neighbors contributing to winning vote). Red dot indicates mean of median F1 scores across 48 experiments (used for ordering the methods along the x-axis). [...]

Added to Supplementary Fig. 3:

Updated Main Text:

We used the resulting harmonized feature embedding to predict query cell types using three downstream models: 5-NN, SVM with radial kernel, and multinomial logistic regression. The Symphony-based classifiers achieve consistently high cell type F1-scores (average median F1 of 0.79-0.87) comparable to the top three supervised classifiers for this benchmark (scmapcell, singleCellNet, and SCINA, average median F1 of 0.77-0.83; **Fig. 3a, Supplementary Fig. 3**). As discussed in Abdelaal et al.³⁵, some classifiers (including SCINA) leave low-confidence cells as “unclassified.” Hence, for the Symphony-based k-NN model, we also enabled the option for Symphony to leave cells as unclassified based on a “prediction confidence score” (**Methods**), which measures the proportion of reference neighbors with the winning vote. For this option, we only assigned labels for cells with >60% confidence (which excluded ~14% of cells). Notably, a limitation of this benchmark is that the reference in each experiment consists of a single dataset (no reference integration involved).

Updated Methods:

Given the resulting Symphony joint feature embeddings, we used three downstream classifiers to predict query cell types: 5-NN, SVM with a radial kernel, and multinomial logistic regression (glm_net with ridge). We note that other methods in the original benchmark were permitted to have a “rejection option” (leave uncertain cells as “unclassified” and not included in F1 score calculation). Hence, we also added a version for each of the two Symphony 5-NN versions that only assigned a label if the cell had >0.6 prediction confidence (at least 4 or 5 neighbors with the winning vote). A total of 8 Symphony-based classifiers were tested (2 gene selection methods * 3 downstream classifiers + 2 rejection option versions).

Reviewer #3, Minor Comment #2:

After line 102 “Symphony builds upon the linear mixture model framework first introduced by Harmony.”, the authors should emphasize the differences between the two algorithms.

We thank the reviewer for the suggestion. We have updated the text to make the distinction between the two methods clearer.

Modified Main Text:

Symphony builds upon the linear mixture model framework first introduced by Harmony¹⁷. Briefly, in a low-dimensional embedding, such as principal component analysis (PCA), the model represents cell states as soft clusters, in which a cell’s identity is defined by probabilistic assignments across one or more clusters. For *de novo* integration of the reference datasets (using Harmony), cells are iteratively assigned soft cluster memberships, which serve as weights in a linear mixture model to remove unwanted covariate-dependent effects. Then, Symphony compresses the reference into a mappable entity, leveraging the reference-learned model parameters to add new query cells to the embedding. It maps cells into the reference without any iterative assignment and keeps reference cells stable.

References

1. Popescu D-M, Botting RA, Stephenson E, Green K, Webb S, Jardine L, et al. Decoding human fetal liver haematopoiesis. Nature. 2019;574: 365–371.
2. Fang R, Preissl S, Li Y, Hou X, Lucero J, Wang X, et al. Comprehensive analysis of single cell ATAC-seq data with SnapATAC. Nat Commun. 2021;12: 1337.
3. Buenrostro JD, Ryan Corces M, Lareau CA, Wu B, Schep AN, Aryee MJ, et al. Integrated Single-Cell Analysis Maps the Continuous Regulatory Landscape of Human Hematopoietic Differentiation. Cell. 2018. pp. 1535–1548.e16. doi:10.1016/j.cell.2018.03.074
4. Chen H, Lareau C, Andreani T, Vinyard ME, Garcia SP, Clement K, et al. Assessment of computational methods for the analysis of single-cell ATAC-seq data. Genome Biol. 2019;20: 241.
5. Ren X, Wen W, Fan X, Hou W, Su B, Cai P, et al. COVID-19 immune features revealed by a large-scale single-cell transcriptome atlas. Cell. 2021;184: 1895–1913.e19.
6. Korsunsky I, Nathan A, Millard N, Raychaudhuri S. Presto scales Wilcoxon and auROC analyses to millions of observations. bioRxiv. 2019. p. 653253. doi:10.1101/653253
7. Lähnemann D, Köster J, Szczurek E, McCarthy DJ, Hicks SC, Robinson MD, et al. Eleven grand challenges in single-cell data science. Genome Biol. 2020;21: 31.
8. Nathan A, Beynor JI, Baglaenko Y, Suliman S, Ishigaki K, Asgari S, et al. Multimodally profiling memory T cells from a tuberculosis cohort identifies cell state associations with demographics, environment and disease. Nat Immunol. 2021;22: 781–793.
9. Hao Y, Hao S, Andersen-Nissen E, Mauck WM III, Zheng S, Butler A, et al. Integrated analysis of multimodal single-cell data. Cell. 2021; 2020.10.12.335331.
10. Gayoso A, Steier Z, Lopez R, Regier J, Nazor KL, Streets A, et al. Joint probabilistic modeling of single-cell multi-omic data with totalVI. Nat Methods. 2021;18: 272–282.
11. Stewart BJ, Ferdinand JR, Young MD, Mitchell TJ, Loudon KW, Riding AM, et al. Spatiotemporal immune zonation of the human kidney. Science. 2019;365: 1461–1466.
12. Bi K, He MX, Bakouny Z, Kanodia A, Napolitano S, Wu J, et al. Tumor and immune reprogramming during immunotherapy in advanced renal cell carcinoma. Cancer Cell. 2021;39: 649–661.e5.
13. Cairns P. Renal cell carcinoma. Cancer Biomark. 2010;9: 461–473.
14. Korsunsky I, Wei K, Pohin M, Kim EY, Barone F, Kang JB, et al. Cross-tissue, single-cell stromal atlas identifies shared pathological fibroblast phenotypes in four chronic inflammatory diseases. bioRxiv. 2021; 2021.01.11.426253.
15. Lagattuta KA, Kang JB, Nathan A, Pauken KE. Repertoire analyses reveal TCR sequence features that influence T cell fate. bioRxiv. 2021. Available: <https://www.biorxiv.org/content/10.1101/2021.06.23.449653v1.abstract>
16. Abdelaal T, Michielsen L, Cats D, Hoogduin D, Mei H, Reinders MJT, et al. A comparison of automatic cell identification methods for single-cell RNA sequencing data. Genome Biol. 2019;20: 194.

REVIEWERS' COMMENTS

Reviewer #1 (Remarks to the Author):

I'd like to thank the authors for their incredibly detailed and thoughtful revision. All of my original comments have been addressed, and I recommend acceptance of the manuscript.

Reviewer #2 (Remarks to the Author):

The authors satisfyingly answered to all my comments, the new introduced features greatly improved the manuscript.

Just 2 observations: in figure S14E it is not clear the separation between the barplots for the 3 groups.

I think that an important detail reported at page 37 is that reference and query must be normalized in the same manner. Since the authors provided precomputed references, I suggest that they highlight the type of normalization adopted. This could also be saved in a slot inside the symphony object.

Reviewer #3 (Remarks to the Author):

I thank the authors for their thorough revision. All my questions and concerns have been sufficiently addressed.

NCOMMS-21-05747C: Efficient and precise single-cell reference atlas mapping with Symphony

Point-by-point response to reviewers

Reviewer #1 (Remarks to the Author):

I'd like to thank the authors for their incredibly detailed and thoughtful revision. All of my original comments have been addressed, and I recommend acceptance of the manuscript.

We thank the reviewer for their positive feedback.

Reviewer #2 (Remarks to the Author):

The authors satisfyingly answered to all my comments, the new introduced features greatly improved the manuscript.

We thank the reviewer for their positive comments.

Reviewer #2, Comment #1:

Just 2 observations: in figure S14E it is not clear the separation between the barplots for the 3 groups.

We thank the reviewer for noting this. We have updated Supplementary Fig. 14e by adding dotted lines to make the separation between the 3 groups more clear.

Reviewer #2, Comment #2:

I think that an important detail reported at page 37 is that reference and query must be normalized in the same manner. Since the authors provided precomputed references, I suggest that they highlight the type of normalization adopted. This could also be saved in a slot inside the symphony object.

We thank the reviewer for noting that it would be helpful to highlight the type of normalization adopted in the reference so that the query can be normalized in the same way. We have updated the GitHub tutorial for pre-built references to indicate the type of normalization used for each reference; we have also updated the tutorial for how to build a reference to show how a user can save this information as a custom slot inside the Symphony object.

Reviewer #3 (Remarks to the Author):

I thank the authors for their thorough revision. All my questions and concerns have been sufficiently addressed.

We thank the reviewer for their positive feedback.